# Causal Modelling Agents: Causal Graph Discovery through Synergising Metadata- and Data-driven Reasoning

**Ahmed Abdulaal**[1]* **Adamos Hadjivasiliou**[1] **Nina Montaña-Brown**[1] **Tiantian He**[1]
**Ayodeji Ijishakin**[1] **Ivana Drobnjak**[1] **Daniel C. Castro**[2] **Daniel C. Alexander**[1]

[1]Centre for Medical Image Computing, UCL, London, United Kingdom
[2]Microsoft Research, Cambridge

## Abstract

Scientific discovery hinges on the effective integration of metadata, which refers to a set of conceptual operations such as determining what information is relevant for inquiry, and data, which encompasses physical operations such as observation and experimentation. This paper introduces the Causal Modelling Agent (CMA), a novel framework that synergizes the metadata-based reasoning capabilities of Large Language Models (LLMs) with the data-driven modelling of Deep Structural Causal Models (DSCMs) for the task of causal discovery. We evaluate the CMA's performance on a number of benchmarks, as well as on the real-world task of modelling the clinical and radiological phenotype of Alzheimer's Disease (AD). Our experimental results indicate that the CMA can outperform previous purely data-driven or metadata-driven approaches to causal discovery. In our real-world application, we use the CMA to derive new insights into the causal relationships among biomarkers of AD.

## 1 Introduction

Scientific discovery is the output of successful scientific inquiry, and its objects include events, causes, processes, and hypotheses. Arguably, the main mechanism of scientific reasoning is the generation of novel hypotheses that align with or explain observed datasets, or that allow for the derivation of testable consequences (Schickore, 2014). The 'pragmatic logics of discovery' refer to a set of reasoning strategies which are used in knowledge generation (Letseka & Zireva, 2013), and include both mental operations such as determination of relevant information for a given inquiry, and physical operations such as observation and experimentation (Schickore, 2014; Schiller, 1917). We refer to the former operations as 'metadata' and the latter as 'data'. Both metadata and data are foundational to producing, communicating, and validating conjectures, and are important aspects of several theories of the scientific method (Hanson, 1965; Godfrey-Smith, 2009). Whilst we can represent conjectures or hypotheses as Directed Acyclic Graphs (DAGs) in several fields (Spirtes et al., 2000; Sachs et al., 2005; Zhang et al., 2013), we are often faced with the challenging problem of inferring causal structure from its empirical implications, which is known as the causal discovery problem (Peters et al., 2017).

There are a number of algorithms which attempt to solve the causal discovery problem by identifying the correct DAG given a dataset (Zheng et al., 2018; Yu et al., 2019; Nauta et al., 2019). However, in attempting to identify the generative process by leveraging asymmetries in the numerical data alone, even state-of-the-art causal discovery algorithms can be ineffective on real-world datasets (Tu et al., 2019; Huang et al., 2021b; Kaiser & Sipos, 2022), and can struggle in the setting where the data consists of as few as five synthetic variables, where each pair of variables can at most have a single confounder (Ashman et al., 2023).

More recently, it was demonstrated that Large Language Models (LLMs) establish new state-of-the-art performance on multiple causal benchmarks including counterfactual reasoning, actual causality, and causal discovery (Kıcıman et al., 2023; Lampinen et al., 2023). It is hypothesized that LLMs are able to capture domain knowledge (as encoded in natural language), which can then be translated into causal graphs or used to identify background causal context. In essence, this describes metadata-based reasoning; a task previously assumed to be restricted to humans (Sahu et al., 2022; Trott et al., 2023).

---

*rmapabd@ucl.ac.uk

Coextensive to the causal discovery problem, causal modelling has seen numerous innovations in creating increasingly flexible models capable of causal reasoning on complex or even multi-modal data. For example, recent advances in probabilistic generative modelling have led to the emergence of Deep Structural Causal Models (DSCMs) (Pawlowski et al., 2020; Khemakhem et al., 2021; Sanchez & Tsaftaris, 2022; Dash et al., 2022), which combine modular Deep Learning (DL) elements with Structural Causal Models (SCMs). DSCMs can perform interventional and counterfactual queries in high-dimensional data settings, including in the imaging space (Pawlowski et al., 2020). However, they are limited in that they require the causal relationships between the variables (i.e., the causal graph) to be known *a priori*, which is rarely the case for real-world problems. Another limitation is the assumption of no unmeasured confounding, which requires yet more flexible types of graphical models (such as chain graph models) to account for hidden variables.

In this work, we investigate the effectiveness of combining LLM-based methods with a generalization of the DSCM framework by proposing the Causal Modelling Agent (CMA). The CMA combines the data-based modelling from DSCMs with the complementary and distinct metadata-based reasoning that LLMs utilise for the task of causal discovery, including for multi-modal datasets. Combining LLM-based methods with data-driven causal modelling approaches is beneficial in two main ways: 1) The LLMs can act as proxies of human knowledge and allow for the efficient exploration of causal graph space, and 2) data-driven causal methods can allow LLMs to formalize, communicate, and ground their reasoning (Kıcıman et al., 2023). We assess this framework on a number of synthetic experiments and causal discovery benchmarks, before applying it to the real-world task of modelling the clinical and radiological phenotype of Alzheimer's disease (AD). Our main contributions are: 1) A unified framework for causal discovery which combines LLMs with deep probabilistic graphical models, and in which the LLMs act as priors, critics, and post-processors over the training of such models; 2) a generalised approach for the automatic construction of a DSCM from an experiment-description file, which allows for flexible experimentation; 3) a novel modelling strategy which parametrises chain graphs with deep learning elements to account for unmeasured confounding; 4) in our real-world application, we use the CMA to derive new insights into the causal relationships among biomarkers of AD.

## 2 RELATED WORK

**Deep Structural Causal Models** A multi-modal DSCM framework is proposed by Pawlowski et al. (2020). Whilst previous work has extended DSCMs and similar causal models to a number of application areas (Reinhold et al., 2021; Li et al., 2023a), previous DSCMs require that the causal graph be known a priori. As a general model class for counterfactual inference in multi-modal data, DSCMs do not directly attempt to solve the causal discovery problem and have not previously been used as part of a causal discovery algorithm. Additionally, there is no method with which to handle unmeasured confounding/associative relationships.

**LLMs and Causality** Data-driven causal reasoning methods are well-established (Li et al., 2023b), but LLMs' application in this field is recent (Willig et al., 2022). Zečević et al. (2023) found that LLMs may recite causal knowledge instead of reason per se, with improved performance noted under Chain of Thought (CoT) prompting (Wei et al., 2022). Yadlowsky et al. (2023) showed LLMs' capacity to learn new tasks deteriorates with tasks increasingly divergent from their pretraining data. Contrarily, (Lampinen et al., 2023) observed that transformer-based agents can passively learn and apply generalizable causal strategies if allowed intervention during tests. In light of this nascent research field, a number of encouraging results have been demonstrated for the task of causal discovery. For example, Long et al. (2023b) demonstrated that LLMs can construct correct 3–4 variable graphs. Tu et al. (2023) considered causal discovery for a neuropathic pain dataset (Tu et al., 2019) using LLMs alone. Choi et al. (2022) demonstrated that LLMs can produce a prior hypothesis which improved the accuracy of data-based causal discovery algorithms, and Zhiheng et al. (2022) attempted to partially redefine the causal discovery problem such that it included relevant metadata. In these cases, LLMs were used in one stage of the causal modelling process, centring mostly around producing a causal graph alone or creating a causal graph which is used as a prior, for example in Ban et al. (2023). Long et al. (2023a) demonstrated that LLMs can reduce the size of a Markov equivalence class assuming an optimal output from a discovery algorithm; their work viewed LLMs as a post-processing step alone. Kıcıman et al. (2023) investigated the graph discovery capabilities of LLMs over a broader set of real-world datasets; however, they did not assess combinations of LLMs with existing causal methods.

**Agents and Reasoning**    In our setting, an agent is an LLM which has access to one or more 'tools', such as an internet search engine. We briefly summarise recent agent frameworks for reasoning tasks and compare them with our agent, whose tool is a causal modelling framework. The ReAct framework (Yao et al., 2022) aimed to combine reasoning and acting capabilities in language models by using reasoning traces to design and update action plans. Reflexion (Shinn et al., 2023) built on ReAct by adding a 'self-reflection' mechanism to aid in inferring future actions, and AutoGPT (Fırat & Kuleli, 2023) is a framework which decomposed overarching goals into sub-goals, carrying them out in a ReAct-like loop. DERA (Nair et al., 2023) attempted to enhance task completion using dialogue between two LLM agents, and Generative Agents (Park et al., 2023) stored agent experiences as memories which can be retrieved for future planning, however actions here were not executable. An extension of these ideas was Voyager (Wang et al., 2023), an LLM-based embodied agent which combined an automatic curriculum with an iterative prompting mechanism. These frameworks did not consider LLMs for the task of causal modelling, or indeed for inferring causal structure from its empirical implications.

**Related topics**    Additional related topics include works by Feder et al. which introduced the statistical challenge of estimating causal effects with text (Feder et al., 2022), and investigated how language representation models can effectively learn a counterfactual representation for a given concept (Feder et al., 2021). Veitch et al. (2021) investigated counterfactual invariance in the context of text classification, and Abraham et al. (2022) introduced a benchmark for assessing concept-based explanation methods in natural language processing (NLP). Mind's Eye (Liu et al., 2022) attempted to ground language model reasoning through simulation. The MuJoCo physics engine was used to simulate outcomes for given physics questions, and the results were passed to the LLM to improve model reasoning ability. However, this process did not iterate nor consider a causal modelling task. To the best of our knowledge, this is the first work which combines LLM-based methods with a general deep causal modelling approach in which the LLM acts as a prior, critic and hypothesis engine, as well as post-processor. Additionally, in contrast to previous work (Long et al., 2023a; Ban et al., 2023), the CMA does not make any explicit assumptions about the initial graph structure, and is assessed on complex data scenarios beyond synthetic cases.

## 3    CAUSAL MODELLING AGENT (CMA)

We introduce the CMA framework (Figure 1), which unifies metadata- and data-based modelling paradigms to reason over a dataset and its associated data-generating process. Conceptually, the framework is defined as a function that takes as input a dataset $D$ and associated metadata $M$, which represents the knowledge in the LLM's training corpus, and outputs a trained causal model and associated causal graph. The framework makes use of four concepts: 1) Hypothesis generation, 2) Model fitting, 3) Post-processing, and 4) Hypothesis amendment.

**Hypothesis generation**    In hypothesis generation, we produce a causal graph by use of an LLM. The only required input is a set of variable names. In our case, during the first iteration ($t = 0$), the language model function LLM acts as a prior which proposes a representation of a causal graph $\mathcal{G}_{t=0}$ of the current variables given metadata $M$ and an empty graph $\mathcal{G}^{\emptyset}$ (that is, a causal graph with no relationships between variables): $\mathcal{G}_{t=0} = \mathsf{LLM}(\mathcal{G}^{\emptyset}; M)$. Graph $\mathcal{G}_{t=0}$ can be encoded into an appropriate structured format $\mathcal{G}^s_{t=0} = e(\mathcal{G}_{t=0})$, where $e(\cdot)$ is an encoding function and $\mathcal{G}^s_{t=0}$ is a structured causal graph (Figure 1a).

**Model fitting**    At iteration $t$, the model fitting stage uses a data-driven approach to calculate a metric of fit $F_t$. As an example, the metric of fit could represent the log-likelihood of a dataset $D$ under the model, given a structured graph $\mathcal{G}^s_t$: $F_t = \log P_{\mathcal{G}^s_t}(D)$. Indeed, this will be the metric of fit considered in this work. In our instantiation of a CMA, we use a modelling approach which requires that we account for the order in which modules are constructed to avoid errors and model misspecifications. Therefore, let $\mathcal{K}$ be a function which takes as input $\mathcal{G}^s_t$ and transmutes it into a (correctly specified) computational graph $CG_t$, that is, $CG_t = \mathcal{K}(\mathcal{G}^s_t)$. Then, the model fitting stage is defined:

$$F_t := \log P_{G^s_t}(D) = ME(D, CG_t), \tag{1}$$

where $ME$ is a 'Modelling Engine' function which is described in more detail (alongside the motivations and specific choices for the function $\mathcal{K}$) in Appendix A.1.4 (Figure 1c). The model fit $F_t$ is stored at this stage.

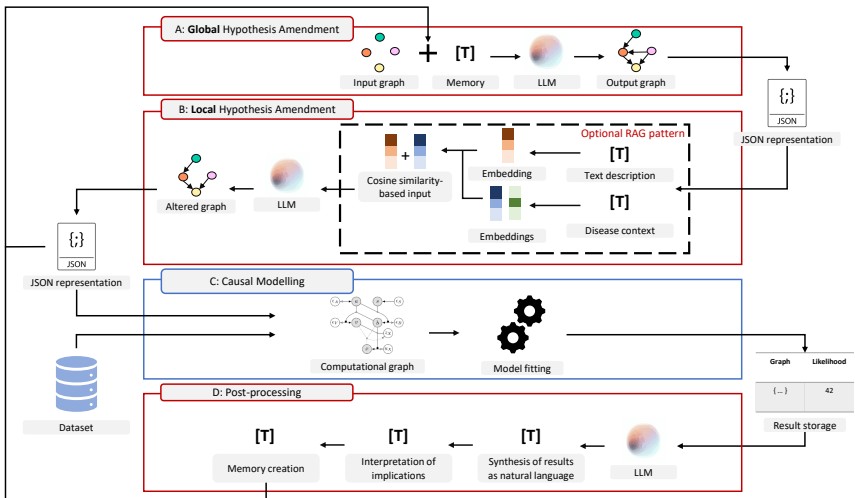

Figure 1: A schematic of the CMA. The CMA designs a hypothesis which is represented in a JSON file (Panels A and B), which is then transmuted into a computational graph and trained on data (Panel C). The data can be a set of scalar variables or a combination of scalar variables and images. The CMA then post-processes the results, producing a 'memory', which keeps track of previous amendments and their impact on model fit (Panel D). This process then iterates. The CMA can critique hypotheses on a whole-graph basis (Panel A), or a per-edge basis (Panel B). During the local hypothesis amendment phase, an optional Retrieval Augmented Generation (RAG) pattern can be used (Panel B; dashed black) to provide additional context for decision-making. The red rectangles indicate LLM-based modules, whilst the blue rectangle indicates data-driven modelling.

**Post-processing** The post-processing stage is used to produce a 'memory' $\mu$. This stage requires as input the graph at the current iteration $\mathcal{G}_t^s$, the graph at the previous iteration $\mathcal{G}_{t-1}^s$, and their associated metrics of fit, $F_t$ and $F_{t-1}$, respectively. The function $\mathsf{LLM}_\mu$ produces a memory at iteration $t$ as $\mu_t = \mathsf{LLM}_\mu(\mathcal{G}_t^s, \mathcal{G}_{t-1}^s, F_t, F_{t-1})$. Memories encode information about changes to the causal graph, their implications, and their impact on model fit, and are stored at this stage.

**Hypothesis amendment** Hypothesis amendment is divided into two phases, which we call the 'global' and 'local' phases. In the global phase, the function $\mathsf{LLM}$ acts as a critic which proposes a set of amendments $\mathcal{A}_{\text{glob}}$ by taking as input the (structured) graph at $t-1$, metadata $M$, and an optional memory $\mu_{t-1}$: $\mathcal{A}_{\text{glob}} = \mathsf{LLM}(\mathcal{G}_{t-1}^s, \mu_{t-1}; M)$. This phase is expected to keep track of high-level relationships and account for the overarching structure of the DAG, and can be guided by previous amendments through the memory system (Figure 1a). It should be noted that the hypothesis generation stage can be seen as a special case of the global amendment phase, which takes an empty graph as input (with no memory).

The local phase considers pairwise comparisons between vertices $u, v \in V, u \neq v$, where $V$ is the set of vertices in the structured causal graph. This phase makes amendments $\mathcal{A}_{\text{loc}}$ as:

$$\mathcal{A}_{\text{loc}}(u, v) = \begin{cases} \text{ADJUST}(u, v; M) & \text{if edge exists between } u \text{ and } v \\ \text{ASSESS}(u, v; M) & \text{otherwise,} \end{cases} \tag{2}$$

where $\text{ADJUST}(u, v; M)$ is an LLM-based function that outputs one of three actions: $\{\text{keep}, \text{remove}, \text{reverse}\}$, and $\text{ASSESS}(u, v; M)$ is a similar function which outputs one of the following options: $\{\text{No direct causality}, u \rightarrow v, v \rightarrow u\}$. We can augment the metadata of the LLM to $M \cup C$, where $C$ is additional domain-specific context. This is known as a Retrieval Augmented Generation (RAG) pattern (Lewis et al., 2020) and can be used to encourage or enforce specific relationships according to domain expertise (Figure 1b). This is expected to introduce edges (under specific domain considerations), and otherwise to critique, validate, and/or prune the edge outputs from the global phase. Hypothesis amendment at iteration $t$, $\mathsf{HA}_t$, can be seen as a compositional set of amendments to the structured graph from iteration $t-1$:

$$\mathsf{HA}_t(\mathcal{G}_{t-1}^s, \mu_{t-1}; M) = \{\mathcal{A}_{\text{glob}}\} \cup \{\mathcal{A}_{\text{loc}}(u, v) \,|\, \forall u, v \in V, u \neq v\}. \tag{3}$$

The hypothesis amendment, model fitting, and post-processing stages (Figure 1a-b, 1c, and 1d) iterate throughout a given experiment. The order of iteration is shown in Algorithm 1. In our implementation of the CMA, the model fitting stage creates a DSCM (Pawlowski et al., 2020), which enables reasoning over multi-modal datasets. We additionally propose a novel modelling approach, whereby we parameterise chain graph models with DL elements to represent associative but not necessarily causal links (for example, due to unmeasured confounding between the endogenous variables). Additional background on DSCMs and chain graphs, as well as further implementation details, can be found in Appendix A.1.

---

**Algorithm 1** Iterative procedure of the CMA Framework

---

**Require:** Empty graph $\mathcal{G}^\emptyset$, data $D$ and metadata $M$
    **Iteration 0**
1: $\mathcal{G}_0^s := e(\mathsf{LLM}(\mathcal{G}^\emptyset, \emptyset; M))$               ▷ *Hypothesis Generation (**no** memory)*
2: $F_0 := \log P_{\mathcal{G}_0^s}(D)$                              ▷ *Model Fitting*
    **Iteration 1**
1: $\mathcal{G}_1^s := e(\mathsf{HA}_1(\mathcal{G}_0, \emptyset; M))$             ▷ *Hypothesis Amendment (**no** memory)*
2: $F_1 := \log P_{\mathcal{G}_1^s}(D)$                              ▷ *Model Fitting*
3: $\mu_1 := \mathsf{LLM}_\mu(\mathcal{G}_1^s, \mathcal{G}_0^s, F_1, F_0)$              ▷ *Post-processing*
    **Iteration $\geq 2$**
1: **while** ¬ *Early stopping criterion* **do**          ▷ *Early stopping criterion*
2:     $\mathcal{G}_t^s := e(\mathsf{HA}_t(\mathcal{G}_{t-1}, \mu_{t-1}; M))$     ▷ *Hypothesis Amendment (**with** memory)*
3:     $F_t := \log P_{\mathcal{G}_t^s}(D)$                          ▷ *Model Fitting*
4:     $\mu_t := \mathsf{LLM}_\mu(\mathcal{G}_t^s, \mathcal{G}_{t-1}^s, F_t, F_{t-1})$        ▷ *Post-processing*
5: **end while**

---

## 4   EXPERIMENTS

Our overarching hypothesis is that the CMA is effective at the task of causal discovery. The experimental logic is as follows: First, we assess the constituent elements of the CMA to ensure they are appropriate for reasoning over causal graphs. The **synthetic neuropathic protein experiment** assesses the data-driven module of the CMA (described in more detail in Appendix A.1.4). We hypothesise that correctly defined causal graphs lead to higher data likelihoods under the model (up to Markov equivalence; see Appendix A.2.3) than misspecified graphs using a simple DAG setting. Whilst we focus on the data-driven module of the CMA, we present additional results on LLM behavioural patterns in Appendix A.2.4.

Second, we wish to assess the CMA relative to other data- or metadata-driven approaches for causal discovery. We assess the performance of various methods on three **causal discovery benchmarks** where a ground-truth graph is known *a priori*: 1) The Arctic Sea Ice benchmark; 2) the Sangiovese benchmark; 3) an Alzheimer's Disease (AD) benchmark. We develop the AD benchmark in collaboration with 5 domain experts, with expertise in either clinical neurology (with a specialist interest in AD) or neuroradiology, with a specialist interest in neurodegenerative diseases. Additional details can be found in Appendix A.5.1.

Finally, to assess the CMA in a real-world setting with multi-modal data, we apply it to data from the **Alzheimer's Disease Neuroimaging Initiative (ADNI)**, where the task is to jointly model the clinical and radiological phenotype of the disease. In contrast to the AD benchmark, the ADNI dataset contains noisy real-world data representing the complex aetiological process of the disease, for which the ground-truth ultimately remains contentious (Herrup, 2015; Gulisano et al., 2018). Additionally, we demonstrate the CMA's ability to perform a causal discovery task in a multi-modal data setting (tabular data and images), which is not possible for any of the other baseline methods. We assess the implications of the causal graph proposed by the CMA and extract various insights relating to biomarkers of the disease.

### 4.1   SYNTHETIC NEUROPATHIC PROTEIN EXPERIMENT

**Experimental setup**   To validate the modelling engine for the model fitting stage (Appendix A.1.4), we consider a simple DAG whose node description relates to the health sciences.

Phosphorylated-tau (P-tau) and $\beta$-amyloid are proteins which are involved in neurodegenerative processes in the human brain, and both are affected by the ageing process. We consider the DAG 'Age → AV45', 'Age → P-tau', and 'AV45 → P-tau', as per the classical amyloid cascade hypothesis (Hardy & Higgins, 1992). Here, AV45 refers to $\beta$-amyloid levels as measured by the Florbetapir F 18 tracer following Positron Emission Tomography (PET) scanning (Varghese et al., 2013; Márquez & Yassa, 2019). Based on the synthetic distributions, we generate 10,000 independent and identically distributed (i.i.d.) data points from the resulting graph using ancestral sampling and test the hypothesis that a correctly defined DAG can appropriately induce the observational distribution. We train each model 100 times on the data points and calculate an average data likelihood. Additional implementation details as well as results relating to Markov equivalence in graphs and the CMA's behavioural patterns can be found in Appendix A.2.

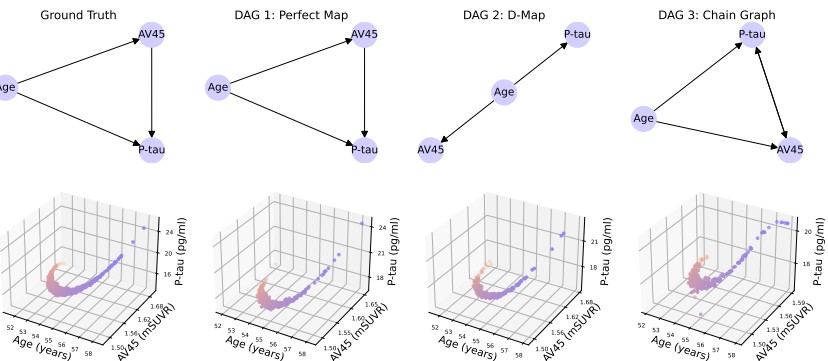

Figure 2: Observational samples for different modelling strategies in the synthetic neuropathic protein experiment. DAG 1 represents the correct data-generating process (a perfect map), DAG 2 adds an independence statement not present in the ground-truth distribution, and DAG 3 considers a chain component between the neuropathic proteins, which represents an associative link (represented as a bidirectional edge). Row 1 illustrates the causal graphs, and Row 2 illustrates observational samples from the trained models (or ground-truth function) representing those graphs.

**Results** Figure 2 illustrates the ground-truth DAG alongside three modelling strategies. We assess the three strategies by comparing the data likelihoods under each model. The first DAG (DAG 1) is a perfect map of the data-generating process, which is defined as an equivalence of independence statements in the graph and the ground-truth distribution. In other words, the graph matches the data-generating process. This model has an average data likelihood of 4402.68, with a standard deviation of $\pm 26.70$, and this acts as the reference ground-truth likelihood. The second DAG (DAG 2) removes a causal link which exists in the data-generating process, thereby adding a graphical independence statement not present in the ground-truth distribution. The second model is, therefore, a deficient representation of the ground-truth distribution, and has an average data likelihood of $3857.53 \pm 18.98$. The third DAG (DAG 3) adds a chain component between AV45 and P-tau, which represents an associative but not necessarily causal link. This model has an average data likelihood of $3757.55 \pm 60.80$. The Tukey HSD test (Abdi & Williams, 2010) is used for pairwise model comparisons whilst ensuring the overall error rate for the family of comparisons is kept to the 0.05 threshold by control of the Familywise Error Rate (FWER). There is a statistically significant difference in likelihood between all three models, with the model representing the data-generating process having the highest data likelihood. Detailed results of the comparisons are shown in Table 1. As can be seen, the results validate our initial hypothesis: The model that aligns most closely with the true data-generating process produces the highest data likelihood. As expected, we find that this is only valid up to the Markov equivalence class of the ground-truth DAG (see Appendix A.2.3).

## 4.2 BENCHMARKING EXPERIMENTS

**Experimental setup** We consider the task of recovering causal graphs on a number of causal discovery datasets. The Arctic sea ice dataset (Huang et al., 2021b) is from the field of atmospheric science and is an increasingly popular dataset for the task of full causal graph discovery (Kıcıman et al., 2023). This dataset considers the relations of several geophysical variables to sea ice thick-

Table 1: Pairwise comparisons between three different graphical hypotheses for the neuropathic protein experiment, using the Tukey HSD test. DAG 1 represents the data-generating process. DAGs 2 and 3 represent a missing edge and an associative edge where a causal one should exist, respectively.

| Model A | Model B | Lower Bound | Upper Bound | Mean Difference | P-Value | Reject Null Hypothesis |
|---|---|---|---|---|---|---|
| DAG 1 | DAG 2 | -558.499 | -531.799 | -545.149 | <0.001 | True |
| DAG 1 | DAG 3 | -658.479 | -631.780 | -645.130 | <0.001 | True |
| DAG 2 | DAG 3 | -113.330 | -86.630 | -99.980 | <0.001 | True |

ness (12 nodes; 48 true edges). A full description can be found in Appendix A.3.1. The Sangiovese dataset is from the field of agricultural science and is a conditional linear Gaussian Bayesian Network from the popular `bnlearn` R package (Magrini et al., 2017). The DAG considers several variables that relate to grape quality in Sangiovese vineyards in Tuscany (15 nodes; 55 true edges). Additional information can be found in Appendix A.4.1. The Alzheimer's dataset is another conditional linear Gaussian Bayesian Network that we developed in collaboration with 5 domain experts. The synthetically generated dataset considers demographic, clinical, imaging-based biomarker, and cognitive assessment variables, and how they relate to the aetiology of AD (11 nodes; 19 true edges). We construct the ground-truth graph based on a consensus heuristic, for which full details can be found in Appendix A.5.1. We compare the CMA with commonly used, state-of-the-art, data-driven algorithms including NOTEARS (Zheng et al., 2018), DAG-GNN (Yu et al., 2019), and TCDF (for the Arctic sea ice dataset; more details in A.3.2) (Nauta et al., 2019). We also compare the CMA with metadata-based benchmarks as per Kıcıman et al. (2023). The experimental setup for the metadata-based (LLM) benchmarks is given in Appendix A.3.2.

**Results** The results of the benchmarking experiments can be seen in Table 2. We report the normalised Hamming distance (NHD), which is defined for a predicted graph $\mathcal{G}'$ and ground-truth $\mathcal{G}$ as $1/m^2 \sum_{i,j=1}^{m} \mathbb{I}[\mathcal{G}_{i,j} \neq \mathcal{G}'_{i,j}]$: The edges in one graph and not the other, normalised by the total number of possible edges. It should be noted that the NHD depends on the number of edges reported by a causal discovery algorithm. Consequently, we use the approach described in Kıcıman et al. (2023), whereby we report the ratio of the NHD with a Baseline Hamming distance (BHD). The BHD is defined as a graph that contains the same number of edges as $\mathcal{G}$, but all of them are incorrect. A lower NHD/BHD ratio is the multiple by which the causal discovery algorithm outperforms the 'floor' baseline graph.

Table 2: A comparison of CMA performance against a number of data-driven or metadata-driven approaches to causal discovery on our three benchmark datasets. NHD: Normalised Hamming Distance, BHD: Baseline Hamming Distance, No. Edges: Total number of predicted edges, Ratio: NHD/BHD. A lower ratio (↓) is better. Extended results can be found in Appendix A.3.3, A.4.3, and A.5.3, respectively.

| Algorithm | Arctic sea ice | | | | Alzheimer's disease | | | | Sangiovese | | | |
|---|---|---|---|---|---|---|---|---|---|---|---|---|
| | NHD | No. Edges | BHD | Ratio (↓) | NHD | No. Edges | BHD | Ratio (↓) | NHD | No. Edges | BHD | Ratio (↓) |
| TCDF | 0.33 | 9 | 0.37 | 0.89 | - | - | - | - | - | - | - | - |
| NOTEARS (Static) | 0.30 | 8 | 0.38 | 0.82 | 0.21 | 7 | 0.26 | 0.74 | 0.26 | 1 | 0.27 | 0.96 |
| NOTEARS (Temporal) | 0.31 | 8 | 0.38 | 0.82 | - | - | - | - | - | - | - | - |
| DAG-GNN (Static) | 0.32 | 14 | 0.38 | 0.85 | 0.37 | 28 | 0.44 | 0.83 | 0.27 | 27 | 0.34 | 0.79 |
| DAG-GNN (Temporal) | 0.32 | 16 | 0.43 | 0.74 | - | - | - | - | - | - | - | - |
| gpt-3.5-turbo | 0.42 | 57 | 0.53 | 0.80 | 0.20 | 18 | 0.35 | 0.57 | 0.41 | 89 | 0.56 | 0.73 |
| gpt-4 | 0.35 | 58 | 0.51 | 0.68 | 0.11 | 23 | 0.41 | 0.27 | 0.32 | 67 | 0.49 | 0.65 |
| CMA | 0.25 | 36 | 0.54 | **0.46** | 0.07 | 16 | 0.35 | **0.21** | 0.23 | 24 | 0.36 | **0.63** |

The CMA outperforms causal discovery techniques which rely exclusively on data- or metadata (LLM)-driven approaches. In multiple instances, the CMA is capable of proposing additional causal relations within 'internal' variables, which we define as variables for which we have direct access to data. We give an illustrative example from the Arctic sea ice dataset. The CMA proposes that there

is a 70% probability that 'net longwave flux at the surface' (LW) should have a causal relationship to 'sensible plus latent heat flux' (HFLX). The CMA's reasoning trace is shown in Appendix Figure 12. Observational samples as well as counterfactual inference under the causal model which encodes this edge suggest a positive relationship between the variables (Appendix A.3.4). Whilst this relationship is not present in the original causal graph, nor are its dynamics present in the domain literature used to construct it (Huang et al., 2021b), increased absorption of longwave radiation (LW) produces tropospheric warming, leading to an increase in sea surface temperature (SST), until SST-dependent cooling increases to establish equilibrium, which occurs through LW's effect on latent and sensible heat fluxes (HFLX) (Bates et al., 2012). We therefore contend this is an example of a missing edge from the ground-truth graph detected by the CMA. Extended results for the Arctic sea ice, Sangiovese, and AD benchmarks can be found in Appendix sections A.3.3, A.4.3, and A.5.3, respectively.

## 4.3 CASE STUDY: ALZHEIMER'S DISEASE NEUROIMAGING EXPERIMENT

**Experimental setup** In this experiment, a CMA is applied to a challenging real-world dataset from the Alzheimer's Disease Neuroimaging Initiative (ADNI) (Petersen et al., 2010). We consider the same variables as in the AD benchmark experiment above (Section 4.2), with the exception that we link the clinical and radiological phenotypes of AD by considering how the variables might impact a brain Magnetic Resonance Image (MRI) for a given participant. This allows us to assess the effects of interventions and counterfactual queries in the imaging space. For example, if we intervene on brain volume by reducing it, we would expect a reduction in brain size on the MRI image in a reasonable model. We assess the counterfactual implications of the model proposed by the CMA and illustrate the framework's utility for identifying potentially useful external variables for inclusion in the causal graph. Additional experimental setup details can be found in Appendix A.6.

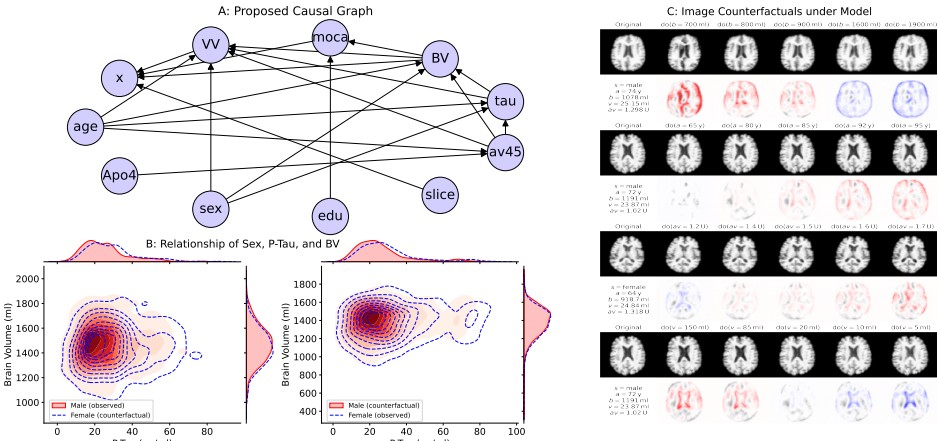

Figure 3: Outputs of the CMA on the Alzheimer's Disease NeuroImaging (ADNI) dataset. The CMA proposes a high-quality, biologically plausible causal graph (A), which contains insightful causal relationships which are not currently considered conventional domain knowledge (B). The model is multi-modal and produces plausible counterfactuals of brain MRIs. Best viewed zoomed in.

**Results** The CMA outputs a trained SCM that encodes functional causal relationships between the disease covariates and the presentation of the disease in MRI images. Figure 3a illustrates the CMA's proposed causal graph, whilst 3c illustrates the biologically plausible output of counterfactual queries in the imaging space for qualitative evaluation. For example, increasing age produces visible cortical degeneration in the counterfactual image (Figure 3c; row 2). It should be noted that increasing the age value not only demonstrates cortical neurodegeneration, but the ventricles are also expanded, which is the expected neuro-radiological result of increasing age (Dinsdale et al., 2021). Directly intervening on whole brain and ventricular volumes (Figure 3c; rows 1 and 4, respectively) leads to expected morphological effects in both instances.

As with the Arctic sea ice dataset, we note interesting edges proposed for internal variables in the causal graph. In particular, we note the relationship 'biological sex→tau pathology'. This relationship was not proposed by any of the domain experts for the AD benchmarking task in Section 4.2. The graphs proposed by the experts can be seen in Appendix A.5. Nonetheless, counterfactual inference using the model trained by the CMA (under the ADNI dataset) provides evidence of increased P-tau protein levels in females (Figure 3b). Counterfactually intervening on sex led to a statistically significant shift in P-tau levels following Welch's $t$-test of unequal variances. This was observed when intervening on both males and females ($t = -6.84, p < 0.001$, and $t = 6.05, p < 0.001$, respectively). Despite not reflecting current expert consensus, there is emerging evidence that such a mechanism exists. In recent work by Yan et al. (2022), it was shown in both in-vitro and in-vivo models that the X chromosome-linked protein ubiquitin-specific peptidase 11 (USP11) alters tau aggregation patterns by deubiquitination initiated at lysine-218. Deubiquitination (removal of ubiquitin, a 76 amino acid protein), enhances tau buildup, which produces a damaging effect on brain tissue. USP11 'escapes' complete X-inactivation (Yan et al., 2022), meaning females exhibit greater levels than males, and delimits the causal relationship between having an XX genotype and increased tau pathology.

As the CMA leverages LLMs to produce hypotheses, it is capable of proposing potentially confounding or modulating variables for which we do not have direct access to data ('external' variables). For example, whilst assessing a potential relationship between the APOE4 gene and tau pathology (APOE4→tau pathology), the CMA proposes that the TREM2 gene may be a potential confounder. The reasoning trace is shown in full in Appendix A.7. We accrue data for the soluble form of TREM2 (sTREM2) from the ADNI dataset and include it in the causal graph with a directed edge to P-tau (sTREM2→tau pathology). By counterfactual inference, we find a statistically significant positive relationship between sTREM2 and P-tau ($t = 6.55, P < 0.001$). Whilst several associational studies exist with inconclusive results (Suárez-Calvet et al., 2019; Zhao et al., 2022), we believe this analysis to be the first counterfactual-inference-based approach to analysing this relationship. TREM2 is an immune receptor expressed by support cells of the central nervous system (Bouchon et al., 2001), and is broken down by ADAM metalloprotease (Kleinberger et al., 2014) to produce its soluble form. Whether this breakdown (and therefore reduction of TREM2 on the cell surface membrane) leads to increased tau pathology, or whether sTREM2 itself exerts a specific biological function is unclear (Filipello et al., 2022). Whilst our work proposes a potential relation from sTREM2 to tau deposition, the underlying mechanism is currently unknown (Filipello et al., 2022).

## 5 CONCLUSION

We introduce the CMA, a modular framework for causal discovery with large language models and deep SCMs (DSCMs), enabling causal discovery for multi-modal data. Through a diverse set of experiments on synthetic and real-world datasets from a range of scientific fields (agriculture, geophysics, and health), we demonstrate that the CMA outperforms existing data- or metadata-driven approaches for causal discovery. The CMA is capable of proposing insightful relationships between variables internal and external to the DSCMs, and we provide a number of illustrative examples in the geophysics and health domains.

Future work should address several important limitations. Whilst discrete variables with parents currently require continuous relaxation, DSCMs could support discrete mechanisms with a Gumbel–max parametrisation (Pawlowski et al. (2020), Appendix C). In our AD case study, our model used 2D axial slices of the brain, which may not be optimal for producing counterfactuals based on 3D volumes; hence, incorporating 3D volume modelling in DSCMs is a potential extension. Although the observational distribution is determined by a graph's Markov Equivalence Class (MEC), there is evidence that LLMs can effectively reduce the MEC by ruling out implausible graphs (Long et al., 2023a). We discuss these ideas in more detail in Appendix A.2.4. Finally, The CMA relies on DSCMs to fit data, which assumes a Markovian DAG. Whilst we extend the framework to include chain graph elements, in practice, we found that LLMs struggle to represent chains in a fully automated manner- instead, the CMA proposes potentially bidirectional relationships and human intervention is necessary to allow appropriate Deep Chain Graph modelling. Investigating techniques to enable fully automated chain graph modelling, and indeed more generally extending deep SCMs to represent more flexible, non-Markovian causal graphs (for example, models which allow for feedback loops) represents a natural avenue of future research.

REPRODUCIBILITY STATEMENT

To increase reproducibility, we have included all implementation details in Appendix A.1. We also include implementation and prompting code at `https://anonymous.4open.science/r/causal_modelling_agent-F443/`.

ACKNOWLEDGMENTS

This work is supported by an EPSRC Industrial Case grant [EP/W522077/1] and the EPSRC-funded UCL Centre for Doctoral Training in Intelligent, Integrated Imaging in Healthcare (i4health) [EP/S021930/1]. AA is supported by a Microsoft Research PhD Scholarship. Wellcome Trust award 221915/Z/20/Z, JPND and MRC award MR/T046422/1 and the NIHR ULCH Biomedical Research Centre support DCA's work on this topic.

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

## A  APPENDIX

**Summary of Appendices.**

- A.1: Causal Modelling Agent
- A.2: Synthetic neuropathic protein experiment
- A.3: Benchmark: Arctic Sea Ice
- A.4: Benchmark: Sangiovese
- A.5: Benchmark: Alzheimer's Disease
- A.6: Case Study: Alzheimer's Disease Neuroimaging Experiment

### A.1  CAUSAL MODELLING AGENT

In this section, we provide additional background on DSCMs and chain graph models, before introducing our novel parameterisation of a chain graph with deep learning elements. We then provide implementation details for our global and local hypothesis amendment phases, the modelling engine, and the post-processing phase of the CMA.

#### A.1.1  ADDITIONAL BACKGROUND

**Deep Structural Causal Models**  A structural causal model (SCM) $\mathfrak{C} := (\mathbf{S}, P_{\boldsymbol{\epsilon}})$ consists of a collection $\mathbf{S} = (f_1, \ldots, f_D)$ of mechanisms $X_j := f_j(\mathsf{PA}_j, \epsilon_j)$, $j = 1, \ldots, D$, where $\mathsf{PA}_j \subseteq \{X_1, \ldots, X_D\} \setminus \{X_j\}$ are assumed to be direct causes (parents) of $X_j$, and a jointly independent distribution over exogenous noise ($\epsilon$) variables $P_{\boldsymbol{\epsilon}} = \prod_{d=1}^{D} P(\epsilon_d)$, where $D$ is the number of variables for which there is associated exogenous noise. An SCM $\mathfrak{C}$ entails a unique distribution over $\{X_d\}_{d=1}^{D}$, denoted by $P_{\mathbf{X}}^{\mathfrak{C}}$. SCM $\mathfrak{C}$ satisfies the Markov condition (Geiger & Pearl, 1990), whereby every node is independent of its non-descendents given its parents. Each conditional probability

can be seen as being defined by its corresponding mechanism and noise $P(X_d|\mathsf{PA}_d) = f_d(\epsilon_d; \mathsf{PA}_d)$ (Pawlowski et al., 2020; Peters et al., 2017). We perform interventions by altering the mechanism for variable $X_k$ such that $X_k := \tilde{f}(\widetilde{\mathsf{PA}}_k, \tilde{\epsilon}_k)$. An intervention induces a new interventional distribution $P_{\mathbf{X}}^{\mathfrak{C}'} =: P_{\mathbf{X}}^{\mathfrak{C}; do(X_k := \tilde{f}(\widetilde{\mathsf{PA}}_k, \tilde{\epsilon}_k))}$ (Peters et al., 2017). Interventions which place a point mass on real value $m$ are called atomic interventions and are written as $P_{\mathbf{X}}^{\mathfrak{C}; do(X_k := m)}$. In the scenario where we already have some observations, we can define a SCM in which the distribution of noise variables has been modified as $\mathfrak{C}_{\mathbf{X}=\mathbf{x}} := (\mathbf{S}, P_\epsilon^{\mathfrak{C}|\mathbf{X}=\mathbf{x}})$, where $P_\epsilon^{\mathfrak{C}|\mathbf{X}=\mathbf{x}} := P_{\epsilon|\mathbf{X}=\mathbf{x}}$, and $\mathbf{X} = \mathbf{x}$ refers to a specific observation of random variable $\mathbf{X}$. Performing an intervention in this setting amounts to performing a counterfactual query (Pearl, 2011; Peters et al., 2017). Computing counterfactual queries can be performed in three steps known as abduction, action, and prediction (Pearl, 2011): 1) In abduction, we infer the exogenous noise which is compatible with the observations $\mathbf{x}$, i.e., compute $P_\epsilon^{\mathfrak{C}|\mathbf{X}=\mathbf{x}}$; 2) Next, we perform an intervention of interest, such as $do(x_{I,1} := x'_{I,1})$; 3) Finally, it is possible to infer a quantity of interest. If we are interested in quantity $\mathbf{z}$, we could compute $P_{\mathbf{z}}^{\mathfrak{C}|\mathbf{X}=\mathbf{x}; do(x_{I,1} := x'_{I,1})}$ (Peters et al., 2017; Pearl, 2011; 2018).

For DSCMs, mechanisms are parameterized by invertible DL elements so that the exogenous noise variables can be computed as per the abduction step. Conditional normalizing flows (Rezende & Mohamed, 2015; Trippe & Turner, 2018) are used to learn bijective mappings between the observed variables and their exogenous noise. The mappings operate in the data space, which means they are costly for modelling high-dimensional data (e.g., images). Pawlowski et al. (2020) propose instead to decompose such a mechanism $f_k$ into invertible $h_k$ and non-invertible $g_k$ functions. The noise is correspondingly decomposed as $e_k = (u_k, z_k)$, with $p(e_k) = p(u_k)p(z_k)$. The non-invertible noise term $z_k$ is computed by the recognition model of a conditional variational autoencoder (CVAE) (Sohn et al., 2015). This approach has been termed an 'amortized, explicit-likelihood mechanism'. Additional details can be found in Pawlowski et al. (2020).

**Deep Chain Graph Models** Chain graphs (CGs) are a form of probabilistic graphical model which contains both directed and undirected edges. A CG can be more expressive than either SCMs or Markov networks by introducing conditional independence statements not possible by either model type alone (Barber, 2012; Lauritzen, 1996). A CG can be interpreted as a Directed Acyclic Graph (DAG) over chain components (Frydenberg, 1990), where the chain components of a graph $\mathcal{G}$ can be identified by first removing all directed edges from $\mathcal{G}$ to produce graph $\mathcal{G}'$. The remaining connected components in $\mathcal{G}'$ (i.e., those vertices with undirected edges connecting them) are chain components $\tau$. Each component represents a distribution over the variables within it, conditioned on parental components (Barber, 2012):

$$P_{\mathbf{X}} = \prod_\tau p(\mathcal{X}_\tau \,|\, \mathsf{PA}_\tau), \quad p(\mathcal{X}_\tau \,|\, \mathsf{PA}_\tau) \propto \prod_{d \in \mathcal{D}_\tau} p(x_d \,|\, \mathsf{PA}_d) \prod_{c \in \mathcal{C}_\tau} \phi(\mathcal{X}_c), \tag{4}$$

where $\mathcal{D}_\tau$ is the set of variables in component $\tau$ with directed terms, and $\mathcal{C}_\tau$ denotes the union of the cliques in $\tau$. Unlike a fully specified DAG on singleton nodes, CGs can also be used to represent potential unmeasured confounding between two or more variables. We can take advantage of this increased flexibility by operationalising them using modular DL elements. Namely, we use (conditional) multivariate normalizing flows (Lu & Huang, 2019; Winkler et al., 2019) as the mechanisms $f_i \in \mathbf{S}$ on maximal cliques. We call this model a Deep Chain Graph Model (DCGM). Training a DCGM proceeds with the same likelihood-based objective as the DSCM. Whilst previous work has focussed on interpreting neural networks or elements within them as CGs (Shen & Cremers, 2020; Hwang et al., 2022), as far as we know, this is the first time that a CG is explicitly defined and then parameterized with DL modules such that it remains compatible with the counterfactual inference procedure described in A.1.1.

### A.1.2 GLOBAL HYPOTHESIS AMENDMENT

The global phase implementation requires two inputs, an input JSON and an optional memory. During the first two iterations of the CMA, the memory is empty, as can be seen in the main text. The output of the call is a JSON with the same format as the input JSON. To ensure an appropriately structured output, we leverage `gpt-4-0613`'s 'function calling' capability (OpenAI, 2023b;a), which represents the encoding function $e(\cdot)$ that operates on the representation of a given graph $G$

to produce a structured graph $G^s$, as in the main text. The parameters of the input JSON are the nodes of the input graph, which have as properties an array called 'parents'. The LLM is instructed to fill the 'parents' array of each node. Where a memory is included, it is embedded into the prompt directly. The algorithm and system prompt for the global phase are shown in Algorithm 2 and Listing 1, respectively.

---

**Algorithm 2** Global Hypothesis Amendment

---

**Require:** An input JSON $\mathcal{G}$ representing the initial graph and a Memory $\mu$.
  1: **procedure** GLOBAL PHASE($\mathcal{G}(V, E), \mu$)
  2:     **if** First iteration of CMA **then**
  3:         Initialize $\mu$ as empty
  4:     **end if**
  5:     **if** Second iteration of CMA **then**
  6:         Initialize $\mu$ as empty
  7:     **end if**
  8:     Extract nodes and their properties from $\mathcal{G}$
  9:     **if** $\mu$ exists **then**
 10:        Embed the memory $\mu$ into the prompt directly
 11:     **end if**
 12:     **for** each node $n$ in $\mathcal{G}$ **do**
 13:        Make API call to `gpt-4-0613` with $n$ to get updated edges
 14:        Update $\mathcal{G}$ with the response from the API call
 15:     **end for**
 16:     **return** modified JSON $\mathcal{G}'(V, E')$ with updated edges for each node.
 17: **end procedure**

---

Listing 1: System prompt and schema for the global phase.

```
"""
Schema and system prompt for the global hypothesis amendment phase
"""
# Schema for function calling output
schema {'name': 'change_DAG', 'description': 'The new improved DAG json',
    'parameters': {'type': 'object', 'properties': {'node1': {'type': '
    object', 'id': 'node1', 'parents': {'type': 'array', 'description': "
    List of parents of node1. Think about what variables of ['node2', '
    node3',...] can affect it  "}}, 'node2': {'type': 'object', 'id': '
    node2', 'parents': ...}.

# Base prompt
prompt = f"""Consider the following JSON object: {input_json_dict}. Are
    there any links you'd like to add? Any you think should be removed?
    Any whose direction should be reversed? Consider directed changes of
    the variables in the JSON ONLY. Do not propose changes on variables
    not in the given JSON. Output a new valid JSON that you think would
    be an improved representation of the causal relationships between the
     variables."""

====nodes list:
{nodes}
=======

Do not give an explanation. Only output a JSON. Background knowledge:
    This JSON describes a directed acyclic graph (DAG), which describes
    the causal relationships between nodes. Each node may be an ancestor
    (have an empty parent array), or may be the child of some other nodes
    (where you can see the parents in the JSON)."""

# if memory
prompt = f"{prompt}\n\n{memory}"
```

### A.1.3 Local Hypothesis Amendment

This section outlines the details of the local hypothesis amendment phase. As described in the main text, this phase considers pairwise comparisons between the variables. Let $V$ be the set of vertices in the structured causal graph $G^s$. For each pairwise comparison between the vertices in the graph, $(u, v), \forall\, u, v \in V, u \neq v$, the local phase outputs a set of probabilities over all possible actions. The action space varies based on two types of pairwise comparisons: 1) 'Relationship assessment', and 2) 'Relationship adjustment'. Relationship assessment assumes that there is currently no causal link between variables $u$ and $v$. We ask the LLM to output three probability values based on the assumption that $u$ and $v$ are not related causally. The probabilities reflect three possible decisions:

1. **'No direct causality'**: There is no direct causal relationship between the variables.
2. **'u causes v'**: Changing $u$ causes a change in $v$ in a causal manner.
3. **'v causes u'**: Changing $v$ causes a change in $u$ in a causal manner.

If the probability values for **'u causes v'** and **'v causes u'** are within a 5% threshold of each other, and are both individually greater than the probability that there is **'No direct causality'**, then the CMA saves these variables as potentially having a 'bidirectional' causal relationship. The CMA randomly chooses a direction in this case (as long as the condition that the output is a Markovian DAG is satisfied), and implements a causal edge. Following training, a user can assess whether the flagged 'bidirectional' relationships warrant additional modelling with a Deep Chain Graph model. An example output of a relationship assessment is shown below:

```
{"NO DIRECT CAUSALITY": 90, "U CAUSES V": 5, "V CAUSES U": 5}
```

Relationship adjustment, on the other hand, assumes that a causal relationship already exists in either direction between variables $u$ and $v$. The LLM outputs probability values which reflect three different possible decisions:

1. **'Keep'**: The causal relationship is correct in the current direction and should be kept.
2. **'Remove'**: Remove this relationship completely; there is no causal relationship between $u$ and $v$.
3. **'Reverse'**: Flip the direction of this relationship. There is evidence of a causal relationship in the opposite direction to the currently proposed direction.

An example output of a relationship adjustment is shown below:

```
{"KEEP": 5, "FLIP": 0, "REMOVE": 95}
```

Naively, one could make API calls to the LLM for each CMA iteration (after each global phase) to obtain the relevant probabilities for each edge (and non-edge). However, this could be less efficient and may incur significant costs. Instead, to obtain probabilities for use in decision-making from the LLM, an initial (two-stage) 'pre-computation' phase occurs before the first CMA iteration. We gather all relevant probabilities as a preliminary step and apply them as appropriate for each subsequent iteration of the CMA. First, for each pairwise comparison $(v, u)$, two API calls are made to `gpt-4-0613` (OpenAI, 2023b). Specifically, one call is made which assumes that the variables are independent (relationship assessment), and one call is made assuming a relationship already exists (relationship adjustment); i.e., $v \rightarrow u$. A run is defined as a complete iteration of all pairwise comparisons in the graph (two API calls are made for each comparison). Multiple runs are made to mitigate potential variability or inconsistency in the outputs. We use `gpt-4-0613` with a temperature setting of 0.6 for all experiments.

In addition, the local phase can optionally utilise a Retrieval Augmentation Generation (RAG) pattern, whereby each edge in the DAG is described using natural language (for example, for the edge 'precipitation $\rightarrow$ humidity', we can write 'precipitation has a causative effect on humidity'). The edge descriptions are embedded into a vector space using the `text-embedding-ada-002` model, where all embeddings are normalized to unitary length. The embeddings are used to perform semantic text search with a vector store to retrieve the most relevant information which relates to the current edge. Retrieval occurs by calculating the cosine similarity between the edge description and

the document embeddings in the vector store as $\cos(\omega) = \langle \mathbf{x}, \mathbf{y} \rangle / \|\mathbf{x}\|\|\mathbf{y}\|$, which simplifies to a dot-product due to the normalisation. The retrieved documents can be passed into the '*system prompt*' as additional context. Aside from the (optional) additional context, the system prompt (Listing 2) contains instructions to consider the monotonicity, temporality, subjectivity, and potential spillover effects for each relationship, as well as potential confounders or mediating factors when outputting probability values.

The output probabilities are stored in JSON format. An optional explanation which represents the reasoning trace of the model can also be saved. The output is a JSON file for every pair of variables, with the name:

$pair[0]pair[1]\_temp\_prompt\_loop\_(if\_exists).json$. In the second stage of the pre-computation phase, the probabilities stored in the JSONs for every combination are grouped, averaged over, and saved into two tables, one for 'empty' relationships (when there is no prior causal link assumed) and another for 'existing' relationships (assuming a prior causal relationship in either direction). During the CMA iterations, for any given pairwise comparison, if an edge already exists (e.g. from a previous iteration), the CMA looks into the 'existing' relationships table, otherwise into the 'empty' table. A final decision is made by sampling from a uniform distribution and then mapping the value to the probabilities output by the LLM. Algorithms 3 and 4 give an exposition of the pre-computation and local amendment operations, respectively.

Listing 2: System prompt for the local phase. Includes output options for both existing and empty relationships.

```
1  """
2  System prompt for the local hypothesis amendment phase
3  """
4
5  "You are an expert {INSERT DOMAIN} with a sub-specialist interest in {
        INSERT SUB-FIELD}. Your task is to explore the hypothesis space of
        the relationships that might exist between a number of variables
        relating to {INSERT SPECIFIC TOPIC}."
6
7  # The options when assuming a pre-existing causal relationship:
8  Please decide the probabilistic nature of the relationship using the
        following options:
9  KEEP: "The causal relationship is correct in the current direction and
        should be kept."
10 FLIP: "FLIP the direction of this relationship. There is evidence to
        suggest that it is plausible that changing {pair[1]} causes a change
        to {pair[0]}."
11 REMOVE: "Remove this relationship completely; that is, no causal
        relationship exists between {pair[0]} and {pair[1]}."
12
13
14 # The options when assuming no prior causal relationship:
15 Please decide the probabilistic nature of the relationship using the
        following options:
16 NO DIRECT CAUSALITY: "There is no direct causal relationship between the
        {pair[0]} and {pair[1]}."
17 changing {pair[0]} causes changes to {pair[1]}: "Changing {pair[0]}
        causes a change in {pair[1]} in a causal manner."
18 changing {pair[1]} causes changes to {pair[0]}: "Changing {pair[1]}
        causes a change in {pair[0]} in a causal manner."
19
20
21 ===============================
22 To make your decision, consider the following criteria:
23 # The first criterion applies if an edge already exists.
24 Reversal and Causality: "Could the direction of this relationship be
        inverted, such that the {pair[1]} might cause changes to {pair[0]}
        instead? If yes, the probability of option [FLIP] should be high."
25 Confounding: "Could this relationship be confounded by any other
        variables?"
```

```
26 Mediation_or_Modulation: "Are there any additional variables that might
       either mediate or moderate this relationship?"
27 Monotonicity: "Do you think the relationship is monotonic?"
28 Temporality: "Is there a time-lag between causation?"
29 Subjectivity: "Could this relationship be subjective?"
30 Spillover: "Could this relationship be spillover?"
31 ================================
32
33
34 You may want to consider the following contextual information about {pair
       [0]} and {pair[1]}:\n\n
35 {CONTEXT}
36 "
```

---

**Algorithm 3** Local Hypothesis Precomputation Phase

---

1: **for** each node $v \in V$ **do**
2:     **for** each node $u \in V$ where $u \neq v$ **do**
3:         Formulate the API requests with pair $(v, u)$
4:         Make API call to `gpt-4-0613` assuming no causal link between $v$ and $u$
5:         Store output probabilities in JSON file in ASSESSED-FOLDER
6:         Make API call to `gpt-4-0613` for potential causal relationship in direction $v \rightarrow u$
7:         Store output probabilities in JSON file in ADJUSTED-FOLDER
8:     **end for**
9: **end for**
10: Process JSON files in ASSESSED-FOLDER to group and average probabilities
11: Store averaged probabilities in ASSESSED-TABLE CSV
12: Process JSON files in ADJUSTED-FOLDER to group and average probabilities
13: Store averaged probabilities in ADJUSTED-TABLE CSV

---

**Algorithm 4** Local Hypothesis Amendment

---

**Require:** An input JSON $\mathcal{G}$ representing the output graph from the global phase.
1: **procedure** LOCAL PHASE($\mathcal{G}(V, E)$)
2:     **for** each node $v \in V$ **do**
3:         **for** each node $u \in V$ where $u \neq v$ **do**
4:             **if** directed edge exists between $v$ and $u$ **then**
5:                 Retrieve probabilities from ADJUSTED-TABLE CSV for relationship $v \rightarrow u$
6:             **else**
7:                 Retrieve probabilities from ASSESSED-TABLE CSV for the pair $(v, u)$
8:             **end if**
9:             Decide relationships based on retrieved probabilities and threshold
10:             **if** adding or flipping a directed edge creates a cycle **then**
11:                 Skip this amendment
12:             **end if**
13:         **end for**
14:     **end for**
15:     **return** modified JSON $\mathcal{G}'(V, E')$ with amended hypothesis.
16: **end procedure**

---

### A.1.4 MODELLING ENGINE

We use the term Modelling Engine ($ME$) to refer to a software abstraction of both DSCMs and DCGMs, whereby either model type can be specified using a single JavaScript Object Notation (JSON) file which includes the name of each variable, its parent nodes, its type (binary/discrete/continuous/image), and whether it exists in a chain component $\tau$. Because SCMs are specified by DAGs, and CGs can be interpreted as DAGs over chain components (Barber, 2012), it is important to account for the order in which we define the modules, as they must reflect the child→parent pathways

in the DAG. For example, attempting to construct a conditional network for a child node without first having constructed the network(s) for its parent(s) can lead to errors/model misspecification.

We implement the function $\mathcal{K}$ from the main text using Kahn's topological sorting algorithm (Kahn, 1962). In this case, the purpose of $\mathcal{K}$ is to sort the nodes and order network construction. As previously discussed, a structured graph $\mathcal{G}_t^s$ is taken as input to $\mathcal{K}$ to produce a computational graph $CG_t$. Let $V$ be the set of vertices in the structured causal graph, and $E$ be the set of edges. The in-degree of vertex $a \in V$ is defined as the number of incoming edges; in-degree$(a) = |\{b \in V : (b, a) \in E\}|$. We identify a list $\mathcal{N}$ of nodes with in-degree$(n) = 0$. These nodes are removed from $\mathcal{N}$ alongside any out-going edges they have and added to list $\mathcal{L}$. This process iterates until there are no nodes in $\mathcal{N}$, leaving a sorted list in $\mathcal{L}$. The computational graph is then defined by iteratively constructing the appropriate neural networks, hypernetworks, normalizing flows, and CVAE architectures where appropriate for each variable in the order given by $\mathcal{L}$.

### A.1.5 POST-PROCESSING

The post-processing phase consists of three steps:

1. In the first step, we compare the graph JSONs from the current and previous iterations. This allows us to identify differences in relationships, which we then convert into textual descriptions.

2. In the second step, we compare the average data likelihoods from the DSCM models of both the current and previous iterations. This comparison allows us to gauge the impact of changes on data likelihood.

3. The final step involves using the identified differences in relationships and likelihood fit to make an API call to `gpt-4-0613` using the prompt in Listing 3. We instruct the LLM to generate an explanation about the potential causes of the observed differences in fit. This explanation constitutes a 'memory', and is saved as a text file. The memory file can be used to guide the global phase of the subsequent iteration.

The post-processing phase is summarised in Algorithm 5.

Listing 3: System prompt for the memory generation.

```
1  "You are an expert {INSERT DOMAIN} with a sub-specialist interest in {
       INSERT SUB-FIELD}. Your task is to explore the hypothesis space of
       the relationships that might exist between a number of variables
       relating to {INSERT SPECIFIC TOPIC}."
2
3  I am going to present you two JSONs in <XML> tags that represent directed
       acyclic graphs which relate to {INSERT DOMAIN}.
4  For each node, you can see its parents in its relevant 'parents' array.
       Note that if two nodes have a non-zero chain number, they are in a '
       chain', which means they have an associative but NOT a causal
       relationship.
5
6  <PREVIOUS GRAPH> { (previous_graph_topo_sorted)} </PREVIOUS GRAPH>
7
8  <CURRENT GRAPH> { (current_graph_topo_sorted)} </CURRENT GRAPH>
9
10 In an experiment where we fit these graphs onto real data,
11 {string_comparison}
12 Based on your expert knowledge, try to provide an explanation for why
       this might be the case by considering all the individual differences
       between both graphs. Here is a summary of the differences you should
       focus on:
13 <GRAPH DIFFERENCES>{string_differences}</GRAPH DIFFERENCES>
14 Contextualize your analysis by thinking about the current literature on {
       adverb}. Provide specific advice for improving the current graph
       using these variables, given the literature. IF there are chain
       differences, do you recommend making the relationships between the
       variables causal or associative? Explain your reasoning.
```

---

**Algorithm 5** Post-processing and memory generation

---

**Require:** 2 input JSONs $\mathcal{G}_t$, $\mathcal{G}_{t-1}$ and 2 numbers $F_t$, $F_{t-1}$ representing the graphs and likelihood fits produced in current and previous iterations.
1: **procedure** POST-PROCESSING($\mathcal{G}_t$, $\mathcal{G}_{t-1}$, $F_t$, $F_{t-1}$)
2:      Find differences $\mathcal{D}_g$ between graphs $\mathcal{G}_t$, $\mathcal{G}_{t-1}$
3:      Convert $\mathcal{D}_g$ into text
4:      Calculate difference $\mathcal{D}_f$ between $F_t$, $F_{t-1}$
5:      Embed $\mathcal{D}_f$ and $\mathcal{D}_g$ into an API call to `gpt-4-0613` to get a memory $\mu_t$
6:      **Save memory** $\mu_t$ in a text file
7: **end procedure**

---

## A.2 SYNTHETIC NEUROPATHIC PROTEIN EXPERIMENT

### A.2.1 DATA GENERATION

As described in the manuscript, we consider the following DAG structure: 'Age → AV45', 'Age → P-tau', and 'AV45 → P-tau'. The data-generating process is defined as follows:

$$
\begin{aligned}
\text{age} &:= f^*_{\text{age}}(\epsilon^*_{\text{age}}) = 55 + \epsilon^*_{\text{age}}, & \epsilon^*_{\text{age}} &\sim \mathcal{N}(0,1), \\
\text{av45} &:= f^*_{\text{av45}}(\epsilon^*_{\text{av45}}; \text{age}) = (\text{age}^2 + \epsilon^*_{\text{av45}} + 90)/60, & \epsilon^*_{\text{av45}} &\sim \mathcal{N}(0,0.3), \\
\text{tau} &:= f^*_{\text{tau}}(\epsilon^*_{\text{tau}}; \text{av45}, \text{age}) = (\text{age}^3 + \text{av45} + \epsilon^*_{\text{tau}} + 110)/6, & \epsilon^*_{\text{tau}} &\sim \mathcal{N}(0,0.2),
\end{aligned}
\tag{5}
$$

### A.2.2 EXPERIMENTAL SETUP

For mechanisms $\mathbf{S} = (f_1, \ldots, f_d)$, all continuous singleton nodes are represented by (conditional) rational spline normalizing flows (Reinhold et al., 2021; Pawlowski et al., 2020), whereas continuous nodes in chains are represented by (conditional) multivariate normalizing flows (Lu & Huang, 2019; Winkler et al., 2019). The flows are composed of components that fit the distribution as well as constrain the support of the output distribution. We consider two variables $K$ and $V$, where $K$ is an ancestor with no parents, and $V$ is a child of $K$. Their mechanisms are defined as:

$$
K := f_K(\epsilon_K) = (\exp \circ \text{AffineNormalisation} \circ \text{Spline}_\theta)(\epsilon_K),
\tag{6}
$$

$$
V := f_V(\epsilon_V; K) = (\exp \circ \text{AffineNormalisation} \circ \text{ConditionalTransform}_\theta(\hat{K}))(\epsilon_V),
\tag{7}
$$

where the subscript $\theta$ denotes elements with learnable parameters. The $\text{Spline}_\theta$ transformation is a first-order neural spline flow (Durkan et al., 2019), and $\text{ConditionalTransform}_\theta(\cdot)$ is a conditional transformation which might represent a conditional affine transformation in the case of univariate nodes, or a combination of spline coupling and conditional spline transformations $[\text{SplineCoupling}_\theta, \text{ConditionalSpline}_\theta]$ in the case of chain components (Lu & Huang, 2019; Winkler et al., 2019). We require conditional flows for singleton nodes or chains with parents. Conditional flows use hyper-networks, which are multi-layer perceptrons (MLPs) that predict the transformation parameters of otherwise invertible functions. The MLPs are composed of two hidden layers, with 10 and 16 nodes in the first and second layers, respectively. All variables are singly bounded and therefore an exponential transform is applied to constrain the unbounded values following a fixed AffineNormalisation transformation. The AffineNormalisation has the location and scale parameters set to the logarithm of the mean and variance of the training data, respectively, and is equivalent to a whitening operation in unbounded log-space (Pawlowski et al., 2020). Note the hat $\hat{\cdot}$ in equation (8), which refers to an unconstrained value (i.e., we invert the constraint transforms of exponentiation and AffineNormalisation). For all exogenous noise variables $\epsilon \sim P_\epsilon$, we use unit Gaussians as base distributions. All learnable flow parameters were optimized by maximizing the likelihood using the AdamW optimizer (You et al., 2019) with a learning rate of $3 \times 10^{-3}$ for 300 epochs.

### A.2.3 ADDITIONAL RESULTS I - MARKOV EQUIVALENCE

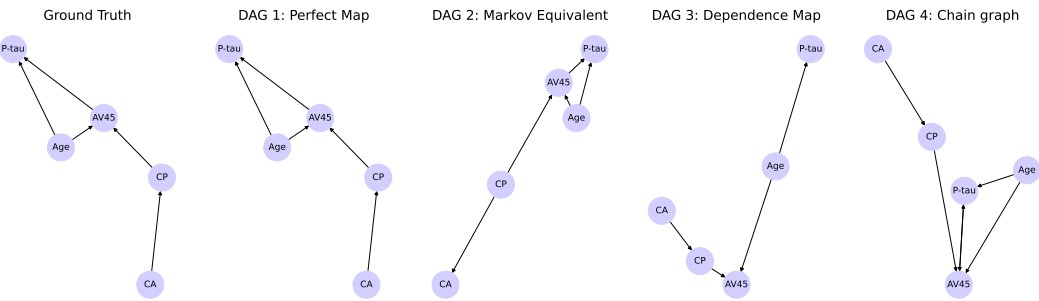

Figure 4: Figure illustrating the Directed Ayclic Graphs (DAGs) used in a synthetic neuropathic protein experiment. DAG 2 is Markov equivalence to the data generating DAG (DAG 1).

In this section, we explicitly define the notion of equivalence classes for SCMs and present further numerical results as they relate to such a class.

**Markov equivalence in graphs**    Given a DAG $\mathcal{G}$ and a distribution $P$, the distribution is said to be Markovian if it satisfies the local (or global) Markov property. The local Markov property states that all variables are independent of their non-descendants given their parents. The global Markov property with respect to $\mathcal{G}$ is defined as:

$$\mathbf{X} \perp\!\!\!\perp_{\mathcal{G}} \mathbf{Y} \mid \mathbf{Z} \implies \mathbf{X} \perp\!\!\!\perp \mathbf{Y} \mid \mathbf{Z} \tag{8}$$

for all disjoint vertex sets $\mathbf{X}, \mathbf{Y}, \mathbf{Z}$ ($\perp\!\!\!\perp_{\mathcal{G}}$ denotes d-separation in the graph) (Peters et al., 2017). Let $\mathcal{M}(\mathcal{G})$ be the set of all distributions that are Markovian with respect to $\mathcal{G}$. Two DAGs $\mathcal{G}_i$ and $\mathcal{G}_j$ are Markov equivalent if $\mathcal{M}(\mathcal{G}_i) = \mathcal{M}(\mathcal{G}_j)$. The set of all DAGs which are Markov equivalent to a DAG $\mathcal{G}$ is called the Markov Equivalence Class (MEC) of $\mathcal{G}$ (Pearl, 2011). This implies that a MEC for a given DAG is the set of DAGs which entail the same set of (conditional) independence statements.

From a graphical perspective, two DAGs are Markov equivalent if they have the same skeleton and v-structure. A skeleton is defined as the set of edges in the graph after all directed edges are converted into undirected ones, and a v-structure is any structure of the form $A \to C \leftarrow B$; here, $C$ is a collider, and $A$ and $B$ are not connected.

**MEC experiment**    We consider an expanded DAG from Figure 2 which includes a causal parent (CP) and causal ancestor (CA) of AV45. We perform a similar experiment as in section 4.1, however, we additionally consider a graph from the MEC of the expanded DAG. Figure 4 illustrates all DAGs considered for analysis. We conducted an ANOVA test for group differences and a Tukey HSD test for pairwise assessments. The ANOVA indicates a statistically significant difference between model likelihoods ($p < 0.01$). Table 3 shows the results of the pairwise Tukey HSD test. There is a statistically significant difference in data likelihood between all pairs of DAGs except the perfect map graph and its Markov equivalent graph. This would indicate that the reward signal produced by the Modelling Engine (the 'data-driven' element of the CMA; section A.1.4) is valid only up to the MEC of the data-generating process.

Table 3: Pairwise comparisons between four different graphical hypotheses using the Tukey HSD test. Mean differences in data likelihood, p-values, lower and upper bounds, and whether to reject the null hypothesis are reported.

| Model A | Model B | Mean Difference | P-Value | Lower Bound | Upper Bound | Reject Null Hypothesis |
|---------|---------|-----------------|---------|-------------|-------------|------------------------|
| Chain | D-Map | -1124.3537 | 0.001 | 1108.7990 | 1139.9084 | True |
| Chain | MEC | 466.7370 | 0.001 | -482.2917 | -451.1824 | True |
| Chain | Perfect Map | 466.0028 | 0.001 | -481.5575 | -450.4481 | True |
| D-Map | MEC | 1591.0907 | 0.001 | -1606.6454 | -1575.5361 | True |
| D-Map | Perfect Map | 1590.3565 | 0.001 | -1605.9112 | -1574.8018 | True |
| MEC | Perfect Map | -0.7342 | 0.900 | -14.8204 | 16.2889 | False |

### A.2.4    ADDITIONAL RESULTS II - LLM BEHAVIOURAL PATTERNS

Here, we discuss the use of LLMs in the context of a MEC and highlight early evidence of the utility of using LLMs to reduce the size of the MEC by excluding implausible graphs. Finally, we present additional analyses which delimit the behavioural patterns of the LLM portion of the CMA.

**The role of LLMs in the context of Markov Equivalence**    Due to any given DAG having a MEC, it is generally not possible to learn the correct graph for a given dataset of observational data alone (Peters et al., 2017). Learning the correct graph given a dataset is known as the identifiability problem. Classically, there have been two approaches to this issue (Kıcıman et al., 2023). First, one could assume that the data-generating process follows a specific functional form under which identifiability of a single graph might be possible (Glymour et al., 2019), for example by assuming a linear functional form and adding non-gaussian noise (Shimizu et al., 2006), or assuming a non-linear

functional form with additive noise (Zhang & Chan, 2006; Zhang & Hyvärinen, 2009). However, even the simple setting of a dataset with linear equations and Gaussian noise is non-identifiable (Peters et al., 2017). The second approach is to jointly model all variables using deep learning techniques. This approach is not empirically more effective on many real-world datasets and does not resolve the identifiability issue. Kıcıman et al. (2023) hypothesise that LLMs might represent a powerful utility to alleviate this issue by leveraging metadata-based reasoning to construct causal graphs. Indeed, there is early empirical evidence that LLMs can reduce the size of the MEC of a given DAG with a high probability of leaving behind the ground-truth graph when the LLMs are used as a post-processing step (Long et al., 2023a). To better understand this phenomenon, we investigate LLM behavioural patterns for reasoning over edges in the causal graph space.

**Behavioural experiment**   Figure 5 illustrates hypothesis amendments across three different types of relationship. We begin by considering the relationship between the degenerative proteins $\beta$-amyloid and P-tau. If the starting point is an 'empty' relationship, the CMA is inclined to output the edge Amyloid→Tau the majority of the time. This is also the case where an edge already exists in this direction (Figure 5; top-left and top-centre, respectively). This is in line with the 'amyloid cascade' hypothesis, which states that amyloid aggregation is upstream, and may also be causative of tau protein deposition (Hardy & Higgins, 1992; Gulisano et al., 2018). On the other hand, if the proposed edge is Tau→Amyloid, the CMA either removes this relationship or 'flips' it to align with the amyloid cascade hypothesis a majority of the time. Nonetheless, it should be noted that this edge is 'kept' in many instances (Figure 5; top-right). Overall, the CMA can be seen to prefer the causal relationship Amyloid→Tau, however will consider the opposite relationship Tau→Amyloid a non-trivial proportion of the time, irrespective of the initial orientation and/or existence of the edge. This likely reflects recent doubt in the literature about the veracity of the amyloid cascade hypothesis (Herrup, 2015), and whether there might be a feedback loop running from Tau→Amyloid, or whether both proteins may be confounded by other variables (Gulisano et al., 2018). However, this is unlikely to be the full story. In recent work, Berglund et al. (2023) described a phenomenon dubbed the 'reversal curse', which exposes a failure of generalization in auto-regressive LLMs. In brief, if a model is trained on 'A is B', it will not automatically generalize to 'B is A'. The authors do not specifically assess the implications of this phenomenon for causal questions. Nevertheless, we hypothesize that the differences in model behaviour may in part be explained by the order in which a causal relationship is presented. Additional work is required to investigate the 'reversal curse' in the context of causal reasoning.

The second row of Figure 5 considers the relationship between biological sex and P-tau. The relationship Sex→P-tau has found some empirical support in recent work (Yan et al., 2022). Where the initial relationship is empty, or already in the direction Sex→P-tau, the CMA will consider a potential causative path from sex to P-tau a majority of the time, however, will also posit that there may not be a causal relationship in many cases. This relationship is interesting because one would expect that P-tau should not have a causal effect on sex, and indeed if the relationship P-tau→Sex is proposed to the CMA (e.g. from a previous iteration), the model will either flip or remove it in almost all instances.

The third row of Figure 5 illustrates results for a relationship which should not plausibly exist in either direction: Biological sex and chronological age. Indeed, we see the CMA does not introduce this edge if it does not exist, and if the edge already exists (e.g. from a previous iteration), it is nearly always removed. The LLM modules of the CMA thus behave in a broadly expected manner; where relationships are not fully understood, there is often a default to current hypotheses, however, the CMA *can* make proposals which do not necessarily align with expert consensus in a conventional manner. Relationships which should not exist altogether are not suggested and/or removed.

Finally, we assess the effect of Retrieval Augmented Generation (RAG) patterns on causal graph construction. We use `gpt-4-0613` (with no RAG pattern) for all our main experiments, which has a cut-off date of September 2021. Utilizing a RAG pattern could allow it to account for more recent scientific literature. We consider the relationship between biological sex and P-tau once more, this time allowing the CMA to utilise a RAG pattern which accesses a vector store that contains embeddings of recently published abstracts to PubMed, including the works by (Yan et al., 2022). Results are illustrated in Figure 6. As can be seen, the additional context strongly enforces the new edge Sex→P-tau; this underscores the large effect additional in-context learning can provide.

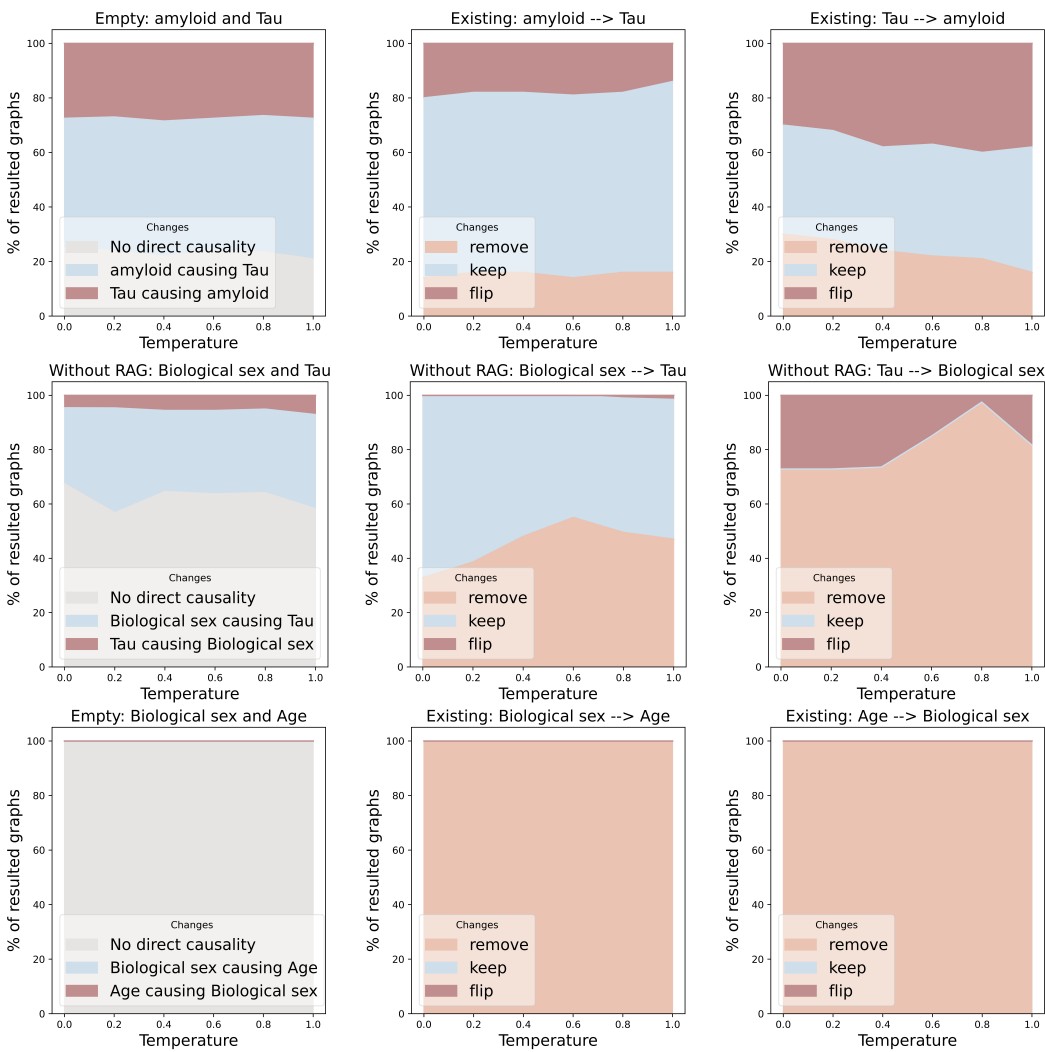

Figure 5: Hypothesis amendments across three different types of relationship; Relationships which could plausibly exist in either direction (Amyloid and Tau), relationships which plausibly exist in a single direction (Sex and Tau), and variables which have no biologically plausible causal link (Sex and Age).

Whilst the CMA represents relationships in a probabilistic manner, a strong enforcement can be made nonetheless based on a RAG pattern.

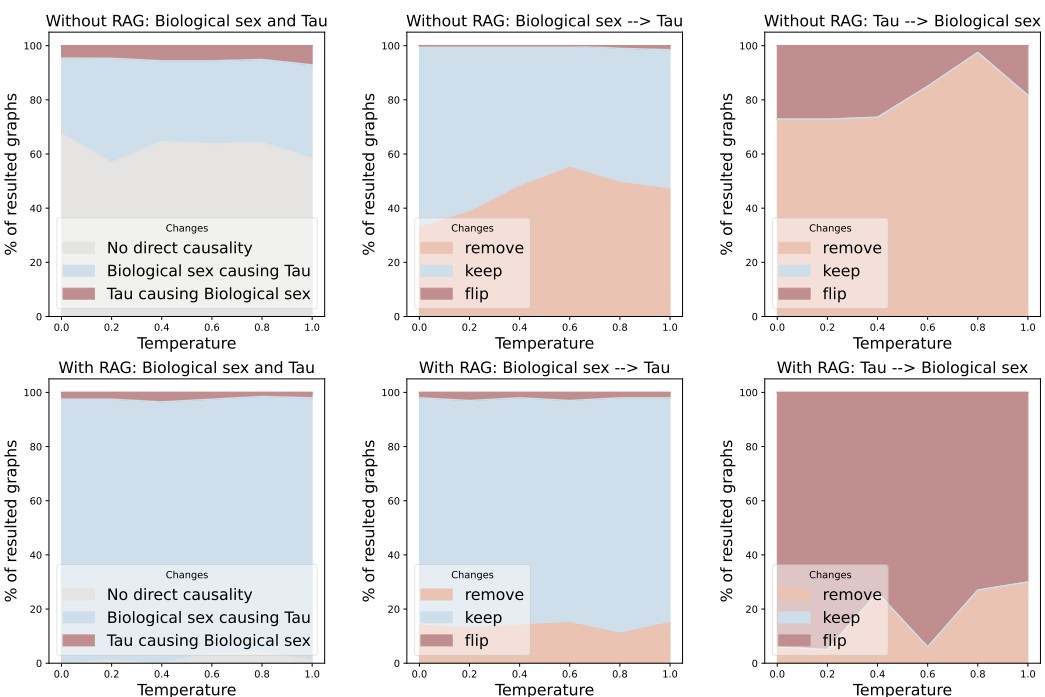

Figure 6: Hypothesis amendments with and without a Retrieval Augmented Generation (RAG) pattern for in-context learning. The additional context contains recently published information that provides evidence for a causal link between biological sex and tau pathology.

### A.3 BENCHMARK: ARCTIC SEA ICE

#### A.3.1 DATASET

The dataset describes Arctic sea ice concentration in relation to a number of atmospheric variables such as total cloud water path, relative humidity, wind components, and geopotential height (Huang et al., 2021a). The full dataset was originally collected by analysing sea ice concentration levels provided by the National Snow and Ice Data Center, and the atmospheric data was obtained via the European Centre for Medium-Range Weather Forecasts' Integrated Forecast System (IFS) (Huang et al., 2021a). Additionally, a domain-knowledge graph with 12 nodes and 48 edges was proposed to describe causal relationships between the atmospheric variables and the sea ice concentration. Full details of the dataset are provided at (Huang et al., 2021a). We summarise variables in the dataset in Table 4.

Table 4: Description of variables in Arctic Sea Ice dataset

| Variable Name | Description of Variable |
| --- | --- |
| HFLX | Sensible plus latent heat flux |
| SW | Net shortwave flux at the surface |
| LW | Net longwave flux at the surface |
| SLP | Sea level pressure |
| Precip | Total precipitation |
| RH | Relative humidity |
| u10m | Zonal (u-component) wind at 10m |
| v10m | Meridional (v-component) wind at 10m |
| sea ice | Sea ice extent in the Northern Hemisphere |
| CC | Total cloud cover |
| CW | Total cloud water path |
| GH | Geopotential heights |

#### A.3.2 EXPERIMENTAL SETUP

**Data-driven methods** We compare the CMA to the Non-combinatorial Optimization via Trace Exponential and Augmented lagrangian for Structure learning (NOTEARS) (Zheng et al., 2018), DAG Structure Learning with Graph Neural Networks (DAG-GNN) (Yu et al., 2019), and the Temporal Causal Discovery Frameworks (TCDF) (Nauta et al., 2019).

We perform a grid hyperparameter search over the two principal hyperparameters of NOTEARS, $\lambda$ and $t$ (the L1 penalty parameter and the threshold parameter, respectively). We consider the hyperparameter sets: $\lambda = \{0.001, 0.01, 0.1\}$ and $t = \{0, 0.1, 0.3\}$. This contains the default hyperparameter set $[0.1, 0.3]$ (Zheng et al., 2018). We use the implementation available at `https://github.com/xunzheng/notears`. We use the default hyperparameters for DAG-GNN, and report results over two threshold parameters, $t = \{0.1, 0.3\}$.

Whilst Huang et al. (2021a) and Kıcıman et al. (2023) both report the performance of 'temporal' versions NOTEARS and DAG-GNN, it should be noted that both of these techniques assume i.i.d. observations of the variables. For the 'static' models, all observations at different time points are considered i.i.d., and fed directly into the algorithms. The 'temporal' versions require that the data is augmented by adding lagged versions of each variable, then considering the newly created lagged versions as a set of i.i.d. observations, before running NOTEARS and DAG-GNN on the augmented datasets. The TCDF algorithm requires time-series data and is therefore given temporal data from the Arctic sea ice dataset directly.

**Metadata-driven methods** We report LLM benchmarking results based on the approach developed by Kıcıman et al. (2023). Specifically, we use the 'single prompt' approach ((Kıcıman et al., 2023); Appendix A.1, Table 14), which has been modified for the task of full graph discovery ((Kıcıman et al., 2023); section 3.2.1; p13). The full prompt alongside an example is shown in table 5. We define two types of error for a given LLM output: 1) A missing answer tag, and 2) an incor-

Table 5: Prompt asking a single question to establish whether a causal relationship should exist between two variables, and to orient the direction of the edge as appropriate, as per Kıcıman et al. (2023).

| **Single prompt** |
| --- |
| **Template:** |
| - Which cause-and-effect relationship is more likely? |
| A. changing {A} causes a change in {B}. |
| B. changing {B} causes a change in {A}. |
| C: No causal relationship exists |
| Let's work this out in a step by step way to be sure that |
| we have the right answer. Then provide your final answer |
| within the tags <Answer>A/B/C< /Answer>. |
| **Example:** |
| - Which cause-and-effect relationship is more likely? |
| A. changing the altitude causes a change in temperature. |
| B. changing the temperature causes a change in altitude. |
| C: No causal relationship exists |
| Let's work this out in a step by step way to be sure that |
| we have the right answer. Then provide your final answer |
| within the tags <Answer>A/B/C< /Answer>. |

rectly formatted answer tag. For example, the output "`<Answer>A and B</Answer>`" is not correct, because the model should choose a single output.

### A.3.3 ADDITIONAL RESULTS I - BENCHMARKS

Results for the data-driven approaches on the Arctic Sea Ice dataset are reported in Table 6. The original graph contains 48 edges; the maximum and minimum number of edges found by the NOTEARS benchmark was 15 and 0, respectively, with the maximum and minimum number of true positives being 7 and 0, respectively. An illustration of the best-performing NOTEARS output can be seen in Figure 7. The DAG-GNN algorithm discovered a higher number of total and true-positive edges with a threshold of 0.1, with a similar result over the threshold of 0.3. The best performing DAG-GNN output is shown in Figure 8. Table 7 shows the results of the LLM benchmark for the Arctic Sea Ice dataset. The minimum number of predicted edges is 47, and the maximum number is 64. Error rates were as high as 16.6% with `gpt-3.5-turbo` at a temperature of 1.0. Results show that `gpt-3.5-turbo` (Figure 9) has worse performance in comparison to `gpt-4` (Figure 10), and that the metadata-driven (LLM-based) approaches outperform the data-driven techniques. The CMA (Figure 11) outperforms both metadata- or data-driven approaches alone, with fewer false positives in particular than the LLM benchmarks.

It is interesting to note that these results vary from the metadata-based results reported by (Kıcıman et al., 2023). We believe this to be down to two reasons. First, Kıcıman et al. (2023) do not explicitly describe their error handling procedure, which may differ from ours (described above). Second, it appears that `gpt-3.5-turbo` and `gpt-4` performance on a multitude of tasks can change over time, as demonstrated by (Chen et al., 2023), which may partly explain the observed differences.

### A.3.4 ADDITIONAL RESULTS II - LW AND HFLX

The CMA identifies a high probability that net longwave flux at the surface (LW) has a direct effect on the sensible and latent heat flux (HFLX). The reasoning trace is given in Figure 12. Whilst an observational relationship exists between the variables (as shown in Figure 13, $t = 4.72, p < 0.01$), to investigate this relationship from a causal perspective, the output of the model trained by the CMA is shown in Figure 14. This illustrates that by counterfactual inference, an increase in LW leads to an increased measurement of HFLX. There is a borderline statistically significant difference between the observational and counterfactual distributions ($t = -1.89, p = 0.059$). We contend that this likely represents a missing edge from the ground-truth graph in Huang et al. (2021a).

Table 6: Results for data-driven benchmarks on Arctic Sea Ice dataset. Dashes indicate a cyclic graph was predicted for the corresponding parameters.

| | Lambda | Threshold | Edges | NHD | BHD | Ratio | TP | Prec. | Recall | F1 |
|---|---|---|---|---|---|---|---|---|---|---|
| NOTEARS | 0.001 | 0 | - | - | - | - | - | - | - | - |
| | 0.001 | 0.1 | 15 | 0.340 | 0.382 | 0.891 | 7 | 0.304 | 0.091 | 0.092 |
| | 0.001 | 0.3 | 8 | 0.301 | 0.376 | 0.815 | 6 | 0.600 | 0.045 | 0.084 |
| | 0.01 | 0 | - | - | - | - | - | - | - | - |
| | 0.01 | 0.1 | 4 | 0.319 | 0.347 | 0.920 | 3 | 0.600 | 0.021 | 0.042 |
| | 0.01 | 0.3 | 2 | 0.333 | 0.333 | 1 | 1 | 0.333 | 0.007 | 0.014 |
| | 0.1 | 0 | 0 | 0.333 | 0.333 | 1 | 0 | 0 | 0 | 0 |
| | 0.1 | 0.1 | 0 | 0.333 | 0.333 | 1 | 0 | 0 | 0 | 0 |
| | 0.1 | 0.3 | 0 | 0.333 | 0.333 | 1 | 0 | 0 | 0 | 0 |
| NOTEARS (temporal) | 0.001 | 0.3 | 8 | 0.306 | 0.375 | 0.815 | 6 | 0.750 | 0.125 | 0.214 |
| DAG-GNN | N/A | 0.1 | 43 | 0.424 | 0.396 | 1 | 15 | 0.211 | 0.132 | 0.162 |
| | N/A | 0.3 | 14 | 0.319 | 0.375 | 0.852 | 8 | 0.4 | 0.063 | 0.108 |
| DAG-GNN (temporal) | N/A | 0.3 | 16 | 0.319 | 0.431 | 0.742 | 9 | 0.562 | 0.188 | 0.281 |
| MMHC | N/A | N/A | 16 | 0.319 | 0.389 | 0.821 | 9 | 0.562 | 0.188 | 0.281 |
| GES | N/A | N/A | 43 | 0.4 | 0.479 | 0.826 | 17 | 0.395 | 0.354 | 0.374 |
| PC | N/A | N/A | 17 | 0.326 | 0.381 | 0.855 | 9 | 0.529 | 0.188 | 0.277 |
| LiNGAM | N/A | N/A | 14 | 0.375 | 0.389 | 0.864 | 4 | 0.286 | 0.083 | 0.129 |
| TCDF | N/A | N/A | 9 | 0.326 | 0.368 | 0.887 | 5 | 0.556 | 0.104 | 0.175 |

Table 7: Large language model benchmarks for the Arctic Sea Ice dataset.

| Model | Temp. | Error (%) | Edges | NHD | BHD | Ratio | TP | Prec. | Recall | F1 |
|---|---|---|---|---|---|---|---|---|---|---|
| gpt-3.5 | 0.0 | 0.0 | 57 | 0.38 | 0.49 | 0.77 | 25 | 0.44 | 0.52 | 0.48 |
| gpt-3.5 | 0.2 | 4.5 | 60 | 0.42 | 0.47 | 0.88 | 24 | 0.4 | 0.5 | 0.44 |
| gpt-3.5 | 0.4 | 6.1 | 56 | 0.4 | 0.44 | 0.91 | 23 | 0.41 | 0.48 | 0.44 |
| gpt-3.5 | 0.6 | 9.1 | 54 | 0.4 | 0.47 | 0.85 | 22 | 0.41 | 0.46 | 0.43 |
| gpt-3.5 | 0.8 | 4.5 | 57 | 0.38 | 0.53 | 0.71 | 25 | 0.44 | 0.52 | 0.48 |
| gpt-3.5 | 1.0 | 16.6 | 47 | 0.44 | 0.44 | 1.0 | 16 | 0.34 | 0.33 | 0.34 |
| gpt-4 | 0.0 | 1.5 | 64 | 0.39 | 0.57 | 0.68 | 28 | 0.44 | 0.58 | 0.5 |
| gpt-4 | 0.2 | 3.0 | 56 | 0.39 | 0.54 | 0.72 | 24 | 0.43 | 0.5 | 0.46 |
| gpt-4 | 0.4 | 1.5 | 54 | 0.35 | 0.51 | 0.68 | 26 | 0.48 | 0.54 | 0.51 |
| gpt-4 | 0.6 | 3.0 | 60 | 0.4 | 0.46 | 0.88 | 25 | 0.42 | 0.52 | 0.46 |
| gpt-4 | 0.8 | 6.1 | 51 | 0.37 | 0.42 | 0.87 | 23 | 0.45 | 0.48 | 0.46 |
| gpt-4 | 1.0 | 3.0 | 59 | 0.35 | 0.48 | 0.74 | 28 | 0.47 | 0.58 | 0.52 |
| CMA | 0.6 | 0 | 36 | 0.25 | 0.54 | 0.46 | 24 | 0.67 | 0.5 | 0.57 |

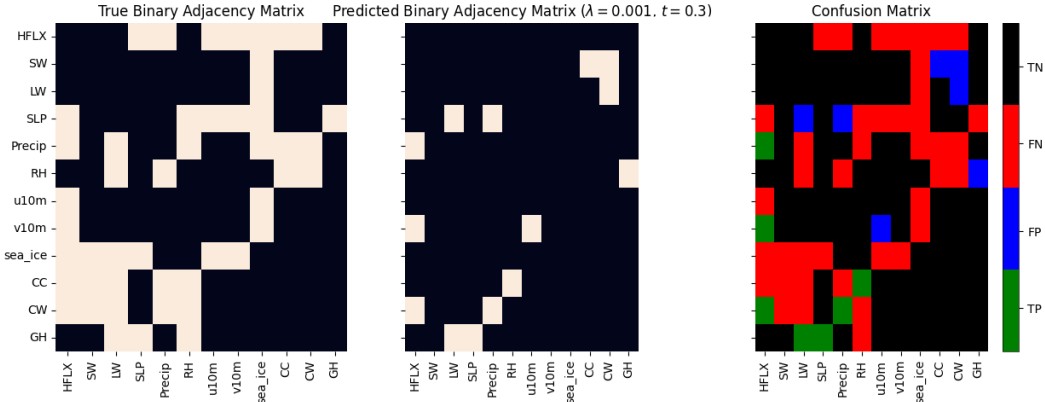

Figure 7: Comparison between ground-truth adjacency matrix and predicted adjacency matrix for the **NOTEARS** predicted DAG for the Arctic Sea Ice dataset ($\lambda$=0.001, $t$=0.3). We additionally provide an illustration of True Positives (TP), False Positives (FP), True Negatives (TN), and False Negatives (FN) in the third panel.

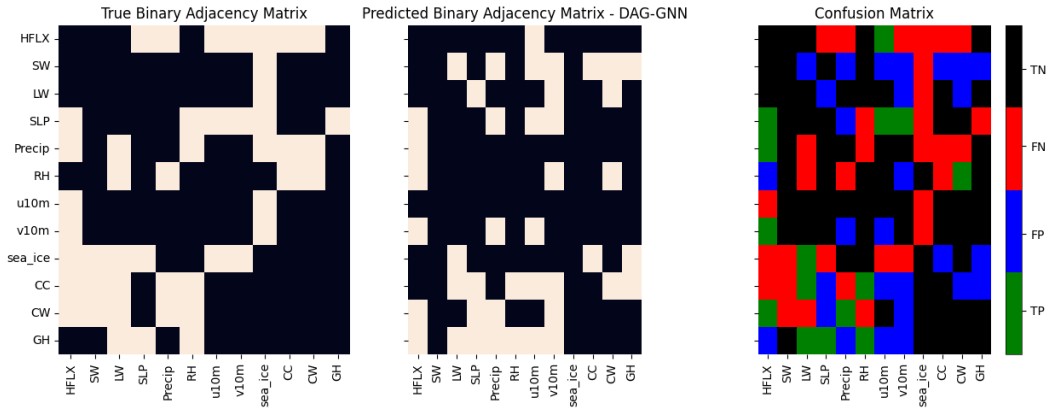

Figure 8: Comparison between ground-truth adjacency matrix and predicted adjacency matrix for the **DAG-GNN** predicted DAG for the Arctic Sea Ice dataset. We additionally provide an illustration of True Positives (TP), False Positives (FP), True Negatives (TN), and False Negatives (FN) in the third panel.

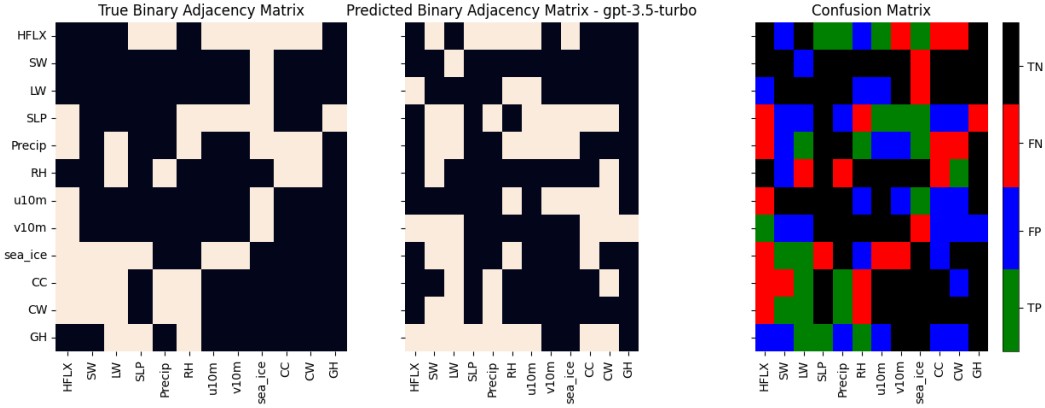

Figure 9: Comparison between ground-truth adjacency matrix and predicted adjacency matrix for the **gpt-3.5-turbo** predicted DAG for the Arctic Sea Ice dataset. We additionally provide an illustration of True Positives (TP), False Positives (FP), True Negatives (TN), and False Negatives (FN) in the third panel.

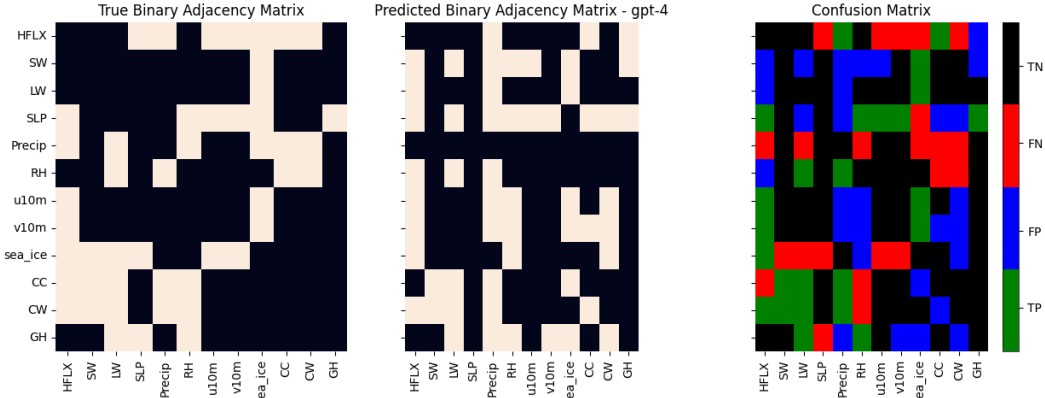

Figure 10: Comparison between ground-truth adjacency matrix and predicted adjacency matrix for the **gpt-4** predicted DAG for the Arctic Sea Ice dataset. We additionally provide an illustration of True Positives (TP), False Positives (FP), True Negatives (TN), and False Negatives (FN) in the third panel.

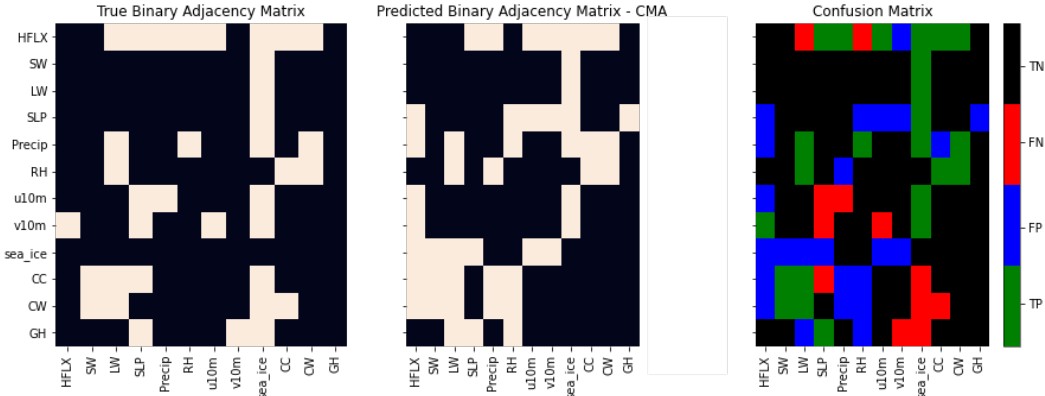

Figure 11: Comparison between ground-truth adjacency matrix and predicted adjacency matrix for the **CMA** predicted DAG for the Arctic Sea Ice dataset. We additionally provide an illustration of True Positives (TP), False Positives (FP), True Negatives (TN), and False Negatives (FN) in the third panel.

> **LW -> HFLX Reasoning trace**
>
> {"**explanation**": "Net longwave flux at the surface directly affects the sensible and latent heat flux. As the net longwave flux increases, the surface warms, leading to increased evaporation (latent heat flux) and conduction (sensible heat flux). However, it is also plausible that changes in the sensible and latent heat flux could affect the net longwave flux at the surface. An increase in evaporation or conduction could cool the surface, reducing the net longwave flux. Therefore, there is a high probability of keeping this relationship and a smaller probability of flipping it. The possibility of removing this relationship is low, as it is well-established in meteorology that these variables are interconnected. Factors such as surface characteristics, atmospheric humidity, and wind speed could mediate or moderate this relationship."}

Figure 12: The CMA reasoning trace for the relationship between net longwave flux at the surface (LW) and sensible and latent heat flux (HFLX). The model produces an average probability of 70% a causal link from longwave flux at the surface to sensible and latent heat flux.

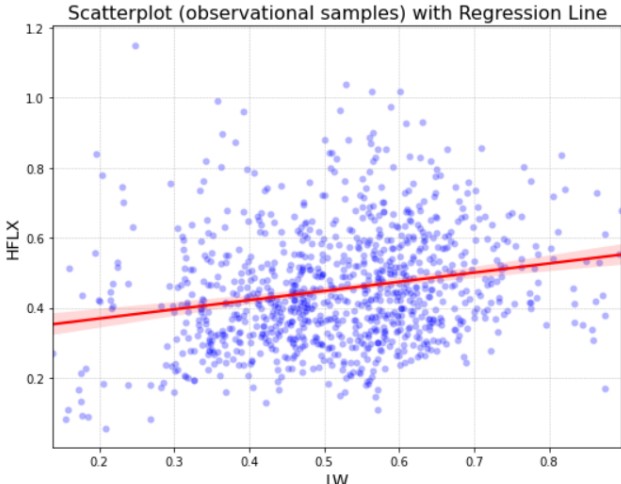

Figure 13: Observational relationship between net longwave flux at the surface (LW) and latent heat flux (HFLX). As can be seen, there is a statistical association between the variables.

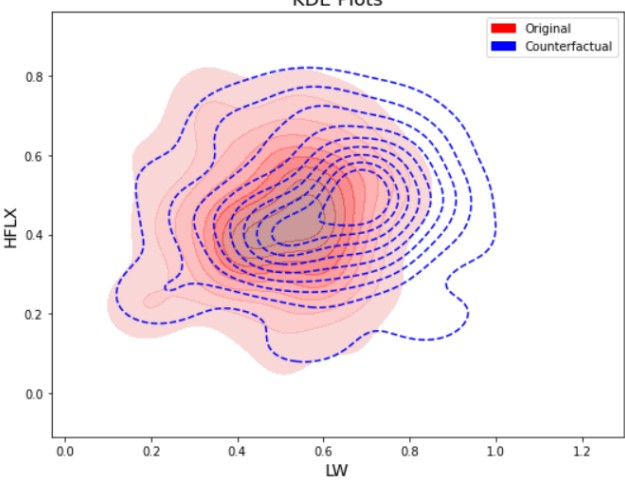

Figure 14: Output from a Deep SCM (DSCM) trained by the CMA demonstrating an experimental intervention on net longwave flux at the surface (LW). The DSCM includes a causal link from LW to latent heat flux (HFLX). We perform a counterfactual query, which sets the value of LW to be 30% greater. The resultant counterfactual distribution (dashed blue) is shifted with respect to the observational distribution (block red). The model suggests that increasing LW leads to an increased measurement of HFLX.

### A.4 BENCHMARK: SANGIOVESE

#### A.4.1 DATASET

The Sangiovese dataset is a causal discovery benchmark derived from the conditional linear Gaussian network model built in Magrini et al. (2017) from data collected about grape quality in Sangiovese vineyards in Tuscany. In summary, the original dataset consisted of the measurement of various characteristics of grapes collected during growth and post-harvest, such as their pH, polyphenol content, or the mean grape weight. Data was collected between 2007 and 2009. Additionally, different well-established interventions for canopy management were applied to different parts of the vineyard. Three intervention types were considered; defoliation and bunch thinning (not applied or applied at 50%), and harvest time (technological or late). The original DAG consists of 15 nodes and 55 edges; the original DAG and its constituent nodes are described in Table 8.

Table 8: Description of variables in Sangiovese dataset

| Variable Name | Description of Variable |
| --- | --- |
| Treatment | Treatment Type Applied to Must |
| SproutN | Mean number of sprouts |
| BunchN | Mean number of bunches |
| GrapeW | Mean weight of grapes |
| WoodW | Weight of Wood |
| SPAD06 | Soil-Plant Analysis Development in June |
| SPAD08 | Soil-Plant Analysis Development in August |
| NDVI06 | Normalized Difference Vegetation Index in June |
| NDVI08 | Normalized Difference Vegetation Index in August |
| Acid | Total acidity of each must |
| Potass | Potassium content of each must |
| pH | pH of each must |
| Anthoc | Total anthocyanin content of each must |
| Polyph | Total polyphenol content of each must |
| Brix | Potential alcohol |

#### A.4.2 EXPERIMENTAL SETUP

The experimental setup is the same as in the Arctic sea ice benchmark, described in Appendix A.3.2.

#### A.4.3 ADDITIONAL RESULTS

Results for NOTEARS and DAG-GNN are shown in Table 9. The maximum and minimum number of edges found by the NOTEARS algorithm were 22 and 1, respectively, with the maximum and minimum number of true positives being 13 and 1, respectively. No significant trend was observed over the NHD for NOTEARS across different hyperparameter settings. A similar performance was observed for DAG-GNN and NOTEARS run at $\lambda$=0.001. The best performing NOTEARS output (hyperparameters $\lambda$=0.001, $t$=0.3) is shown in 15, with results for the DAG-GNN shown in Figure 16. As can be seen, both algorithms show similar patterns in true positives and false negatives.

Table 10 shows the results of the LLM benchmarks. Whilst `gpt-3.5-turbo` is competitive with the data-driven causal discovery approaches, `gpt-4` outperforms them across several temperature settings with an average error rate of $0.25\%$. Figures 17 and 18 illustrate the predicted adjacency matrices as compared to the ground-truth for `gpt-3.5-turbo` and `gpt-4`, respectively.

Both data- and metadata-driven approaches struggle with the Sangiovese dataset. This would suggest that: 1) Establishing causal structure from asymmetries in the dataset is challenging, AND 2) there is a lack of evidence to support making causal claims for many of the variables, which likely reflects less robust domain knowledge in general. As such, the CMA also struggles here and produces only marginally better results than either approach alone 19.

Table 9: Results for data-driven benchmarks on Sangiovese grapes dataset. Dashes indicate a cyclic graph was predicted for the corresponding parameters.

| | Lambda | Threshold | Edges | NHD | BHD | Ratio | TP | Prec. | Recall | F1 |
|---|---|---|---|---|---|---|---|---|---|---|
| NOTEARS | 0.001 | 0 | - | - | - | - | - | - | - | - |
| | 0.001 | 0.1 | 22 | 0.240 | 0.362 | 0.662 | 13 | 0.419 | 0.102 | 0.164 |
| | 0.001 | 0.3 | 16 | 0.255 | 0.286 | 0.770 | 10 | 0.455 | 0.075 | 0.129 |
| | 0.01 | 0 | - | - | - | - | - | - | - | - |
| | 0.01 | 0.1 | 9 | 0.255 | 0.286 | 0.893 | 5 | 0.385 | 0.035 | 0.064 |
| | 0.01 | 0.3 | 6 | 0.240 | 0.281 | 0.855 | 5 | 0.714 | 0.035 | 0.066 |
| | 0.1 | 0 | - | - | - | - | - | - | - | - |
| | 0.1 | 0.1 | 2 | 0.25 | 0.270 | 0.925 | 2 | 1 | 0.013 | 0.026 |
| | 0.1 | 0.3 | 1 | 0.255 | 0.265 | 0.961 | 1 | 1 | 0.006 | 0.013 |
| DAG-GNN | N/A | 0.1 | 71 | 0.398 | 0.439 | 0.907 | 22 | 0.183 | 0.202 | 0.192 |
| | N/A | 0.3 | 27 | 0.265 | 0.337 | 0.788 | 13 | 0.317 | 0.102 | 0.155 |
| MMHC | N/A | N/A | 41 | 0.214 | 0.408 | 0.525 | 25 | 0.610 | 0.490 | 0.543 |
| GES | N/A | N/A | 36 | 0.291 | 0.393 | 0.741 | 15 | 0.417 | 0.294 | 0.345 |
| PC | N/A | N/A | 25 | 0.244 | 0.347 | 0.706 | 14 | 0.560 | 0.275 | 0.368 |
| LiNGAM | N/A | N/A | 23 | 0.296 | 0.357 | 0.823 | 8 | 0.348 | 0.157 | 0.516 |

Table 10: Large language model benchmarks for the Sangiovese dataset.

| Model | Temp. | Error (%) | Edges | NHD | BHD | Ratio | TP | Prec. | Recall | F1 |
|---|---|---|---|---|---|---|---|---|---|---|
| gpt-3.5 | 0.00 | 0.00 | 89 | 0.43 | 0.51 | 0.84 | 28 | 0.31 | 0.55 | 0.40 |
| gpt-3.5 | 0.20 | 0.00 | 89 | 0.41 | 0.56 | 0.73 | 30 | 0.34 | 0.59 | 0.43 |
| gpt-3.5 | 0.40 | 1.50 | 84 | 0.41 | 0.46 | 0.89 | 27 | 0.32 | 0.53 | 0.40 |
| gpt-3.5 | 0.60 | 0.00 | 86 | 0.37 | 0.48 | 0.77 | 32 | 0.37 | 0.63 | 0.47 |
| gpt-3.5 | 0.80 | 3.00 | 88 | 0.39 | 0.49 | 0.79 | 31 | 0.35 | 0.61 | 0.45 |
| gpt-3.5 | 1.00 | 9.10 | 82 | 0.41 | 0.47 | 0.87 | 26 | 0.32 | 0.51 | 0.39 |
| gpt-4 | 0.00 | 0.00 | 67 | 0.32 | 0.49 | 0.65 | 28 | 0.42 | 0.55 | 0.47 |
| gpt-4 | 0.20 | 0.00 | 69 | 0.34 | 0.48 | 0.70 | 27 | 0.39 | 0.53 | 0.45 |
| gpt-4 | 0.40 | 0.00 | 65 | 0.33 | 0.48 | 0.68 | 26 | 0.40 | 0.51 | 0.45 |
| gpt-4 | 0.60 | 0.00 | 63 | 0.35 | 0.47 | 0.74 | 23 | 0.37 | 0.45 | 0.40 |
| gpt-4 | 0.80 | 1.50 | 59 | 0.34 | 0.40 | 0.85 | 22 | 0.37 | 0.43 | 0.40 |
| gpt-4 | 1.00 | 0.00 | 61 | 0.33 | 0.42 | 0.78 | 24 | 0.39 | 0.47 | 0.43 |
| CMA | 0.6 | 0.00 | 24 | 0.23 | 0.36 | 0.63 | 15 | 0.62 | 0.29 | 0.40 |

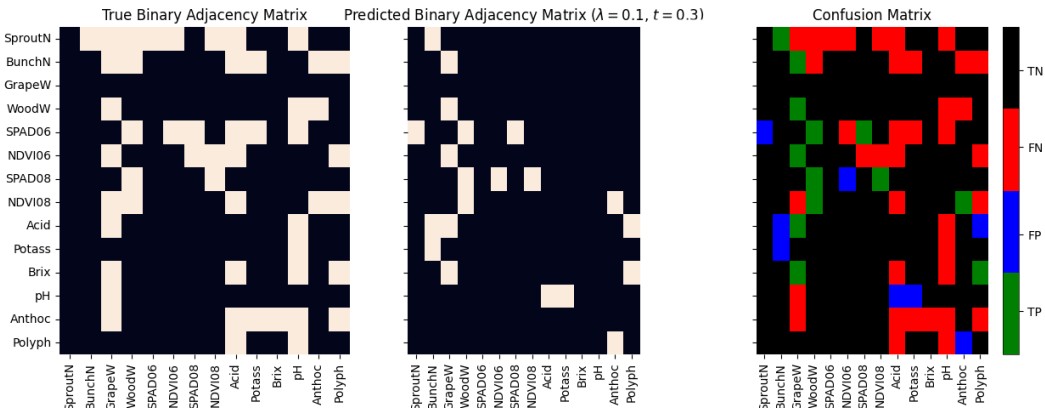

Figure 15: Comparison between ground-truth adjacency matrix and predicted adjacency matrix for the **NOTEARS** predicted DAG from the Sangiovese grapes dataset ($\lambda$=0.001, $t$=0.3). We additionally provide an illustration of True Positives (TP), False Positives (FP), True Negatives (TN), and False Negatives (FN) in the third panel.

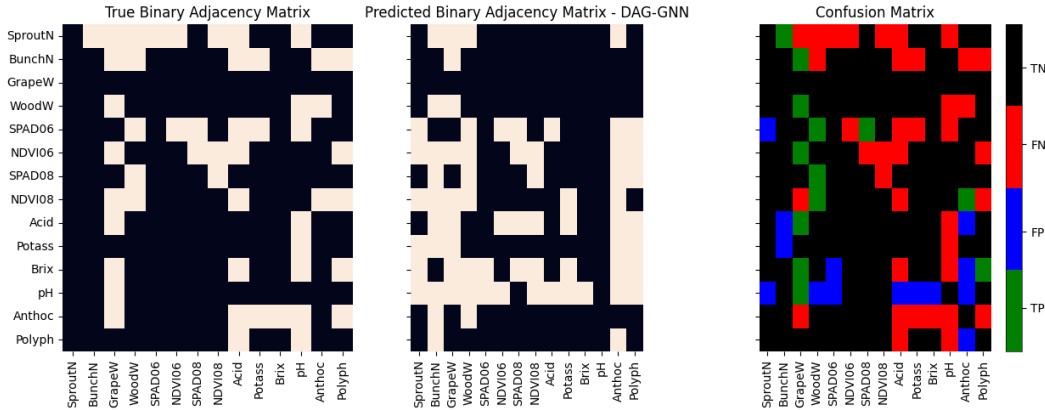

Figure 16: Comparison between ground-truth adjacency matrix and predicted adjacency matrix for the **DAG-GNN** predicted DAG from the Sangiovese grapes dataset. We additionally provide an illustration of True Positives (TP), False Positives (FP), True Negatives (TN), and False Negatives (FN) in the third panel.

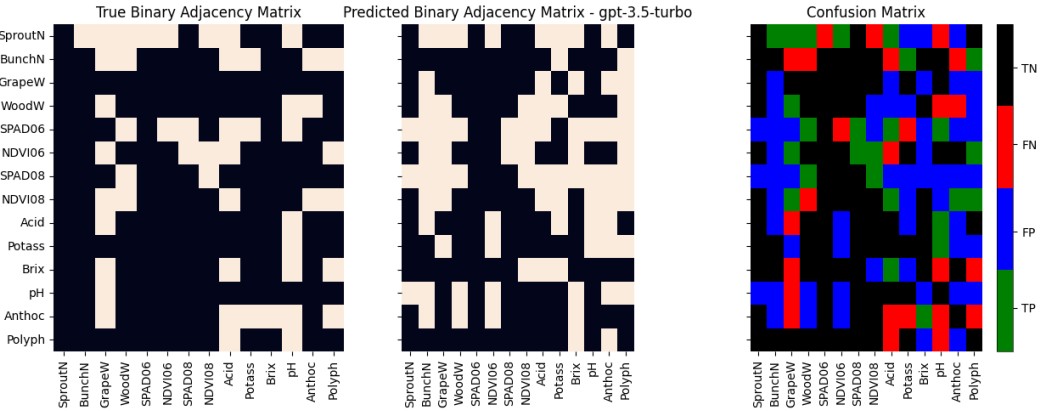

Figure 17: Comparison between ground-truth adjacency matrix and predicted adjacency matrix for the **gpt-3.5-turbo** predicted graph from the Sangiovese grapes dataset. We additionally provide an illustration of True Positives (TP), False Positives (FP), True Negatives (TN), and False Negatives (FN) in the third panel.

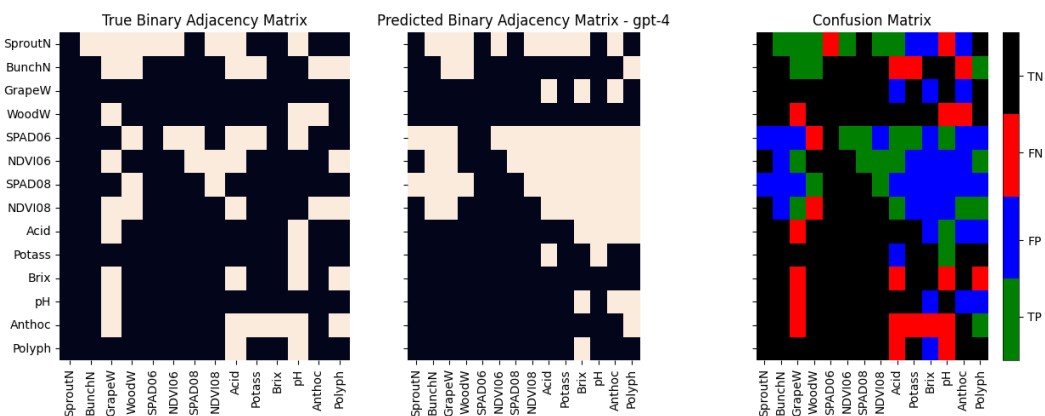

Figure 18: Comparison between ground-truth adjacency matrix and predicted adjacency matrix for the **gpt-4** predicted graph from the Sangiovese grapes dataset. We additionally provide an illustration of True Positives (TP), False Positives (FP), True Negatives (TN), and False Negatives (FN) in the third panel.

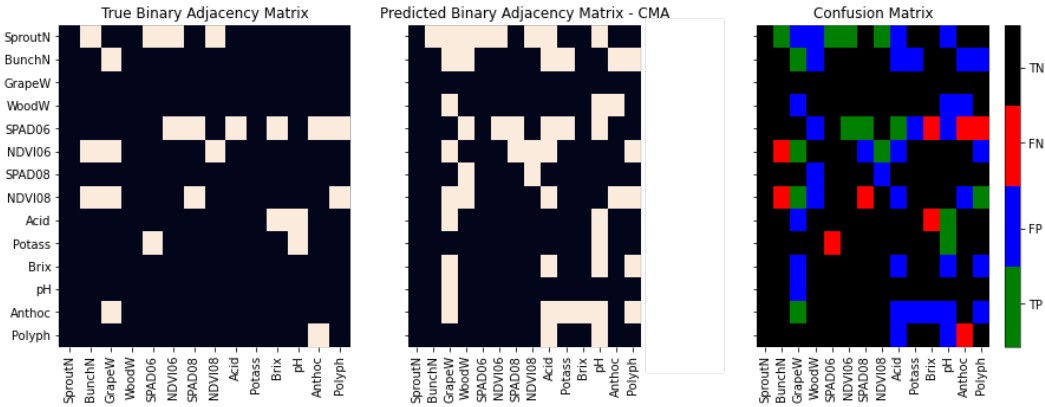

Figure 19: Comparison between ground-truth adjacency matrix and predicted adjacency matrix for the **CMA** predicted DAG from the Sangiovese grapes dataset. We additionally provide an illustration of True Positives (TP), False Positives (FP), True Negatives (TN), and False Negatives (FN) in the third panel.

## A.5 BENCHMARK: ALZHEIMER'S DISEASE

### A.5.1 DATASET

**Variable description** The variables considered for this benchmark are described in table 11.

Table 11: Description of variables in Alzheimer's Disease dataset

| Variable Name | Description of Variable |
| --- | --- |
| APOE4 | Expression level of APOE4 gene |
| Sex | Biological Sex of Patient |
| Age | Age of Patient |
| Education | Educational attainment (years) |
| AV45 | Beta Amyloid protein level measured by Florbetapir F 18 |
| P-tau | Phosphorylated-tau deposition |
| Brain Volume | Total Brain Matter Volume of Patient |
| Ventricular Volume | Total Ventricular Volume of Patient |
| MOCA Score | Montreal Cognitive Assessment Score |

**Data generation** The ground-truth DAG is created in collaboration with 5 domain experts, with expertise in either clinical and/or academic neurology (with a sub-specialist interest in neurodegenerative/Alzheimer's disease) or neuroradiology with a subspecialist interest in neurodegenerative disease. To accrue expert causal graphs, we build a front-end application in the JavaScript programming language (Figure 20). We include edges in the final ground-truth DAG which are proposed by at least 2/5 experts as our consensus heuristic. Figure 21 shows individual expert-based causal graphs, and Figure 22 shows a stacked graph as well as the final ground-truth DAG.

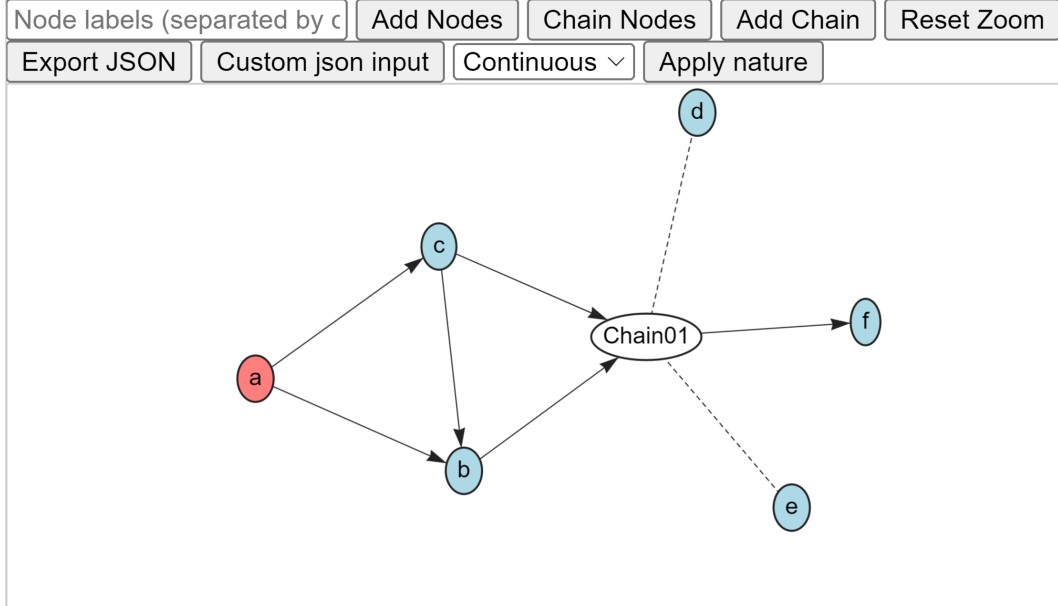

Figure 20: A front-end application to create causal graphs in collaboration with domain experts. Edges can be added to the graph to create a Markovian Directed Acyclic Graph (DAG) as can be seen for a → b, for example. Associative relationships can also be represented using chains, for example, Chain01 contains nodes d and e, and itself represents a causal parent of node f. The colouring represents applying a 'nature' to the node, for example 'discrete' or 'continuous'. In this case, red nodes are discrete and blue nodes are continuous. The graph can be exported as a JSON object, which can be used for downstream analyses.

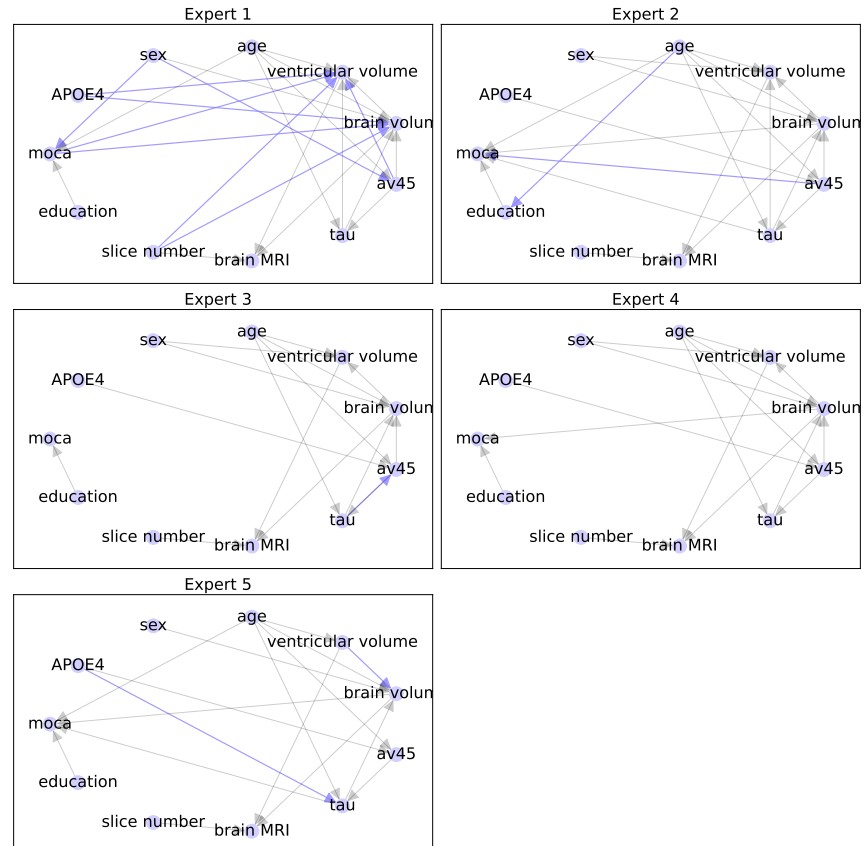

Figure 21: Illustration of causal graphs encoded as Directed Acyclic Graphs (DAGs) for each expert. The purple edges indicate unique graph-specific edges, whilst gray edges indicate that the edge exists in at least one other expert graph.

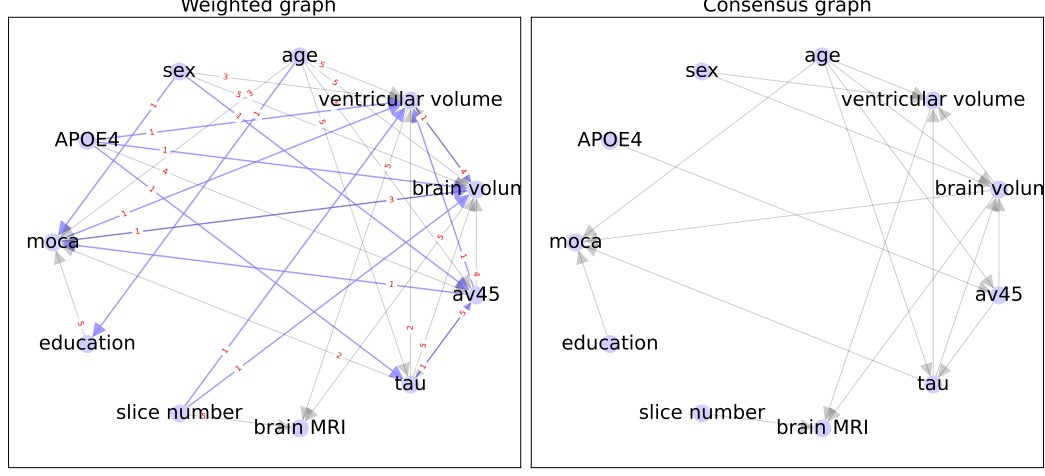

Figure 22: **Left panel**: A weighted graph which represents an overlay of all individual expert graphs. Here, the purple edges represent edges for removal as they are proposed by a single expert. **Right panel**: The final concensus graph. Edges are retained if they are proposed by ≥2 experts.

### A.5.2 EXPERIMENTAL SETUP

The experimental setup is the same as in the Arctic sea ice benchmark, described in Appendix A.3.2.

### A.5.3 ADDITIONAL RESULTS

Results for the NOTEARS and DAG-GNN algorithm can be found in Table 12. The best NOTEARS output is visualized in Figure 23, whilst the DAG-GNN result is shown in Figure 24. The LLM benchmark results are shown in Table 13. The metadata-based (LLM) methods outperform the data-driven approaches. The CMA output is visualised in Figure 27. As can be seen, the CMA outperform both data- and metadata-driven approaches alone.

Table 12: Results for data-driven benchmarks on Alzheimer's Disease dataset. Dashes indicate a cyclic graph was predicted for the corresponding parameters.

|  | Lambda | Threshold | Edges | NHD | BHD | Ratio | TP | Prec. | Recall | F1 |
|---|---|---|---|---|---|---|---|---|---|---|
| NOTEARS | 0.001 | 0 | - | - | - | - | - | - | - | - |
|  | 0.001 | 0.1 | 12 | 0.222 | 0.296 | 0.75 | 5 | 0.263 | 0.132 | 0.175 |
|  | 0.001 | 0.3 | 10 | 0.222 | 0.321 | 0.692 | 4 | 0.250 | 0.1 | 0.143 |
|  | 0.01 | 0 | - | - | - | - | - | - | - | - |
|  | 0.01 | 0.1 | 11 | 0.210 | 0.284 | 0.739 | 5 | 0.294 | 0.132 | 0.182 |
|  | 0.01 | 0.3 | 10 | 0.222 | 0.272 | 0.818 | 4 | 0.250 | 0.1 | 0.143 |
|  | 0.1 | 0 | - | - | - | - | - | - | - | - |
|  | 0.1 | 0.1 | 9 | 0.235 | 0.259 | 0.905 | 3 | 0.200 | 0.071 | 0.105 |
|  | 0.1 | 0.3 | 7 | 0.210 | 0.259 | 0.810 | 3 | 0.273 | 0.071 | 0.113 |
| DAG-GNN | N/A | 0.1 | 28 | 0.370 | 0.370 | 1.0 | 7 | 0.143 | 0.206 | 0.169 |
|  | N/A | 0.3 | 28 | 0.370 | 0.444 | 0.833 | 7 | 0.143 | 0.206 | 0.169 |
| MMHC | N/A | N/A | 11 | 0.160 | 0.284 | 0.565 | 7 | 0.636 | 0.438 | 0.519 |
| GES | N/A | N/A | 10 | 0.173 | 0.272 | 0.636 | 6 | 0.600 | 0.375 | 0.462 |
| PC | N/A | N/A | 16 | 0.198 | 0.321 | 0.615 | 8 | 0.500 | 0.500 | 0.500 |
| LiNGAM | N/A | N/A | 5 | 0.185 | 0.259 | 0.714 | 3 | 0.600 | 0.188 | 0.286 |

Table 13: Large language model benchmarks for the Alzheimer's disease dataset.

| Model | Temp. | Error (%) | Edges | NHD | BHD | Ratio | TP | Prec. | Recall | F1 |
|---|---|---|---|---|---|---|---|---|---|---|
| gpt-3.5 | 0.0 | 4.5 | 21 | 0.21 | 0.38 | 0.55 | 10 | 0.48 | 0.62 | 0.54 |
| gpt-3.5 | 0.2 | 7.6 | 19 | 0.16 | 0.38 | 0.42 | 11 | 0.58 | 0.69 | 0.63 |
| gpt-3.5 | 0.4 | 6.0 | 19 | 0.14 | 0.33 | 0.41 | 12 | 0.63 | 0.75 | 0.69 |
| gpt-3.5 | 0.6 | 9.1 | 18 | 0.20 | 0.35 | 0.57 | 9 | 0.50 | 0.56 | 0.53 |
| gpt-3.5 | 0.8 | 6.0 | 20 | 0.20 | 0.32 | 0.62 | 10 | 0.50 | 0.62 | 0.56 |
| gpt-3.5 | 1.0 | 9.1 | 18 | 0.17 | 0.4 | 0.44 | 10 | 0.56 | 0.62 | 0.59 |
| gpt-4 | 0.0 | 0.0 | 25 | 0.14 | 0.48 | 0.28 | 15 | 0.6 | 0.94 | 0.73 |
| gpt-4 | 0.2 | 0.0 | 23 | 0.11 | 0.46 | 0.24 | 15 | 0.65 | 0.94 | 0.77 |
| gpt-4 | 0.4 | 1.5 | 22 | 0.12 | 0.4 | 0.31 | 14 | 0.64 | 0.88 | 0.74 |
| gpt-4 | 0.6 | 0.0 | 23 | 0.11 | 0.41 | 0.27 | 15 | 0.65 | 0.94 | 0.77 |
| gpt-4 | 0.8 | 1.5 | 21 | 0.11 | 0.41 | 0.27 | 14 | 0.67 | 0.88 | 0.76 |
| gpt-4 | 1.0 | 0.0 | 21 | 0.11 | 0.43 | 0.26 | 14 | 0.67 | 0.88 | 0.76 |
| CMA | 0.6 | 0 | 16 | 0.07 | 0.35 | 0.21 | 13 | 0.81 | 0.81 | 0.81 |

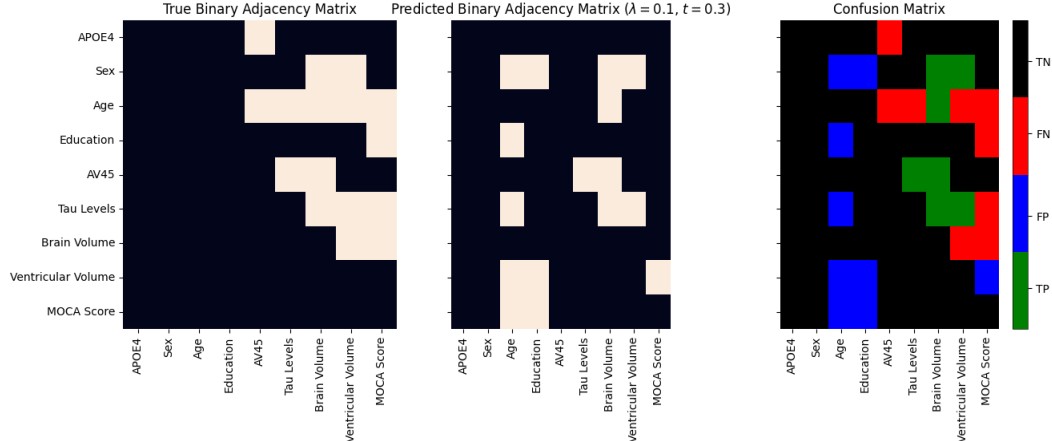

Figure 23: Comparison between ground-truth adjacency matrix and predicted adjacency matrix for the **NOTEARS** predicted DAG for the Alzheimer's dataset ($\lambda$=0.1, $t$=0.3). We additionally provide an illustration of True Positives (TP), False Positives (FP), True Negatives (TN), and False Negatives (FN) in the third panel.

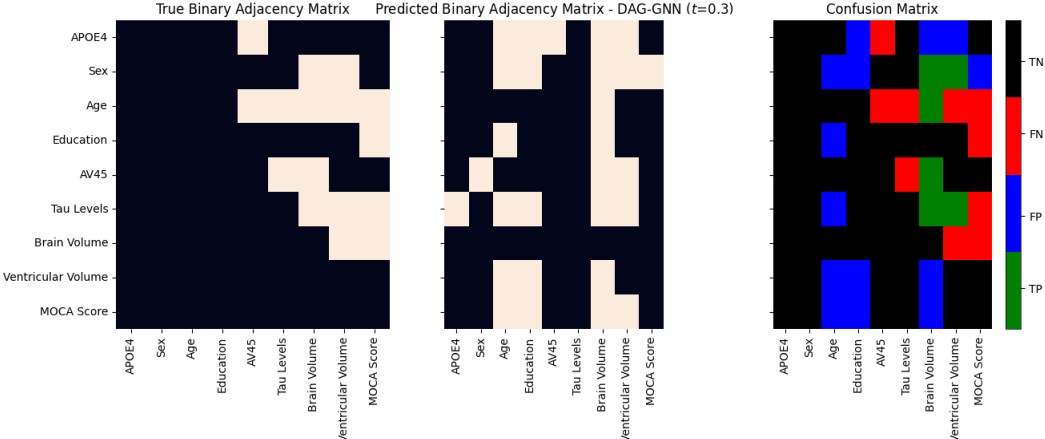

Figure 24: Comparison between ground-truth adjacency matrix and predicted adjacency matrix for the **DAG-GNN** predicted DAG for the Alzheimer's Disease dataset. We additionally provide an illustration of True Positives (TP), False Positives (FP), True Negatives (TN), and False Negatives (FN) in the third panel.

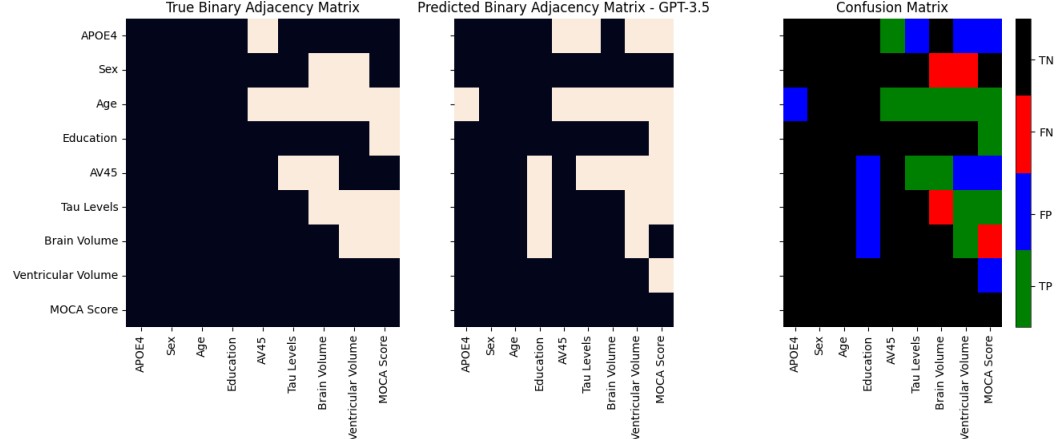

Figure 25: Comparison between ground-truth adjacency matrix and predicted adjacency matrix for the **GPT3.5** predicted DAG for the Alzheimer's dataset. We additionally provide an illustration of True Positives (TP), False Positives (FP), True Negatives (TN), and False Negatives (FN) in the third panel.

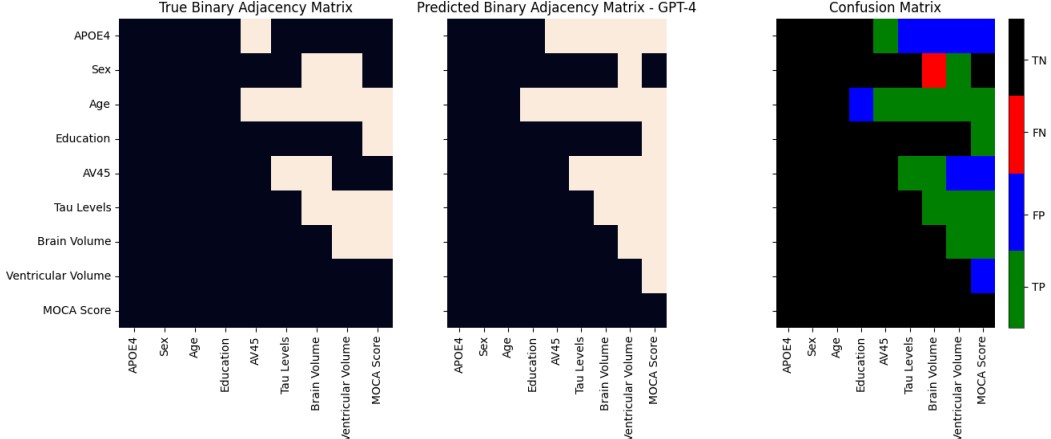

Figure 26: Comparison between ground-truth adjacency matrix and predicted adjacency matrix for the **GPT4** predicted DAG for the Alzheimer's dataset. We additionally provide an illustration of True Positives (TP), False Positives (FP), True Negatives (TN), and False Negatives (FN) in the third panel.

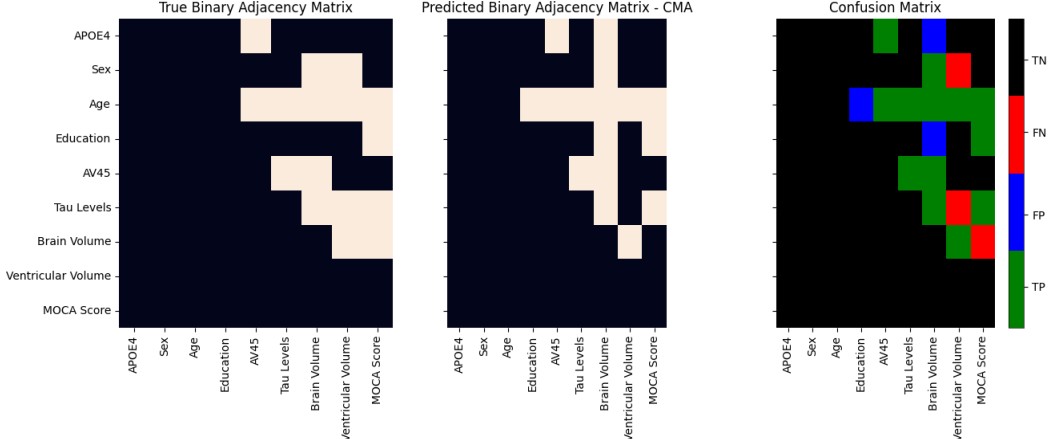

Figure 27: Comparison between ground-truth adjacency matrix and predicted adjacency matrix for the **CMA** predicted DAG for the Alzheimer's dataset. We additionally provide an illustration of True Positives (TP), False Positives (FP), True Negatives (TN), and False Negatives (FN) in the third panel.

### A.5.4 SENSITIVITY ANALYSIS: DEGRADATION OF CAUSAL LINKS

In this sensitivity analysis we perturb the dataset variables using Gaussian noise with increasingly large standard deviation values, which has the effect of reducing the total variation explained by any given variable on its children in the causal graph. The effect of these perturbations can be visualised in Figure 28. We run the NOTEARS, DAG-GNN, and CMA algorithms on each noise level and compare their performances.

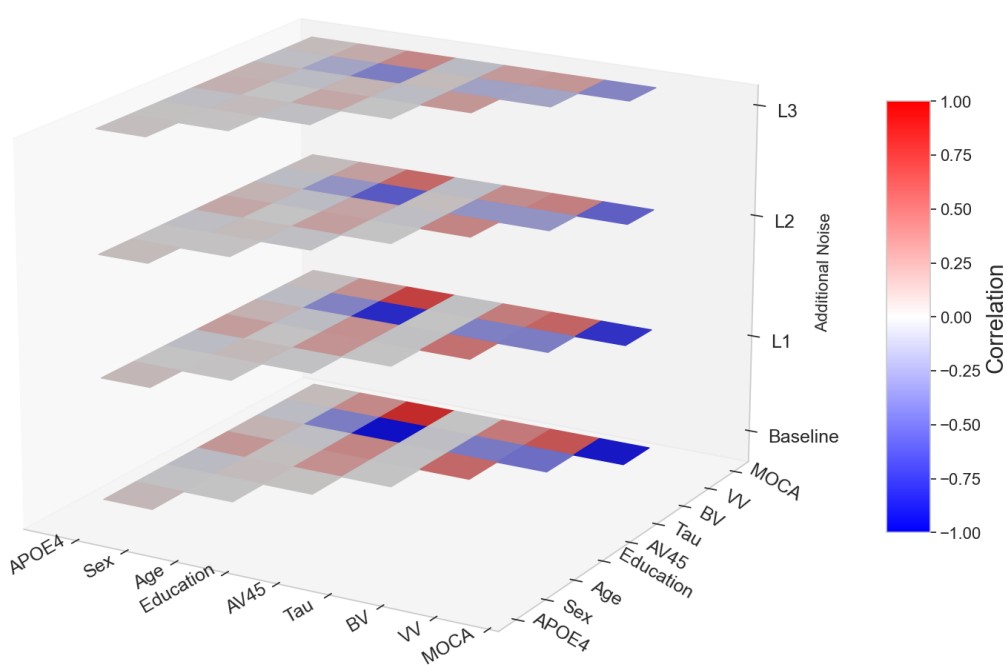

Figure 28: The strengths of the causal links between the variables are systematically degraded in the AD benchmark dataset. There are three noise levels which add Gaussian noise with mean 0 and standard deviations of 0.4 for L1, 0.8 for L2, and 1.2 for L3. As can be seen, the relationships between all variables are reduced as more noise is introduced.

**NOTEARS results**   Inspecting results for NOTEARS in Tables 14, 15, and 16 reveals a number of patterns. First, causal discovery performance is worse with increasing values of the lambda parameter, and this is trend is consistent across all noise levels. Second, performance broadly degrades with higher noise levels. The best performance is observed at the 'L1' noise level (Figure 28; L1) with parameters $\lambda=0.001$ and $t=0.1$. The worst results were osberved with at the highest noise level L3, with parameters $\lambda=0.1$ and $t=0.3$.

**DAG-GNN results**   Inspecting results for DAG-GNN in Tables 14, 15, and 16 showcases the relative robustness of DAG-GNN across noise levels. The same number of true positives are detected across all levels, with less variability observed across other metrics.

**CMA results**   Similarly to the DAG-GNN algorithm, the CMA remains relatively robust across all noise levels, however outperforms both approaches with a lower NHD/BHD ratio. Indeed, it is likely this improved performance is partly due to the metadata-based modules of the framework, which can still propose reasonable causal structures despite weak causal relationships in the dataset. A visual summary of these results is illustrated in Figure 29.

Table 14: Results for NOTEARS, DAG-GNN, and CMA benchmarks on the Alzheimer's benchmark dataset - L1 noise level $\epsilon \sim \mathcal{N}(0, 0.4)$. Dashes indicate a cyclic graph was predicted for the corresponding parameters.

|  | Lambda | Threshold | Edges | NHD | BHD | Ratio | TP | Prec. | Recall | F1 |
|---|---|---|---|---|---|---|---|---|---|---|
| NOTEARS | 0.001 | 0 | - | - | - | - | - | - | - | - |
|  | 0.001 | 0.1 | 14 | 0.247 | 0.346 | 0.714 | 5 | 0.217 | 0.131 | 0.164 |
|  | 0.001 | 0.3 | 11 | 0.259 | 0.284 | 0.913 | 3 | 0.157 | 0.071 | 0.098 |
|  | 0.01 | 0 | - | - | - | - | - | - | - | - |
|  | 0.01 | 0.1 | 14 | 0.247 | 0.259 | 0.714 | 5 | 0.217 | 0.132 | 0.164 |
|  | 0.01 | 0.3 | 11 | 0.259 | 0.259 | 0.913 | 3 | 0.157 | 0.071 | 0.098 |
|  | 0.1 | 0 | - | - | - | - | - | - | - | - |
|  | 0.1 | 0.1 | 11 | 0.259 | 0.259 | 1.000 | 3 | 0.157 | 0.071 | 0.098 |
|  | 0.1 | 0.3 | 9 | 0.235 | 0.259 | 0.905 | 3 | 0.200 | 0.071 | 0.105 |
| DAG-GNN | N/A | 0.1 | 20 | 0.272 | 0.420 | 0.647 | 7 | 0.212 | 0.206 | 0.209 |
|  | N/A | 0.3 | 19 | 0.256 | 0.333 | 0.778 | 7 | 0.226 | 0.206 | 0.215 |
| CMA | N/A | N/A | 16 | 0.086 | 0.309 | 0.280 | 11 | 0.523 | 0.647 | 0.579 |

Table 15: Results for NOTEARS, DAG-GNN, and CMA benchmarks on the Alzheimer's benchmark dataset - L2 noise level $\epsilon \sim \mathcal{N}(0, 0.8)$. Dashes indicate a cyclic graph was predicted for the corresponding parameters.

|  | Lambda | Threshold | Edges | NHD | BHD | Ratio | TP | Prec. | Recall | F1 |
|---|---|---|---|---|---|---|---|---|---|---|
| NOTEARS | 0.001 | 0 | - | - | - | - | - | - | - | - |
|  | 0.001 | 0.1 | 15 | 0.284 | 0.309 | 0.920 | 4 | 0.159 | 0.100 | 0.121 |
|  | 0.001 | 0.3 | 9 | 0.235 | 0.259 | 0.905 | 3 | 0.2 | 0.071 | 0.105 |
|  | 0.01 | 0 | - | - | - | - | - | - | - | - |
|  | 0.01 | 0.1 | 15 | 0.284 | 0.284 | 1 | 4 | 0.154 | 0.100 | 0.121 |
|  | 0.01 | 0.3 | 9 | 0.235 | 0.284 | 0.826 | 3 | 0.200 | 0.071 | 0.105 |
|  | 0.1 | 0 | - | - | - | - | - | - | - | - |
|  | 0.1 | 0.1 | 11 | 0.259 | 0.309 | 0.84 | 3 | 0.158 | 0.071 | 0.098 |
|  | 0.1 | 0.3 | 8 | 0.270 | 0.270 | 1 | 2 | 0.143 | 0.045 | 0.069 |
| DAG-GNN | N/A | 0.1 | 21 | 0.284 | 0.383 | 0.742 | 7 | 0.200 | 0.206 | 0.203 |
|  | N/A | 0.3 | 17 | 0.235 | 0.383 | 0.613 | 7 | 0.259 | 0.206 | 0.230 |
| CMA | N/A | N/A | 18 | 0.090 | 0.280 | 0.300 | 12 | 0.670 | 0.920 | 0.770 |

Table 16: Results for NOTEARS, DAG-GNN, and CMA benchmarks on the Alzheimer's benchmark dataset - L3 noise level $\epsilon \sim \mathcal{N}(0, 1.2)$. Dashes indicate a cyclic graph was predicted for the corresponding parameters.

|  | Lambda | Threshold | Edges | NHD | BHD | Ratio | TP | Prec. | Recall | F1 |
|---|---|---|---|---|---|---|---|---|---|---|
| NOTEARS | 0.001 | 0 | - | - | - | - | - | - | - | - |
|  | 0.001 | 0.1 | 14 | 0.272 | 0.296 | 0.916 | 4 | 0.167 | 0.100 | 0.125 |
|  | 0.001 | 0.3 | 8 | 0.247 | 0.222 | 1.1 | 2 | 0.143 | 0.045 | 0.069 |
|  | 0.01 | 0 | - | - | - | - | - | - | - | - |
|  | 0.01 | 0.1 | 13 | 0.259 | 0.359 | 1 | 4 | 0.182 | 0.100 | 0.129 |
|  | 0.01 | 0.3 | 8 | 0.247 | 0.296 | 0.833 | 2 | 0.143 | 0.045 | 0.069 |
|  | 0.1 | 0 | - | - | - | - | - | - | - | - |
|  | 0.1 | 0.1 | 11 | 0.259 | 0.309 | 0.84 | 3 | 0.158 | 0.071 | 0.098 |
|  | 0.1 | 0.3 | 7 | 0.235 | 0.235 | 1 | 2 | 0.167 | 0.045 | 0.071 |
| DAG-GNN | N/A | 0.1 | 21 | 0.284 | 0.383 | 0.742 | 7 | 0.200 | 0.206 | 0.203 |
|  | N/A | 0.3 | 18 | 0.247 | 0.346 | 0.714 | 7 | 0.241 | 0.206 | 0.222 |
| CMA | N/A | N/A | 12 | 0.110 | 0.310 | 0.360 | 8 | 0.67 | 0.62 | 0.64 |

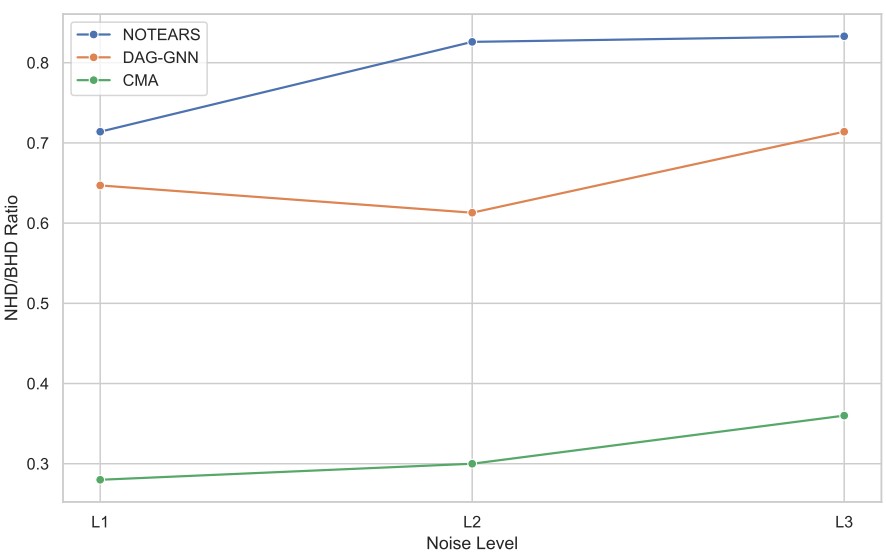

Figure 29: Summary results of the best performing NOTEARS and DAG-GNN configurations for each noise level compared with the CMA. At L1, the additional noise added is $\epsilon \sim \mathcal{N}(0, 0.4)$, at L2 this is $\epsilon \sim \mathcal{N}(0, 0.8)$, and at L3 this is set to $\epsilon \sim \mathcal{N}(0, 1.2)$. A lower Normalised Hamming Distance (NHD) to Baseline Hamming Distance (BHD) ratio is better.

### A.6 Case Study: Alzheimer's Disease Neuroimaging Experiment

#### A.6.1 Experimental setup

**Image pre-processing** Structural T1-weighted MRI scans were collected for all participants and linked to their relevant demographic, disease biomarker, and cognitive assessment variables. The earliest Inversion Recovery Spoiled Gradient echo sequence (SAG IR-SPGR) MRI was accrued for each participant, skull-stripped using the HD-BET brain extraction tool (Isensee et al., 2019), and bias-field corrected with the N4 software package (Tustison et al., 2010). All images were resampled to the size of the MNI ICBM152 brain atlas in the NiLearn software package (Nil) ($197 \times 233 \times 189$) with linear interpolation. The resampled images were subsequently rigidly registered to the atlas using ANTs (Avants et al., 2009). The middle 10 axial slices of each MRI were extracted and their intensity values were normalized by rescaling the minimum and maximum values of each slice to $[0, 255]$. Each 2D image slice is then saved as a PNG file for training. During training, the image slices were uniformly dequantised by the addition of Gaussian noise (Theis et al., 2015). Images were randomly cropped from their original size to $192 \times 192$ and downsampled to $64 \times 64$ during training to prevent overfitting. The slices were centre-cropped during counterfactual image inference.

**Imaging mechanisms** Mechanisms $\mathbf{S}$ were defined in the same way as in the synthetic experiments A.2.2. By decomposing the image mechanism as per 2.1, we model the image as the invertible function

$$H_{\text{img}}(u_{\text{img}}; \text{PA}_{\text{img}}) = [\text{Preprocessing} \circ \text{ConditionalAffine}_\theta(\text{PA}_{\text{img}})](u_{\text{img}}), \qquad (9)$$

where the Preprocessing follows RealNVP (Dinh et al., 2016), and the hyper-network for ConditionalAffine$_\theta(\cdot)$ is the non-invertible mechanism $g_{\text{img}}(z_{\text{img}}; \text{PA}_{\text{img}})$ which is implemented as a decoder that outputs the bias for the ConditionalAffine$_\theta(\cdot)$ transformation with fixed logarithmic-variance of $\log \sigma^2 = -5$. We require an encoder function $e_{\text{img}}(\text{img}; \text{PA}_{\text{img}})$ to generate the latent $z_{\text{img}}$. The images are therefore modelled using a CVAE architecture where both encoder and decoder functions are composed of 5 modules of 3 blocks of $(\text{LeakyReLU}(0.1), \text{BN}_\theta, \text{Conv}_\theta)$, where Conv is a convolutional layer, BN is batch normalisation, and LeakyReLU$(\phi)$ is a leaky rectified linear unit with an angle of negative slope parameter $\phi$. For all other non-imaging mechanisms, we use a similar setup as in the synthetic experiments (Appendix A.2.2).

Binary variables such as biological sex require that we learn the binary probability by sampling from a Bernoulli distribution (female = 1, male = 0). Discrete variables such as APOE4 status and MRI image slice number are sampled from uniform distributions (APOE4 status in $\{0, 1, 2\}$, and minimum to maximum number of slices, respectively), as per Reinhold et al. (2021). All learnable parameters in the flows and the CVAE architecture were optimised by a stochastic variational inference approach to estimate the evidence lower bound (ELBO; estimated using 4 Monte Carlo (MC) samples) using the Adam optimizer (Kingma & Ba, 2015) with learning rates of $10^{-5}$ and $5 \times 10^{-3}$, respectively. For counterfactual inference, 32 MC samples were taken and the inference result was their average. All learnt mechanisms $f_i \in \mathbf{S}$ were fixed during inference and the single world intervention graph (SWIG) formalism was used to produce counterfactuals (Richardson & Robins, 2013). Experiments were parallelized across two NVIDIA RTX 3090 GPUs and one NVIDIA RTX 4090 GPU.

### A.7 Additional Results

Figure 30 illustrates the CMA's reasoning trace for the relationship between APOE4 and P-tau. The model proposes that the TREM2 gene may confound this relationship. We subsequently accrued data for the soluble form of TREM2 (sTREM2) from the Alzheimer's Disease NeuroImaging dataset and trained a DSCM with a causal link between sTREM2 and P-tau. We perform a counterfactual intervention and find a statistically significant relationship between sTREM2 levels and P-tau levels following Welch's ANOVA test of unequal variance ($F = 43.52, p < 0.01$). Figure 31 illustrates counterfactual interventions against the null intervention. As can be seen, there is a positive relationship between increasing sTREM2 levels and P-tau deposition.

We additionally conduct an observational analysis in two stages. First we regress sTREM2 on P-tau levels, and then we add APOE4 as a categorical variable. Results are illustrated in Figure 32 and

Figure 33, respectively. As can be seen, there is a positive association between sTREM2 levels and P-tau ($t = 5.15, p < 0.01$). As expected, patients with $\geq 1$ copies of the APOE4 gene display greater levels of P-tau deposition ($t = 2.76, p = 0.006$), with the greatest levels observable in homozygous present patients (i.e., those who have two copies of the gene). Finally, we conduct an observational analysis by regressing biological sex onto P-tau levels. We can be seen in Figure 34, females have a higher level of P-tau burden ($t = 2.29, p = 0.023$).

APOE4 -> P-tau

{"explanation": "APOE4 is a major genetic risk factor for AD and is known to influence tau pathology. However, it is unlikely that tau influences APOE4 as tau is a downstream effect of the disease process. Other genetic factors, such as TREM2, could potentially confound this relationship. Additionally, environmental factors and lifestyle choices such as diet and exercise could also play a role. The relationship between APOE4 and tau could be mediated or modulated by other factors such as inflammation, oxidative stress, and neuronal loss."}

Figure 30: The CMA reasoning trace for the relationship between the APOE4 gene and tau protein pathology. The model suggests that the TREM2 genotype might confound the causal relationship between APOE4 and tau protein.

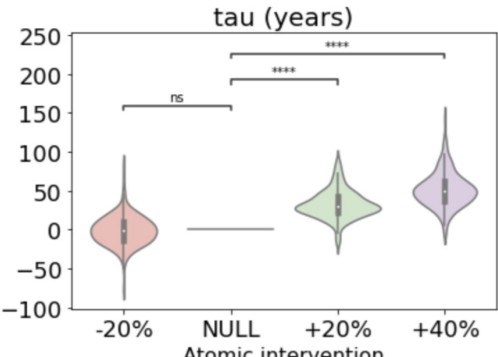

Figure 31: Interventions on STREM2 level and associated counterfactual distributions for P-tau level. The interventions are set as percentages of the original measurement. Statistical annotations represent Welch's t-tests with a Bonferonni correction. **: $p \leq 10^{-2}$, ***: $p \leq 10^{-3}$, ****: $p \leq 10^{-4}$, ns: No statistical significance..

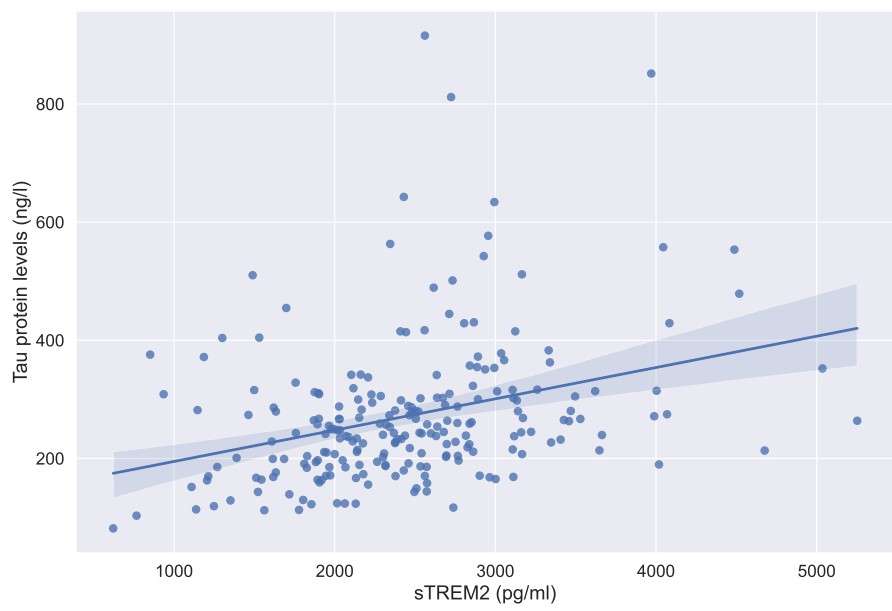

Figure 32: Regression analysis of soluble TREM2 (sTREM2) and phosphrylated tau levels.

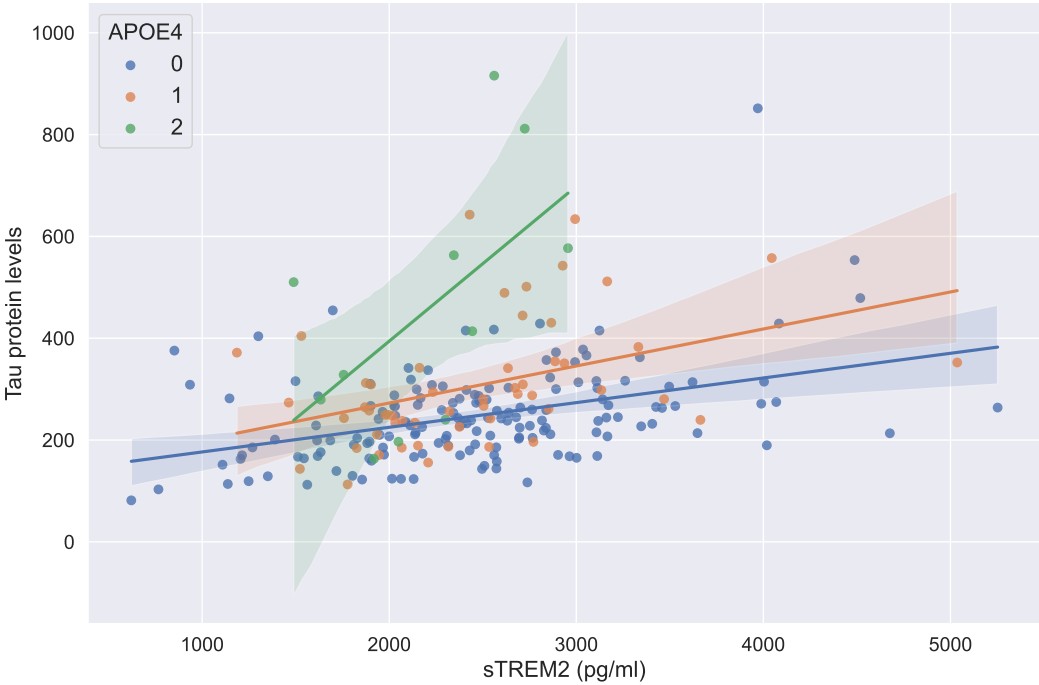

Figure 33: Regression analysis of soluble TREM2 (sTREM2) and phosphrylated tau levels, adjusted for the categorical variables of APOE4 gene.

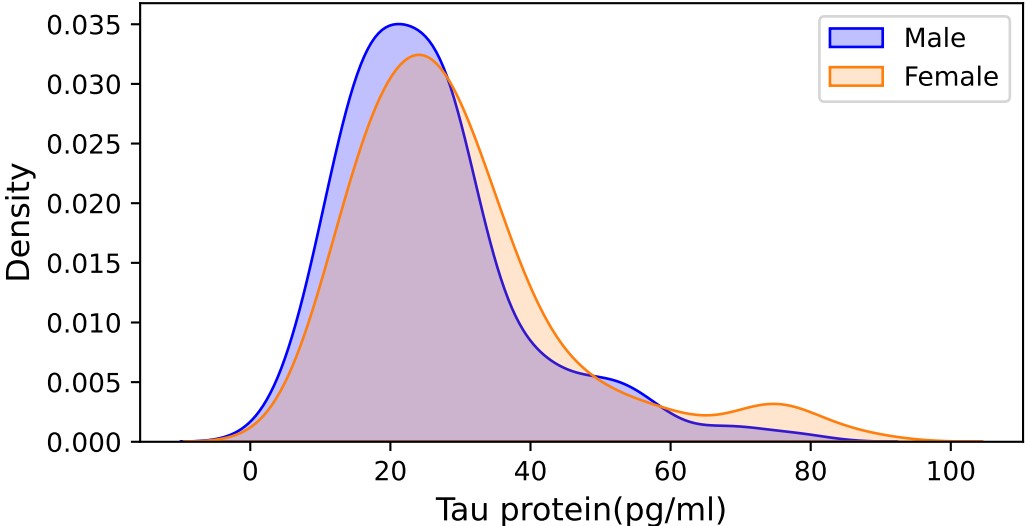

Figure 34: Kernel density estimate plot of the relationship between biological sex and phosphrylated tau levels.

A.7.1 REVIEW OF THE EFFECTS OF sTREM2 AND SEX ON THE TAU PROTEIN

**sTREM2** → **Neuropathic tau protein** Triggering receptor expressed on myeloid cells 2 (TREM2) is an immune receptor in the central nervous system (CNS). It appears to have a positive role in cellular proliferation and survival (Ewers et al., 2019). It has a soluble form (sTREM2), which can be produced from the shedding of TREM2. There are a number of studies which show elevated levels of sTREM2 in AD (Knapskog et al., 2020; Ioannides et al., 2021; Wilson et al., 2020). For example, Suárez-Calvet et al. (2019) demonstrated that tau pathology is associated with an increase in sTREM2 in the cerebrospinal fluid (CSF). However, presence of amyloid protein alone was not associated with an increase in CSF sTREM2. There is also recent evidence of elevated sTREM2 levels in other pathologies including multiple sclerosis (Ioannides et al., 2021) and Parkinson's disease subgroups with increased CSF tau (Wilson et al., 2020).

**Biological sex** → **Neuropathic tau protein** In our analysis, the CMA (fit to the ADNI dataset) proposed that there may be a direct causal link between being biologically female (XX chromosome profile), and neuropathic protein deposition. Recent work by Buckley et al. (2020) aimed to investigate sex differences in tau distribution across multiple brain regions of older adults using Positron Emission Tomography (PET) scanning. They used the Alzheimer's Disease Neuroimaging Initiative (ADNI) and Harvard Aging Brain Study (HABS) datasets. Neuropathic tau protein levels were measured by use of [18F]flortaucipir (FTP), which is a tracer agent that allows in vivo quantification of paired helical filament tau (Ossenkoppele et al., 2018). Their work suggested that women showed statistically significantly higher FTP-signal (greater tau levels) in multiple regions of the cortical mantle ($p < 0.007$). They additionally wanted to assess whether composite FTP signals in Regions of Interest (ROIs) across the brain were associated with a more rapid cognitive decline. In their study, women with higher FTP signals had a borderline significant higher rate of cognitive decline than men ($p = 0.04$). Following this, it was thought that perhaps these sex differences were due to sex hormone profiles. To assess this, Wisch et al. (2021) conducted a cross-sectional neuroimaging study which compared cortical tau deposition (using PET) between cognitively normal males and females. In addition, they also compared preclinical Alzheimer's pathology between females who had and had not used hormone therapy (HT). They observed greater tau deposition in females. However, they also observed decreased tau burden females who were HT users, and highlighted that this relationship should be investigated longitudinally. It was therefore unclear whether hormonal profiles modulated tau protein deposition, or whether this was more directly mediated by genomic/epigenomic markers. deposition. Tau clearance itself is controlled by methylation (Balmik & Chinnathambi, 2021), then (Esteves et al., 2019), and finally ubiquitination (Flach et al., 2014). Yan et al. (2022) demonstrated that the X-linked gene Ubiquitin Specific Peptidase 11 (USP11) removes ubiquitin from tau (the final step required for its clearance), which led to its aggregation in-vivo. USP11 has elevated expression in females, and correlated with tau brain pathology. Whilst this might not yet be the full story, it is an exciting research direction, as the mechanism behind the sexual dimorphism in tau burden is thought to be a foundational event in AD development Yan et al. (2022).

