# OpenReview forum: "Causal Modelling Agents: Causal Graph Discovery through Synergising Metadata- and Data-driven Reasoning"
_ICLR.cc/2024/Conference — ICLR 2024 poster_

### Official Review · Reviewer_pSg2 · 2023-10-28

**Soundness:** 2 fair
**Presentation:** 4 excellent
**Contribution:** 3 good
**Rating:** 6
**Confidence:** 3

**Summary:**

The paper combines the meta-data driven Large Language Models (LLMs) and data-driven Deep Structural Causal Models (DSCMs) to construct a novel framework called Causal Modeling Agent for causal discovery. The framework leverages the LLMs' state-of-the-art capability to capture domain knowledge to discover the causal relationship in DSCMs. The framework is tested against a number of benchmarks on the real-world task of modeling the clinical and radiological phenotype of Alzheimer's Disease (AD), which has a ground-truth causal relationship between the vertices. The experimental results suggest that the CMA outperforms purely data-driven and metadata-driven benchmarks. New insights into the causal relationship among biomarkers of AD have also been obtained by CMA.

**Strengths:**

1. The idea to combine LLM and SCM is interesting and novel.
2. The experimental results are encouraging.
3. New insights on the causal relationship between biomarkers have been obtained.

**Weaknesses:**

The contribution would be stronger if further evidence from experimental or observational data can be provided for the discovered causal relationships with the CMA.

**Questions:**

Can the authors provide further evidence from experimental or observational data for the discovered causal relationships with the CMA?

---

> ### Author Response · Authors · 2023-11-17
> **Response**
>
> We thank the reviewer for their positive review of our work. We are glad they find our experiments encouraging and the combination of LLMs and (deep) structural causal modelling to be interesting and novel. To address their central query, we have updated the paper to add additional analyses based on our observational datasets in **Appendix A.3.4** and **Appendix A.7**.
>
>  In addition, we have also updated the paper to include a new **Appendix A.7.1** which highlights additional experimental/observational evidence from the literature which contextualises and/or corroborates the causal relationships discovered by the CMA in the real-world AD experiment.

---

> > ### Comment · Reviewer_pSg2 · 2023-11-21
> >
> > The statistical correlations presented in Appendix A.3.4 do suggest the causal relationship between LW and HFLX. An experiment that intervenes LW may help reveal such causality.

---

> ### Author Response · Authors · 2023-11-21
> **Response to comment**
>
> >The statistical correlations presented in Appendix A.3.4 do suggest the causal relationship between LW and HFLX. An experiment that intervenes LW may help reveal such causality.
>
>
>
> Thank you for your response. We had previously conducted an experiment which intervenes on the LW variable in **Appendix A.3.4**. To maximise clarity, we have amended the manuscript to illustrate these results in two separate figures (instead of the single **Figure 13**). For the experimental/counterfactual intervention, please also see the [anonymised figure](https://i.imgur.com/3T1eMcb.png).
> Indeed, for each of our discovered novel relationships, we have included:
>
> 1.  A regression (observational) analysis, as requested, to assess for possible statistical associations between the variables.
>
> 2.  An experiment which intervenes on the parent variable (counterfactual intervention) to assess a given relationship from a causal perspective.
>
> 3.  Experimental/observational evidence from the literature to contextualise and/or corroborate our findings.
>
>
> Please find references for these analyses below:
>
> -   LW → HFLX:
>     1.  Observational analysis: **Figure 13, Appendix A.3.4**.
>     2.  Counterfactual intervention: **Figure 14, Appendix A.3.4**.
>     3.  Literature: **Section 4.2**, main manuscript.
>
>
> -   Sex → P-tau:
>      1.  Observational analysis: **Figure 34, Appendix A.7**
>     2.  Counterfactual intervention: **Figure 3B**, main manuscript.
>     3.  Literature: **Section 4.3**, main manuscript.
>     4.  Additional literature: **Appendix A.7.1**
>
>
> -   sTREM2 → P-tau:
>     1.  Observational analysis: **Figures 32 and 33, Appendix A.7**
>     2.  Counterfactual intervention: **Figure 31, Appendix A.7**
>     3.  Literature: **Section 4.3**, main manuscript.
>     4.  Additional Literature: **Appendix A.7.1**
>
> ___
>
> Thank you once more for your time. Please let us know if there are any additional points you’d like to discuss, and if not, we hope that the reviewer might consider an update.

---

> > ### Comment · Reviewer_pSg2 · 2023-11-23
> >
> > What is the respective treatment and control groups for the experiment that generates Figures 3B, 14, and 31?

---

> > > ### Author Response · Authors · 2023-11-23
> > > **Reponse to comment**
> > >
> > > Thank you for your question. These experiments represent counterfactual interventions, in the sense that they satisfy ‘Level 3’ interventions as per Pearl’s causal ladder [1], which we briefly sketch below:
> > >
> > > -   Level 1 - Association: Reasoning over passively observed data. This level deals with correlations in the data and questions of the type *‘What are the odds I observed…?’*. This relates to purely marginal, joint, and conditional probabilities [1,2].
> > >
> > > -   Level 2 - Intervention: Concerns interactions with the environment, and requires knowledge beyond observations; it relies on **structural assumptions**  about the underlying data-generating process. At this level, we can ask questions of the nature *‘What happens if I do…?'*.
> > >
> > > -   Level 3 - Counterfactual intervention: Concerns retrospective hypothetical scenarios, leveraging functional models of the generative process to imagine alternative outcomes for a specific data point. Here, we can answer questions of the nature *‘What if I had done X instead of Y?'*.
> > >
> > >
> > > As an example let’s assess the discovered relationship between biological sex and phosphorylated tau protein (Sex → P-tau). It is not possible to experimentally alter the biological sex of a patient. However, the output of the CMA (a causal graph encoded as Deep SCM) fulfils all three runs of Pearl’s causal hierarchy, meaning we can perform counterfactual interventions of the nature ‘If this patient had been born biologically female (instead of male), what would their P-tau levels be?’. We can then assess the relationship between the original distribution, and our new (counterfactual) distribution. We find that under our model (and data), biological females have higher levels of the P-tau protein. Whilst a number of associative (Level 1) studies exist to describe this relationship, none perform an analysis from a causal perspective (Level 3). This means we do not have a classical ‘treatment’ and ‘control’ group per se, instead, we have an observed variable and a counterfactual variable. The only alternative to validating this type of relationship is to perform genomic/epigenomic studies in the wet lab setting for ex-vivo and/or in-vivo tissue. Indeed, Yan et al. [3] have started to isolate the mechanism of action behind this relationship (through the USP11 X-chromosome-linked gene [**Appendix A.7.1**]). Whilst we demonstrate this relationship by counterfactual intervention (by intervening on the sex variable and assessing downstream P-tau levels), performing further wet-lab studies is outside the scope of the current work.
> > >
> > > Thank you once more for your time and consideration.
> > >
> > >
> > >
> > > References:
> > >
> > > *1.  Pearl, J. (2019). The seven tools of causal inference, with reflections on machine learning. Communications of the ACM, 62(3), 54-60.
> > > 2.  Pawlowski, N., Coelho de Castro, D., & Glocker, B. (2020). Deep structural causal models for tractable counterfactual inference. Advances in Neural Information Processing Systems, 33, 857-869.
> > > 3.  Yan, Y., Wang, X., Chaput, D., Shin, M. K., Koh, Y., Gan, L., ... & Kang, D. E. (2022). X-linked ubiquitin-specific peptidase 11 increases tauopathy vulnerability in women. Cell, 185(21), 3913-3930.*.

---

### Official Review · Reviewer_CmY7 · 2023-11-01

**Soundness:** 2 fair
**Presentation:** 2 fair
**Contribution:** 2 fair
**Rating:** 3
**Confidence:** 2

**Summary:**

In this paper, the authors devised a causal discovery algorithm that utilizes LLM’s ability on causal reasoning using meta-data. In particular, they proposed Causal Modeling Agent (CMA), which iteratively updates a causal graph through: i) asking LLM for updates on current prediction of edges with previous graph update information; and 2) fitting a model constrained over the intermediately constructed causal graph (using deep learning to model causal mechanism for each variable). Through experiments on benchmark datasets (e.g., Kıcıman et al. (2023)) and a case study of Alzheimer's disease, they empirically demonstrated a potential of their framework outperforming some of causal discovery algorithms and LLMs.

**Strengths:**

Paper is overall written concisely due to multiple modules involved in the framework. The idea of encoding intermediate results in a JSON format and feeding them into an LLM seems clever.

**Weaknesses:**

- The use of LLM to tweak intermediate results seems clever but it is hard to assess its technical contribution.
- It is unknown how LLM is doing with respect to its memory. Does LLM always try to update edges in order to maximize data fitting? If the data fitting is based on the currently predicted causal graphs, how can it improve its causal graph? It does not work like an EM algorithm. Does LLM ‘regret’ its previous decision if fitting becomes worse? Considering developing a causal discovery algorithm that is based on local search (incrementally updating causal graph based on its likelihood), how would you compare their learning trajectories?
- Use of data only to fit the intermediate graph seems not using available dataset in full. Such as conditional independence and other information is all unused.
- LLM’s stochastic nature is ignored. LLM may answer differently for the same question.
- It is essential to thoroughly examine the behavior of LLM. How does it adjust the result based on its belief (GPT-4 etc) and intermediate results passed. There are more questions remained than answered.

**Questions:**

The word “metadata” is somewhat used in a mixed manner between domain knowledge already encoded in LLM and memory passed through JSON format. It should be more formally defined.

Results
Given that cases with no edges outnumber those with edges, not predicting edges may lead to an increase in accuracy. Thus, a qualitative analysis is necessary since not predicting edges might lead to an increase in the score. Other metrics such as TPR or FDR can be reported.

Novelty
Given the abundance of similar papers (Long, S., Piché, A., Zantedeschi, V., Schuster, T., & Drouin, A. (2023). Causal discovery with language models as imperfect experts. arXiv preprint arXiv:2307.02390., Ban, T., Chen, L., Wang, X., & Chen, H. (2023). From query tools to causal architects: Harnessing large language models for advanced causal discovery from data. arXiv preprint arXiv:2306.16902.) in the field, the contribution is not clear.

I noticed discrepancies between what was mentioned and the results such as the performance of gpt-4 in the table 7 in Kıcıman et al. (2023). For example, NHD of GPT 4 in Kıcıman et al. (2023) was reported as 0.22 but you reported 0.35 for GPT 4 in the table 2 in your paper.

---

> ### Author Response · Authors · 2023-11-17
> **Response 1/3**
>
> We thank the reviewer for their insight and questions.
>
> We hope to have addressed the reviewer's comments and, if so, they would consider updating their score. We’d be happy to engage in further discussions.
>
> ___
>
> > -   The use of LLM to tweak intermediate results seems clever but it is hard to assess its technical contribution.
> > -   Novelty Given the abundance of similar papers (Long, S., Piché, A., Zantedeschi, V., Schuster, T., & Drouin, A. (2023). Causal discovery with language models as imperfect experts. arXiv preprint arXiv:2307.02390., Ban, T., Chen, L., Wang, X., & Chen, H. (2023). From query tools to causal architects: Harnessing large language models for advanced causal discovery from data. arXiv preprint arXiv:2306.16902.) in the field, the contribution is not clear.
>
> Thank you for noting these important related works. In contrast to previous work, the CMA is the first framework to use LLMs as priors, critics during an iterative learning procedure, and as post-processors. We demonstrate the utility of this approach as per our results in **Section 4.2**, and in **Section 4.3**, we identify novel causal structures in the challenging real-world task of jointly modelling the radiological and clinical phenotype of Alzheimer’s disease. An additional novel aspect of our work is that our framework is the first causal discovery approach to output a multi-modal structural causal model (that is, we generalise and extend the DSCM framework), which enables reasoning over multi-modal data (e.g., for computing counterfactuals in the imaging space, as can be seen in Figure 3, Panel C of the main manuscript).
>
> The method developed by Long et al.[1] assumes knowledge of the Markov Equivalence Class (MEC) containing the true Directed Acyclic Graph (DAG). This is a strong assumption for real-world applications as it presumes optimal outputs from causal discovery algorithms. Their work shows promising results in substituting an LLM in lieu of a human expert in this setting, but their approach serves as a post-processing step, reliant on a priori knowledge of a MEC which contains the true causal graph. In contrast, the CMA does not require any prior assumptions about the graph's structure.
>
> Ban et al.[2] use an LLM's output as a prior for causal discovery algorithms, demonstrating its effectiveness with synthetic datasets, an approach also explored in the ‘LMPriors’ paper by Choi et al.[3]. However, this method lacks a feedback mechanism for hypothesis refinement, potentially limiting novel causal link detection by ignoring signals from the data. An example of this is the 'biological sex' -> 'phosphorylated tau protein' link identified by the CMA in **Section 4.3**, but overlooked by domain experts. We additionally note that in contrast to Ban et al.[2], we consider a real-world multi-modal dataset as one of our experiments, demonstrating our framework’s empirical utility for more complex and varied data scenarios beyond synthetic cases.
>
> We have revised the text in the Related works section of the paper to better highlight these differences.
>
> ___
>
> > It is unknown how LLM is doing with respect to its memory. Does LLM always try to update edges in order to maximize data fitting? If the data fitting is based on the currently predicted causal graphs, how can it improve its causal graph? It does not work like an EM algorithm. Does LLM ‘regret’ its previous decision if fitting becomes worse? Considering developing a causal discovery algorithm that is based on local search (incrementally updating causal graph based on its likelihood), how would you compare their learning trajectories?
>
>  As you correctly point out, the data fitting step at a given iteration is based on the currently predicted causal graph. However, the CMA uses both the current data fit, as well as the data fit from the previous iteration to produce a memory. As we show in the ‘post-processing’ paragraph of **Section 3**: $\mu_t=\textsf{LLM}{\mu}(\mathcal{G}^s_t,\mathcal{G}^s_{t-1},F_t,F_{t-1})$, where for iteration $t$, $\mathcal{G}_t$ is the causal graph and $F_t$ is the metric of fit.
>
>
>
>  The prompting strategy and procedure for memory generation is further outlined in ‘Listing 3: System prompt for memory generation’, and ‘Algorithm 5: Post-processing and memory generation’ in **Appendix A.1.5**. Exactly as you intuited, the LLM is therefore prompt-engineered to ‘regret’ its previous decisions if fitting degrades, and to select an action or set of actions to improve fit. The LLM constrains the super-exponential search space of potential causal graphs, which a naive local-search-based algorithm would need to explore.

---

> ### Author Response · Authors · 2023-11-17
> **Response 2/3**
>
> >  Use of data only to fit the intermediate graph seems not using available dataset in full. Such as conditional independence and other information is all unused.
>
> We believe the reviewer means that there are conditional independence statements which exist within the dataset that are not being utilised. If this is the case, then in fact we do utilise conditional independence information from the data. The metadata-based modules of the CMA produce a hypothesis (DAG) which is then encoded as a Deep Structural Causal Model (DSCM). Conditional independence statements are encoded within the structure of the DSCM at the current iteration and, as shown in **Section 4.1**, this has an effect on data likelihood under the model (up to Markov Equivalence; see **Appendix A.2.3** for more details).
>
> ___
> >LLM’s stochastic nature is ignored. LLM may answer differently for the same question.
>
>
> Thank you for this very important insight. We very much agree that LLMs do not always answer the same question deterministically, however we take account of this by querying the LLMs multiple times to mitigate for potential stochasticity in the outputs. We also attempt to account for failure modes in LLM reasoning. For example, Berglund et al. [4] identified a 'reversal curse' in LLMs, where they fail in logical deduction based on question order, not always generalising from 'A is B' to 'B is A'. To additionally address this layer of variability, queries are also made in both causal directions for local phase amendments. This process is described in more detail in **Appendix A.1.3**.
>
>
> ___
> > It is essential to thoroughly examine the behavior of LLM. How does it adjust the result based on its belief (GPT-4 etc) and intermediate results passed. There are more questions remained than answered.
>
> Thank you once more for this important point. We strongly agree that investigating the behavioural patterns of LLMs is important to better understand their outputs, and indeed had performed a behavioural analysis in **Appendix A.2.4**: ‘Additional Results II - LLM Behavioural Patterns’. Here, we investigate behavioural patterns for three types of relationships: 1) Relationships which might plausibly exist in either causal direction (i.e., A -> B or B -> A), 2) Relationships which should only exist in a single direction (i.e., A -> B, but not B -> A), and, 3) Variables which should have no functional causal relationship between them. Furthermore, for each of these types of relationship, we assess LLM behaviour by first assuming the non-existence of the relationship, and then its existence in the A -> B direction, and then in the B -> A direction (**Appendix A.2.4; Figure 5**).
> We additionally investigated how these behavioural patterns change with the use of a Retrieval Augmented Generation (RAG) pattern [5], and find that in-context learning [6] strongly enforces causal relationships given the retrieved context (**Appendix A.2.4; Figure 6**).
>
> We have amended the manuscript to more explicitly reference this analysis.
>
>
>
> ___
> >The word “metadata” is somewhat used in a mixed manner between domain knowledge already encoded in LLM and memory passed through JSON format. It should be more formally defined.
>
> We appreciate this suggestion. We’ve now made a number of amendments to keep the distinction between both concepts clear.
>
> ___
> >Results Given that cases with no edges outnumber those with edges, not predicting edges may lead to an increase in accuracy. Thus, a qualitative analysis is necessary since not predicting edges might lead to an increase in the score. Other metrics such as TPR or FDR can be reported.
>
> Thank you for this point. Exactly as you point out, the Normalised Hamming Distance (NHD) metric is contingent on the number of edges returned by a causal discovery algorithm. Indeed, predicting no edges at all may have a lower (better) NHD than predicting a number of edges which are incorrect. We therefore follow the approach taken by Kiciman et al. [7] in comparing the ratio of the NHD to a ‘floor’ baseline (Baseline Hamming Distance [BHD]) which outputs the same number of edges but all of them are incorrect. The NHD/BHD ratio is then the multiple by which the discovery algorithm is better than the worst baseline.
> In addition to accounting for the NHD’s contingency on the number of edges predicted by using the NHD/BHD ratio, we believe that we perform sensible qualitative analyses of the results both in the main manuscript and **Appendix A.3.3, A.4.3, A.5.1, and A.5.3**. However, please let us know whether there are any specific types of additional analyses you’d like to see. Finally, in the above mentioned sections, we additionally report the True Positive Rates (TPR), the precision and recall of the algorithms, as well as the F1 score.

---

> ### Author Response · Authors · 2023-11-17
> **Response 3/3**
>
> **References**:
> *1. (Long, S., Piché, A., Zantedeschi, V., Schuster, T., & Drouin, A. (2023). Causal discovery with language models as imperfect experts. arXiv preprint arXiv:2307.02390.
> 2. Ban, T., Chen, L., Wang, X., & Chen, H. (2023). From query tools to causal architects: Harnessing large language models for advanced causal discovery from data. arXiv preprint arXiv:2306.16902.)
> 3. Choi, K., Cundy, C., Srivastava, S., & Ermon, S. (2022). LMPriors: Pre-Trained Language Models as Task-Specific Priors. arXiv preprint arXiv:2210.12530.
>  4. Berglund, Lukas, et al. "The Reversal Curse: LLMs trained on" A is B" fail to learn" B is A"." arXiv preprint arXiv:2309.12288 (2023).
>  5. Lewis, Patrick, et al. "Retrieval-augmented generation for knowledge-intensive nlp tasks." Advances in Neural Information Processing Systems 33 (2020): 9459-9474.
> 6. Brown, T., Mann, B., Ryder, N., Subbiah, M., Kaplan, J. D., Dhariwal, P., ... & Amodei, D. (2020). Language models are few-shot learners. Advances in neural information processing systems, 33, 1877-1901.
> 7. Kıcıman, E., Ness, R., Sharma, A., & Tan, C. (2023). Causal reasoning and large language models: Opening a new frontier for causality. arXiv preprint arXiv:2305.00050.**

---

> > ### Comment · Reviewer_CmY7 · 2023-11-21
> > **follow-up**
> >
> > Section 4.1
> > DAG 3 is Markov equivalent to DAG 1 (and ground truth) (same skeleton, the same v-structure). Please explain the statistically significant difference in model fitting results. This conflicts the authors’ statements: “The model that aligns most closely with the true data-generating process produces the highest data likelihood. As expected, we find that this is only valid up to the Markov equivalence class of the ground-truth DAG”. (definition in Page 25 seems correct.)
> >
> > Section 4.2
> > Why did you choose NOTEARS or DAG-GNN, which are relatively new causal discovery algorithms but it is unclear why classical approaches (PC, MMHC, or GES like algorithms) are not employed. For example, NOTEARS assumes linearity.
> > Further, Arctic-Sea ice dataset’s ground truth is somewhat controversial given that the data does not match ground truth as demonstrated in the paper, which is based on meta-analysis. For LW → HFLX, what about other algorithms? You mentioned that it is not in the ground truth graph but it does not mean that other algorithms couldn’t figure out. If it is not detectable by LLM nor data, then shouldn’t it be considered hallucination, or sort of, even though the authors can argue with the results in Bates 2012? (BTW, Figure 12 seems that LLM can reason about the edge? ) Hence, this ‘anecdote’ cannot be used as an evidence that CMA is superior to LLM- or pure data-based approaches.
> > It is also awkward to me to see that fitting a linear regression line to argue about “observational analysis” and “The output of the model trained by the CMA is shown in Figure 13, which illustrates that by counterfactual inference, an increase in LW leads to an increased measurement of HFLX.” I am not sure how counterfactual inference (level 3 inference based on Pear’s hierarchy) is done here. Typically this involves unobserved variables. Isn’t this fitting a model and change the value? Shouldn’t it be considered intervention? (level 2 inference)? Also how the change of HFLX represent direct cause from LW? Can’t it be indirect causal relationship?
> >
> > Section 4.3 :
> > I am wondering whether the results properly evaluate the quality of CMA. This seems more relevant to the applicability/utility of Deep SCM itself rather than CMA itself.
> > Overall, theoretical justification for the method is a bit lacking and empirical evaluation seems insufficient to understand how memory, LLM, data fitting work in harmony to create the final results.
> >
> >
> >
> > Additional comments on the benchmark results.
> >
> > Although you've mentioned following the approach by Kiciman et al, I'm struggling to find a justification for the discrepancies in comparing the baseline of Kiciman et al. with your framework.
> > Firstly, if the intention is to make a direct comparison with Kiciman et al., it would be advisable to cite the results as they are. This would help avoid any perception of arbitrary selection, editing, or cherry-picking. It seems there may be a discrepancy in the reported increase in the BHD of the data-driven causal discovery algorithms compared to what was claimed in your paper. While you utilized Kiciman et al.'s result for the data-driven causal discovery algorithm, the adjustments made to GPT-4 and GPT-3.5 Turbo results were not explicitly addressed in your paper.
> > Secondly, there appears to be a challenge in reproducing the results of Kiciman et al. According to their findings, there were 46 edges, and the NHD was 0.22. In contrast, you mentioned no correct edges when selecting 16 edges, indicating a significant difference in results. I suggest attempting to reproduce the NHD score as closely as possible to the original result of Kiciman et al., even if there's a variation in the number of edges.
> >
> >
> > (I am trying to genuinly understand the method better given that other reviewers positively assess the paper. I will properly raise the score if I find out the method is novel and is properly evaluated.)

---

> > > ### Author Response · Authors · 2023-11-21
> > > **Response  to follow-up 1/3**
> > >
> > > Thank you for your response, we hope to address your remaining questions below.
> > >
> > > >Section 4.1 DAG 3 is Markov equivalent to DAG 1 (and ground truth) (same skeleton, the same v-structure). Please explain the statistically significant difference in model fitting results. This conflicts the authors’ statements: “The model that aligns [...]
> > >
> > >
> > > DAG 3 is a graph which has a bi-directional edge between variables AV45 and phosphrylated-tau (P-tau) (i.e., AV45 <-> P-tau). This represents a chain component from a chain graph (D. Barber, Chap. 4. [1]). This graph therefore does not have the same v-structure as DAG 1, and is therefore not Markov equivalent as per the definition in Peters et al., Chap. 6 [2]. We have amended the figure caption to highlight this difference.
> > >
> > >  ___
> > >
> > > > Section 4.2 Why did you choose NOTEARS or DAG-GNN, which are relatively new causal discovery algorithms but it is unclear why classical approaches (PC, MMHC, or GES like algorithms) are not employed. For example, NOTEARS assumes linearity.
> > >
> > > We have now run additional experiments for the PC, MMHC, GES, and LiNGAM algorithms for all benchmark tasks. Results for these experiments can now be found in **Appendix A.3.3, A.4.3, and A.5.3**, respectively. For ease, please see [anonymised screenshot 1](https://i.imgur.com/3brjGdk.jpg), [anonymised screenshot 2](https://i.imgur.com/fi0jvdp.jpg), and [anonymised screenshot 3](https://i.imgur.com/MwLbIQC.jpg). Classical algorithms perform similarly to other data-driven algorithms, however are broadly outperformed by metadata-driven techniques, and the CMA.
> > > In prior work, the NOTEARS algorithm was compared against Greedy Equivalent Search (GES) [3] (specifically, the ‘Fast Greedy Search’ [FGS] implementation of this algorithm) [4], the PC algorithm [5], and LiNGAM [6] - significantly outperforming them across several causal structure learning tasks. The DAG-GNN algorithm later matched or outperformed NOTEARS across a similar set of tasks [7]. We had therefore selected these more recent algorithms as they represent state-of-the-art data-driven discovery algorithms.

---

> > > ### Author Response · Authors · 2023-11-21
> > > **Response  to follow-up 2/3**
> > >
> > > ___
> > > > Further, Arctic-Sea ice dataset’s ground truth [...]
> > >
> > > The CMA outperformed both data-driven and metadata-driven algorithms alone for the arctic sea ice dataset, which we assessed through the NHD, NHD/BHD ratio, TPR, precision, recall, and F1 score, as can be seen in **Tables 6 and 7** in **Appendix A.3.4**. Similarly, the CMA outperformed the other approaches for the two other (purely synthetic) benchmarks, where the data-generating process is exactly faithful to the assumed causal graph. Having said that, in the arctic sea ice dataset (as with any other causal structure learning benchmark), the ground truth was decided by first gathering domain knowledge from field experts, and then constructing a DAG to encode this knowledge. In all such cases, there is a (strong) assumption that domain experts know the ground-truth graph. However, as noted by others, this assumption may not hold because: 1) the possible number of edges grows super-linearly with the number of variables, making it difficult for human experts to systematically consider potential causal relationships, and 2) experts may not be able to precisely provide the causal link between two variables given their knowledge [8]. Real-world data may contain causal relationships not encoded in the assumed true graph. We believe that it may be of interest to detect potentially novel causal relationships in this context and use the LW → HFLX relationship as an example.
> > >
> > > As you correctly point out from **Figure 12**, the metadata-based module of the CMA produces a reasoning trace of the potential relationship LW → HFLX. Of course, it is possible that other approaches may suggest such a link. However, if this were a metadata-based approach alone for example, we would have no specific way of validating and/or grounding this relationship in real-world data, however, the CMA enables us to do so by constructing a DSCM to fit this hypothesis. Once training has concluded, we can perform additional counterfactual queries to further investigate the relationship, as can be shown in **Figure 14**. Indeed, this analysis demonstrates a level 3 counterfactual inference as per Pearl’s hierarchy [9]. Specifically, we abduct exogenous noise, perform an (atomic) intervention of interest, and then perform a forward pass of the model to produce a counterfactual prediction. We have amended the manuscript to split our previous **Figure 13** to maximise clarity on this, because previously that figure included a linear model to simply demonstrate a possible statistical association between LW and HFLX, and, separately, Kernel Density Estimate plots of the base and counterfactual distributions obtained through use of a separate model (the Deep SCM output by our CMA) to investigate the relationship from a causal perspective.  By counterfactual analysis, our model proposes that increasing LW leads to an increased measurement of HFLX, which we contextualise using the current literature on the topic, and argue this was a missed edge from the assumed causal graph.
> > >
> > > We demonstrate another illustrative example in Section 4.3, where we perform causal discovery over a real-world, multimodal, clinical and radiological dataset (ADNI). The metadata-based modules of the CMA propose a link between biological sex and phosphorylated tau (P-tau) protein, which was not proposed by our domain experts (five professors of neurology/neuro-radiology), but this relationship finds support in the ADNI dataset, and counterfactual analyses demonstrated that biological females have greater levels of P-tau. Indeed, more recently, Yan et al. [10] have started to provide a mechanistic understanding of this relationship.  Naturally, any discovered relationship(s) might potentially be further confounded or mediated by external variables that we have no direct access to. However, we note that it is possible for the metadata-based modules to suggest potential confounders or modulators for inclusion in a future causal graph. Indeed, we give a separate example of this in Section 4.3, where the CMA proposes that the gene TREM2 may confound the relationship between the APOE4 gene and P-tau. After including TREM2 in the model, we conduct counterfactual analyses which suggest a data-driven link between TREM2 and P-tau. There is at present emerging wet-lab evidence of this relationship, though the exact mechanism is not yet fully understood. We have however included a discussion of the related literature for these relationships in **Appendix A.7.1**.

---

> ### Author Response · Authors · 2023-11-21
> **Response to follow-up 3/3**
>
> Whilst the Deep SCM framework can be used to model tabular and imaging data, it is **not** a causal discovery algorithm. One must 1) know the causal graph a priori and, 2) manually program the graph’s structure as a model. We have therefore generalised, extended, and subsumed this framework into a broader causal discovery algorithm which utilises data-driven and metadata-driven reasoning to produce causal graphs. We could, for example, perform exhaustive fitting of DSCMs, whereby we enumerate all possible DAGs for a given set of variables and fit each one. However, a full ablation of this nature would not be possible for anything but trivial graphs due to the super-exponential search space (Peters el a. [2], Appendix B: ‘Causal Orderings and Adjacency Matrices’). In our pilot study, we found the concept of ‘memory’ to be crucial, as there would be no possible alternative to providing a feedback signal to the metadata-based modules, and therefore each iteration would be entirely independent of previous amendments, which obviates the need for any numerical data (however as we demonstrate, signal from data leads to improved performance for causal discovery). We may be able to improve performance even further by using, e.g., alternative prompt tuning approaches to the memory structures, the user prompt, and the system prompts of the LLMs, but consider this important future work.
>
> ___
>
> > Additional comments on the benchmark results. [...]
>
> Thank you for this point, and well noticed. We re-ran all experiments in this work, but simply had not moved this information from our Appendix to the main manuscript for the data-driven approaches for the arctic sea ice benchmark. This has now been rectified. To replicate the metadata-based benchmarks for gpt-3.5-turbo and gpt-4, we used the same ‘single prompt’ approach as in the kiciman paper (this is described in kiciman et al. [8] in their Appendix A.1, Table 14). They augment this prompt so that it can be used for ‘full graph discovery’ (this modification is described in kiciman et al. [8]; Section 3.2.1, page 13). For convenience, we provide the prompt template in full in **Appendix A.3.2, Table 5** of our own work. We believe the slightly different results could be down to a couple of factors. First, it appears that gpt-4’s performance on a multitude of tasks can change over time, as demonstrated by Chen et al. [11], which may partly explain these differences. Second, due to a lack of description as to how they handled errors, we implemented our own error-handling strategy. Namely, the model is meant to produce a single letter in {A,B,C} which reflects one of three options:
>
>
>
>
> - A: Changing $A$ causes a change in $B$
>
> - B: Changing $B$ causes a change in $A$
>
> - C: No causal relationship exists
>
>
>
>
>
> Occasionally the model might output ‘C/A’. This would be handled as an error in our work, and no edge would be proposed in these cases. A more detailed description of the experimental setup can be found in our **Appendix A.3.2**.
>
>
> We have updated **Appendix A.3.3** to include the points above in our interpretation of the results for the arctic sea ice dataset.
> ___
> Thank you for your continued engagement with our work, and we hope to have addressed the majority of your concerns.

---

> > ### Author Response · Authors · 2023-11-21
> > **References**
> >
> > **References**
> >
> >  *1. Barber, D. (2012). Bayesian reasoning and machine learning.
> >     Cambridge University Press.
> > 2.  Peters, J., Janzing, D., & Schölkopf, B. (2017). Elements of causal
> >     inference: foundations and learning algorithms (p. 288). The MIT
> >     Press.
> >  3. Chickering, D. M. (2002). Optimal structure identification with
> >     greedy search. Journal of machine learning research, 3(Nov),
> >     507-554. Ramsey, J., Glymour, M., Sanchez-Romero, R., & Glymour, C.
> >     (2017).
> >  4. A million variables and more: the fast greedy equivalence
> >     search algorithm for learning high-dimensional graphical causal
> >     models, with an application to functional magnetic resonance images.
> >     International journal of data science and analytics, 3, 121-129.
> >  5. Spirtes, P., Glymour, C. N., & Scheines, R. (2000). Causation,
> >     prediction, and search. MIT press.
> > 6. Shimizu, S., Hoyer, P. O., Hyvärinen, A., Kerminen, A., & Jordan, M.
> > (2006). A linear non-Gaussian acyclic model for causal discovery.
> > Journal of Machine Learning Research, 7(10).
> > 7. Yu, Y., Chen, J., Gao, T., & Yu, M. (2019, May). DAG-GNN: DAG
> > structure learning with graph neural networks. In International
> > Conference on Machine Learning (pp. 7154-7163). PMLR.
> > 8. Kıcıman, E., Ness, R., Sharma, A., & Tan, C. (2023). Causal
> > reasoning and large language models: Opening a new frontier for
> > causality. arXiv preprint arXiv:2305.00050.
> > 9. Pawlowski, N., Coelho de Castro, D., & Glocker, B. (2020). Deep
> > structural causal models for tractable counterfactual inference.
> > Advances in Neural Information Processing Systems, 33, 857-869.
> > 10. Yan, Y., Wang, X., Chaput, D., Shin, M. K., Koh, Y., Gan, L., ... &
> > Kang, D. E. (2022). X-linked ubiquitin-specific peptidase 11
> > increases tauopathy vulnerability in women. Cell, 185(21),
> > 3913-3930.
> > 11. Chen, L., Zaharia, M., & Zou, J. (2023). How is ChatGPT's behavior
> > changing over time?. arXiv preprint arXiv:2307.09009.*

---

> ### Comment · Reviewer_CmY7 · 2023-11-21
> **follow-up**
>
> Okay, I missed the bidirectional edge in the graph. My bad.
> I was thinking 'directed acyclic graph' without an unobserved confounder since the paper keep talks about DAG. Here, chain graph is the type of graph (more like syntax) and the semantics for the bidirected edge is not clear. Do you mean it to be an unobserved confounder?
>
> Given that all three are connected (both ground truth and DAG3), both graph encodes no conditional independence, and, hence, the same power to represent P(Age, AV45, P-tau). What am I missing here?
>
> Thanks for the additional experiments and detailed explanations on my previous questions and comments. I greatly appreciate devoting your time on running additional experiments that could certainly support your claims in the paper.
>
> (I will try to discuss with other reviewers after the author/reviewer discussion period, then reflect all those comments from the authors and other reviewers in my official review and rating.)

---

### Official Review · Reviewer_GSYx · 2023-11-01

**Soundness:** 3 good
**Presentation:** 3 good
**Contribution:** 3 good
**Rating:** 8
**Confidence:** 2

**Summary:**

This paper proposes to integrate large language models into causal discovery algorithms for multi-modal data and shows superiority of this model is shown in a number of examples.

**Strengths:**

The model architecture is convincing and the extensive numerical experiments show strong promise of the proposed method.

**Weaknesses:**

The generalization performance/robustness of the proposed method is not completely clear. One challenge in causal discovery is the sensitiveness of the learned causal graph towards perturbation of the distributions, in the presence of weak causal link.

Post-rebuttal: I thank the authors for their response and additional experiments for the case of weak causal link. I am increasing my score to 8.

**Questions:**

It could be more convincing to analyze the sensitivity of the proposed model in accordance to perturbation of the input parameters, in particular in the presence of weak causal link.

---

> ### Author Response · Authors · 2023-11-17
> **Response**
>
> We thank the reviewer for their insights and are glad they found the model architecture convincing, and our experiments extensive. To address their central question, we have updated the paper to add additional experimental results (**Appendix A.5.4**) which probed the effect of perturbing the synthetic AD data on the output of the numerical causal discovery algorithms.
>
> We provide a high-level summary of these results below.
>
> ___
>
> | Algorithm | L1 | L2 | L3 |
> |-----------|--------------|------------|-----------------|
> | DAG-GNN       | 0.65        | 0.61      | 0.71           |
> | NOTEARS   | 0.71        | 0.83    | 0.83           |
> | CMA | **0.28**        | **0.30**    | **0.36**            |
>
>  *There are three noise levels whereby Gaussian noise is added with mean 0
> and standard deviations of 0.4 for L1, 0.8 for L2, and 1.2 for L3.*
>
> [As can be seen in this anonymised figure,](https://i.imgur.com/G5Pej61.png) we found that with increasingly weaker causal relationships, both NOTEARS and DAG-GNN struggle to learn the causal graph, though the DAG-GNN algorithm maintains a more consistent performance throughout. The CMA outperforms both algorithms, and we believe this is partly due to the metadata-based modules of the framework, which can still propose reasonable causal structures.

---

### Official Review · Reviewer_Jpz4 · 2023-11-01

**Soundness:** 3 good
**Presentation:** 3 good
**Contribution:** 3 good
**Rating:** 8
**Confidence:** 3

**Summary:**

The paper introduces a novel framework that synergizes the metadata-based reasoning capabilities of LLMs with the data-driven modeling of Deep Structural Causal Models for causal discovery. The authors evaluated the performance on benchmarks and real-world tasks. Real-world tasks were related to modeling the clinical and radiological phenotype of Alzheimer’s Disease. The experimental results indicate that the CMA can outperform previous approaches to causal discovery and derive new insights regarding causal relationships.

**Strengths:**

- the paper proposes an original approach to causal modeling
 - the paper has a good quality: benchmark and real-world tasks are considered, showing promising results in both cases
 - the paper is well structured and written, making it easy to follow
 - the topic is a relevant topic on which much research is being invested, given the new capabilities and opportunities LLMs provide to causal modeling

**Weaknesses:**

- we have not found strong weaknesses in the paper

**Questions:**

1. While the authors do a good job regarding the related work, we consider this could be further enhanced by citing surveys that provide an overview of the relevant topics and domains. E.g., the authors may be interested on the following works: (a) for causal deep modelling: Li, Zongyu, and Zhenfeng Zhu. "A survey of deep causal model." arXiv preprint arXiv:2209.08860 (2022); (b) for Alzheimer disease neuroimaging: Varghese, Tinu, et al. "A review of neuroimaging biomarkers of Alzheimer’s disease." Neurology Asia 18.3 (2013): 239. and Márquez, Freddie, and Michael A. Yassa. "Neuroimaging biomarkers for Alzheimer’s disease." Molecular neurodegeneration 14 (2019): 1-14; and (c) Huang, Yiyi, et al. "Benchmarking of data-driven causality discovery approaches in the interactions of arctic sea ice and atmosphere." Frontiers in big Data 4 (2021): 642182, and Kretschmer, Marlene, et al. "Using causal effect networks to analyze different Arctic drivers of midlatitude winter circulation." Journal of climate 29.11 (2016): 4069-4081.
2. In the related work section, the authors may consider weighting the views and findings regarding LLMs and causality expressed in the following paper: Zečević, Matej, et al. "Causal parrots: Large language models may talk causality but are not causal." arXiv preprint arXiv:2308.13067 (2023).
3. When reporting results in Section 4.1, the authors measure average data likelihood and the deviation. It would be helpful to have some reference value to understand whether the reported values are good or not and why.
4. How is the threshold for DAG-GNN selected?
5. Table 1: align results to the right so that differences in magnitude are quickly visualized.
6. Table 2: add up/down arrows near the reported metrics, indicating greater/lower is better.
7. Table 2: for some algorithms (TCDF, NOTEARS (Temporal), NOTEARS (Temporal)), the authors report results only for the Arctic sea ice dataset, but no clarification is provided as to why no results are reported for the Alzheimer’s disease and Sangiovese datasets.

---

> ### Author Response · Authors · 2023-11-17
> **Response 1/2**
>
> We thank the reviewer for their questions and insights. We are glad they found our paper original, of good quality, well-structured, and relevant. To address their questions, we have made a number of amendments to the manuscript, including extending the ‘Related works’ section and incorporating several clarifications throughout.
>
>
> ___
> > While the authors do a good job regarding the related work, we consider this could be further enhanced by citing surveys that provide an overview of the relevant topics and domains. E.g., the authors may be interested on the following works: [..]
>
> Thank you for your kind words. We have included a reference to the causal deep modelling survey at the beginning of our discussion of LLMs and causality in the related works section.
> The Alzheimer’s disease (AD) neuroimaging references provide a nice exposition of the 18F-AV-45 tracer in particular, which is a variable we consider in our AD models. We therefore additionally include these references in the ‘Experiments’ section.
> Finally, Huang et al.’s work on data-driven causal discovery for the arctic sea ice dataset is referenced in **Section 4.2: ‘Benchmarking Experiments’.**
>
> ___
> >  In the related work section, the authors may consider weighting the views and findings regarding LLMs and causality expressed in the following paper: Zečević, Matej, et al. "Causal parrots: Large language models may talk causality but are not causal." arXiv preprint arXiv:2308.13067 (2023).
>
> Thank you for pointing us to this contemporary work. We have added a discussion of this paper in a new section on the use of LLMs for causal reasoning. This can be found in the ‘Related work’ section of the main manuscript.
>
> ___
> > When reporting results in Section 4.1, the authors measure average data likelihood and the deviation. It would be helpful to have some reference value to understand whether the reported values are good or not and why.
>
> In **Section 4.1**, the most meaningful comparison lies in the relative difference in data likelihoods under each model. We have amended the manuscript to highlight that the model that matches the data-generating process acts as the reference likelihood.
> We then demonstrate that making incorrect modelling assumptions about the data-generating process leads to a lower data likelihood. Finally, the difference in data likelihoods is compared with a pairwise Tukey HSD test. Assessing relative model performance in this way has been employed by other related works[1].
>
> ___
> > How is the threshold for DAG-GNN selected?
>
> We assessed two thresholds for the DAG-GNN algorithm and chose the threshold with the best performance for inclusion in the main manuscript. We began with a threshold of 0.3 as per the DAG-GNN paper by Yu et al.[2], who themselves selected this parameter based on prior work by Zheng et al.[3]. We also assessed a threshold of 0.1, which had worse performance on our benchmarks. Results of the DAG-GNN algorithm with different thresholds can be found in **Tables 6, 9, and 12** of **Appendix Sections A.3.3, A.4.3, and A.5.3**, respectively. We had also conducted a similar hyperparameter tuning experiment for the NOTEARS algorithm and reported those results in the same tables.
>
> ___
> > 1.  Table 1: align results to the right so that differences in magnitude are quickly visualized.
> > 2.  Table 2: add up/down arrows near the reported metrics, indicating greater/lower is better.
>
> We very much appreciate the formatting/readability suggestions. We’ve amended the tables and re-ordered some of the columns in **Table 1** to improve clarity.
>
> > Table 2: for some algorithms (TCDF, NOTEARS (Temporal), NOTEARS (Temporal)), the authors report results only for the Arctic sea ice dataset, but no clarification is provided as to why no results are reported for the Alzheimer’s disease and Sangiovese datasets.
>
> As we state in **Appendix A.3.2**, the arctic sea ice dataset includes optional temporal data, which the TCDF algorithm expects by default. The Alzheimer’s disease and Sangiovese benchmarks are cross-sectional datasets, which precludes the use of the TCDF algorithm on them. The ‘temporal’ versions of the NOTEARS and DAG-GNN algorithms require that we pre-process the time-series data as per Huang et al.[4] and Kiciman et al.[5]; a detailed description can also be found in Appendix A.3.2.
>
>
> For clarity, we have now added an explicit reference to Appendix A.3.2 in the main manuscript under **Section 4.2**.

---

> > ### Comment · Reviewer_Jpz4 · 2023-11-23
> >
> > We authors have tackled our comments. We have reviewed the comments from other reviewers and have no further observations.

---

> ### Author Response · Authors · 2023-11-17
> **Response 2/2**
>
> **References**:
>
> *1. Pawlowski, N., Coelho de Castro, D., & Glocker, B. (2020). Deep structural causal models for tractable counterfactual inference. Advances in Neural Information Processing Systems, 33, 857-869.
> 2. Yu, Y., Chen, J., Gao, T., & Yu, M. (2019, May). DAG-GNN: DAG structure learning with graph neural networks. In International Conference on Machine Learning (pp. 7154-7163). PMLR.
> 3. Zheng, X., Aragam, B., Ravikumar, P. K., & Xing, E. P. (2018). Dags with no tears: Continuous optimization for structure learning. Advances in neural information processing systems, 31.
> 4. Huang, Y., Kleindessner, M., Munishkin, A., Varshney, D., Guo, P., & Wang, J. (2021). Benchmarking of data-driven causality discovery approaches in the interactions of arctic sea ice and atmosphere. Frontiers in big Data, 4, 642182.
> 5. Kıcıman, E., Ness, R., Sharma, A., & Tan, C. (2023). Causal reasoning and large language models: Opening a new frontier for causality. arXiv preprint arXiv:2305.00050.*

---

### Public Comment · ~Tao_Feng2 · 2024-05-15
**code url is expired**

I notice that the code URL [https://anonymous.4open.science/r/causal_modelling_agent-F443/] is expired. It would be helpful if a valid GitHub URL is provided.

---

### Public Comment · ~Wanqi_Zhou2 · 2024-05-24
**Code**

It would be better to give a demo code following the pipeline given in the paper.
Thank you.

---

### Public Comment · ~Huaming_Du1 · 2024-06-17
**Code**

Please consider open-sourcing the code to foster community development. Thank you.

---

### Meta-Review · Area_Chair_ogi9 · 2023-12-06

**Metareview:**

This paper introduces the Causal Modelling Agent (CMA), a framework combining the reasoning capabilities of Large Language Models (LLMs) with the data-driven modeling of Deep Structural Causal Models (DSCMs) for causal discovery. Evaluations show the CMA's superior performance over previous approaches, and its application to Alzheimer's Disease biomarkers yields valuable new insights. Despite that the graph learned in the proposed way is not guaranteed to be correct under certain assumptions, the idea is novel to the community and could inspire future approaches that leverage LLMs to perform causal inference. I suggest acceptance of this paper.

**Justification For Why Not Higher Score:**

One limitation is that the correctness of the method cannot be theoretically guaranteed.

**Justification For Why Not Lower Score:**

novel idea and strong empirical results.

---

### Decision · Program_Chairs · 2024-01-16

Accept (poster)